

# Critical loop models are exactly solvable

**Rongvoram Nivesvivat[1][⋆], Sylvain Ribault[2][†] and Jesper Lykke Jacobsen[2,3,4][‡]**

**1** Yau Mathematical Sciences Center, Tsinghua University, Beijing, 100084 China
**2** Institut de physique théorique, CEA, CNRS, Université Paris-Saclay
**3** Laboratoire de Physique de l'École Normale Supérieure, ENS, Université PSL,
CNRS, Sorbonne Université, Université de Paris
**4** Sorbonne Université, École Normale Supérieure,
CNRS, Laboratoire de Physique (LPENS)

⋆ rongvoram.n@outlook.com , † sylvain.ribault@ipht.fr , ‡ jesper.jacobsen@ens.fr

## Abstract

In two-dimensional critical loop models, including the $O(n)$ and Potts models, the spectrum is exactly known, as are a few structure constants or ratios thereof. Using numerical conformal bootstrap methods, we study 235 of the simplest 4-point structure constants. For each structure constant, we find an analytic expression as a product of two factors: 1) a universal function of conformal dimensions, built from Barnes' double Gamma function, and 2) a polynomial function of loop weights, whose degree obeys a simple upper bound. We conjecture that all structure constants are of this form. For a few 4-point functions, we build corresponding observables in a lattice loop model. From numerical lattice results, we extract amplitude ratios that depend neither on the lattice size nor on the lattice coupling. These ratios agree with the corresponding ratios of 4-point structure constants.

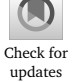

# 1 Introduction and main results

## 1.1 Exact solvability in conformal field theory

**Solvability and degenerate fields in various dimensions**

In conformal field theory in dimensions $d > 2$, the bootstrap approach is effective not only for studying specific models such as the Ising model, but also for carving out the space of unitary theories. However, most results are numerical. No interacting model has been exactly solved, and the known analytic results are about unitarity bounds, or about features that emerge in some limits [1].

The situation is a bit better in planar $N = 4$ super Yang–Mills theory. This $d = 4$ CFT can be mapped to an integrable spin chain, and this reduces the determination of the spectrum to solving functional equations [2]. Computing correlation functions is a harder problem, which may be addressed by combining integrability with numerical bootstrap techniques.

In $d = 2$, on the other hand, a number of CFTs have been exactly solved, starting with Virasoro minimal models in the 1980s. These solutions rely not only on local conformal symmetry, but also on the existence of two independent degenerate fields. In minimal models, two degenerate fields are needed for generating the whole spectrum by repeated fusion; in Liouville theory, two degenerate fields lead to two independent shift equations for structure constants, which are thereby uniquely determined [3,4]. To some extent, similar techniques can also be applied to CFTs with extended chiral symmetry algebras [5,6]. We however restrict our attention to Virasoro-CFTs.

In this article, we will focus on exactly solving critical loop models: a class of Virasoro-CFTs that only have one degenerate field. This includes theories such as the Potts and $O(n)$ models, which are much richer than minimal models and Liouville theory. Another motivation is that in $d > 2$ CFT, there exist weight-shifting operators that are the analogs of degenerate fields, although there is nothing like two independent degenerate fields [7,8]. Solving $d = 2$ CFTs with one degenerate field would therefore bring exact solvability a bit closer to $d > 2$ CFT.

**Hints of exact solvability in loop models**

Originally, loop models arose by reformulating certain statistical lattice models in terms of loops. In particular, in the $Q$-state Potts model, local spins can be traded for Fortuin–Kasteleyn clusters [9], and then for loops defined as the boundaries of these clusters. Similarly, the $O(n)$ model, whose local variables are $n$-dimensional vectors and which has an $O(n)$ global symmetry, can be reformulated in terms of loops [10]. The critical limits of these models may be studied with the methods of conformal field theory, leading to the exact determination of certain conformal dimensions [11]. Mathematically, random curves have been studied in probability theory since the 2000s: open curves may be constructed using Schramm–Loewner Evolutions [12, 13], and closed loops using Conformal Loop Ensembles [14]. This has led to proofs of some previously known results, and to the exact determination of the backbone exponent [15].

The first and biggest hint that critical loop models may be exactly solvable, is that their full spectra of conformal dimensions can be exactly determined [16]. This does not immediately lead to the determination of structure constants, but this gives us access to high-precision numerical bootstrap methods [17].

The existence of a degenerate field implies exact relations between structure constants [18, 19], which may be interpreted in terms of an extension of local conformal symmetry called interchiral symmetry [20, 21]. However, in the absence of two independent degenerate fields, these relations are far from enough for solving critical loop models, although they help simplify bootstrap equations by reducing the number of unknowns.

The first exact result for structure constants was the conjecture by Delfino and Viti for the 3-point cluster connectivity in the Potts model [22]. While it was not immediately clear that the conjecture was exactly true, it was later found to agree with high-precision numerical bootstrap results [23], and generalized to 3-point functions of arbitrary diagonal fields [24, 25]. According to these results, structure constants of diagonal fields in critical loop models coincide with structure constants of Liouville theory with $c \leq 1$. (This does not mean that critical loop models are related to Liouville theory: the former have a discrete spectrum, the latter a continuous spectrum.)

In critical loop models, diagonal fields are however only a small part of the story. Most fields are non-diagonal, i.e. they have nonzero conformal spins. To be precise, we also call non-diagonal a spinless field if its operator product with a degenerate field yields fields with nonzero spins. For structure constants of such non-diagonal spinless fields, Liouville structure constants may be used as an ansatz, and some ratios of structure constants were found to differ from this ansatz by rational functions of the Potts model's number of states [26]. Remarkably, these exact results hold not only in the critical limit, but also on lattices of finite size.

These are strong but limited hints of exact solvability. To actually determine all structure constants, what is missing is an ansatz to which we could compare numerical bootstrap results. As soon as fields with nonzero conformal spins are involved, this ansatz cannot come from Liouville theory.

## 1.2 Structure constants in loop models

**Reference structure constants and normalized structure constants**

We will now write reference two- and 3-point structure constants in critical loop models. We will provide some justification for these expressions in Section 2, but no derivation: these expression emerged by trial and error from analytic considerations, and comparison with numerical bootstrap results.

First let us define the critical loop models' fields. We write the central charge as

$$c = 13 - 6\beta^2 - 6\beta^{-2}, \qquad \text{with} \qquad \begin{cases} \Re\beta^2 > 0, \\ \beta^2 \notin \mathbb{Q}, \end{cases} \tag{1.1}$$

and we introduce standard notations for the conformal dimension $\Delta$ and momentum $P$,

$$\Delta = P^2 - P_{(1,1)}^2, \qquad \Delta_{(r,s)} = P_{(r,s)}^2 - P_{(1,1)}^2, \qquad P_{(r,s)} = \tfrac{1}{2}\left(-\beta r + \beta^{-1} s\right). \tag{1.2}$$

We introduce the following primary fields, with left- and right-moving conformal dimensions $(\Delta, \bar{\Delta})$:

| Name | Notation | Parameters | $(\Delta, \bar{\Delta})$ |
|------|----------|------------|--------------------------|
| Degenerate | $V_{\langle r,s\rangle}^d$ | $r = 1; s \in \mathbb{N}^*$ | $\left(\Delta_{(r,s)}, \Delta_{(r,s)}\right)$ |
| Diagonal | $V_P$ | $P \in \mathbb{C}$ | $\left(P^2 - P_{(1,1)}^2, P^2 - P_{(1,1)}^2\right)$ |
| Non-diagonal | $V_{(r,s)}$ | $r \in \frac{1}{2}\mathbb{N}^*; s \in \frac{1}{r}\mathbb{Z}$ | $\left(\Delta_{(r,s)}, \Delta_{(-r,s)}\right)$ |

(1.3)

The parameters $r,s$ are Kac table indices: in the case of degenerate fields, they are strictly positive integers. In critical loop models, the spectrum contains a one-parameter family of degenerate fields, which we take by convention to be $V_{\langle 1,s\rangle}^d$ (rather than $V_{\langle r,1\rangle}^d$). As we review in Section 2.1, the existence of these degenerate fields constrains non-diagonal fields to obey $r \in \frac{1}{2}\mathbb{N}^*$. Another constraint is that the conformal spin be integer, $rs \in \mathbb{Z}$.

Our list of fields leads to a working definition of critical loop models: we define a correlation function in critical loop models as a correlation function of these fields, whose decompositions into conformal blocks are discrete sums over the same set of fields. In particular, although we allow diagonal fields with arbitrary momenta $P \in \mathbb{C}$, we never integrate over $P$. At the time of this writing, it is not clear whether critical loop models are CFTs obeying axioms such as the existence of an associative operator product expansion [25, 27]. Nevertheless, our working definition allows us to study their 4-point function by numerically solving crossing symmetry.

For diagonal fields, the critical loop models' two- and 3-point structure constants formally coincide with those of Liouville theory with $c \leq 1$, although critical loop models make sense under the less demanding condition $\Re c < 13$ [4, 25]:

$$B_P = \prod_{\pm,\pm} \Gamma_\beta^{-1}\left(\beta^{\pm 1} \pm 2P\right), \qquad C_{P_1,P_2,P_3} = \prod_{\pm,\pm,\pm} \Gamma_\beta^{-1}\left(\tfrac{\beta+\beta^{-1}}{2} \pm P_1 \pm P_2 \pm P_3\right), \tag{1.4}$$

where $\Gamma_\beta^{-1}(x) = \frac{1}{\Gamma_\beta(x)}$. Barnes' double Gamma function $\Gamma_\beta$ obeys $\Gamma_{\beta^{-1}}(x) = \Gamma_\beta(x)$ as well as the shift equations

$$\frac{\Gamma_\beta(x+\beta)}{\Gamma_\beta(x)} = \sqrt{2\pi}\frac{\beta^{\beta x-\frac{1}{2}}}{\Gamma(\beta x)}, \qquad \frac{\Gamma_\beta(x+\beta^{-1})}{\Gamma_\beta(x)} = \sqrt{2\pi}\frac{\beta^{\frac{1}{2}-\beta^{-1}x}}{\Gamma(\beta^{-1}x)}. \tag{1.5}$$

For non-diagonal fields, we define the reference 2-point structure constant

$$B_{(r,s)}^{\text{ref}} = \frac{(-)^{rs}}{2\sin\left(\pi(\text{frac}(r)+s)\right)\sin\left(\pi(r+\beta^{-2}s)\right)}\prod_{\pm,\pm}\Gamma_\beta^{-1}\left(\beta \pm \beta r \pm \beta^{-1}s\right), \tag{1.6}$$

where $\mathrm{frac}(r) \in \{0, \frac{1}{2}\}$ is the fractional part of $r \in \frac{1}{2}\mathbb{N}^*$. This is formally ill-defined if $r \in \mathbb{N}^*$ and $s \in \mathbb{Z}$, but can be regularized by taking a limit from generic values of $s$. We define the reference 3-point structure constant

$$C^{\text{ref}}_{(r_1,s_1)(r_2,s_2)(r_3,s_3)} = \prod_{\epsilon_1,\epsilon_2,\epsilon_3=\pm} \Gamma_\beta^{-1}\left(\frac{\beta+\beta^{-1}}{2} + \frac{\beta}{2}\left|\sum_i \epsilon_i r_i\right| + \frac{\beta^{-1}}{2}\sum_i \epsilon_i s_i\right), \tag{1.7}$$

which is also valid if one or more fields are diagonal, provided we use the identification

$$V_P = V_{(0,2\beta P)}. \tag{1.8}$$

In particular, $C^{\text{ref}}_{(0,2\beta P_1)(0,2\beta P_2)(0,2\beta P_3)} = C_{P_1,P_2,P_3}$.

Whenever we determine some structure constant numerically, for example a 3-point structure constant $C$, we can divide it by the corresponding reference structure constant, and obtain what we will call a normalized structure constant

$$C^{\text{norm}} = \frac{C}{C^{\text{ref}}}. \tag{1.9}$$

Our general expectation is that $C^{\text{norm}}$ is simpler than $C$, to the extent that we can determine its analytic expression from numerical results.

**Polynomial factors**

Let us write the decomposition of a 4-point function into conformal blocks $\mathcal{G}^{(x)}_{\Delta,\bar{\Delta}}$ in one of three possible channels:

$$\left\langle \prod_{i=1}^4 V_i(z_i) \right\rangle = \sum_{\Delta,\bar{\Delta}} D^{(x)}_{\Delta,\bar{\Delta}} \mathcal{G}^{(x)}_{\Delta,\bar{\Delta}}(z_1,z_2,z_3,z_4), \quad \text{with} \quad x \in \{s,t,u\}. \tag{1.10}$$

Here $D^{(x)}_{\Delta,\bar{\Delta}}$ is a 4-point structure constant. We now define $d^{(x)}_{\Delta,\bar{\Delta}}$ as the corresponding normalized structure constant. To be precise, in the case of an $s$-channel 4-point structure constant for a non-diagonal $s$-channel field $V_{(r,s)}$, this definition amounts to

$$D^{(s)}_{(r,s)} = \frac{C^{\text{ref}}_{(r_1,s_1)(r_2,s_2)(r,s)} C^{\text{ref}}_{(r,s)(r_3,s_3)(r_4,s_4)}}{B^{\text{ref}}_{(r,s)}} d^{(s)}_{(r,s)}. \tag{1.11}$$

Our main claim is the conjecture in Section 2.3, which states that $d^{(x)}_{(r,s)}$ is a polynomial in the loop weight

$$n = -2\cos\left(\pi\beta^2\right), \tag{1.12}$$

whose coefficients are $\beta$-independent numbers, and whose degree obeys

$$\deg_n d^{(x)}_{(r,s)} \leq r(r-1). \tag{1.13}$$

If some of the four fields are diagonal, $V_i = V_{P_i}$, or if the decomposition involves a diagonal field $V_{P_x}$ in addition to the non-diagonal fields, then $d^{(x)}_{(r,s)}$ also depends on the corresponding loop weights

$$w(P) = 2\cos(2\pi\beta P). \tag{1.14}$$

The dependence on $w_i = w(P_i)$ is again polynomial. The dependence on $w_x = w(P_x)$ becomes polynomial after we subtract a rational term that is needed for the 4-point function to be holomorphic in $P_x$, see Eq. (2.61). From now on, we will use the notation $w_i$ for the weights $w_1, w_2, w_3, w_4$ (one of them or all four), and $w_x$ for the channel weights $w_s, w_t, w_u$ (one of them or all three).

### 1.3 Conformal bootstrap

**Solutions of crossing symmetry equations**

In critical loop models, given four fields $V_1, V_2, V_3, V_4$ defined by their conformal dimensions, the crossing symmetry equations for the 4-point function $\left\langle \prod_{i=1}^{4} V_i(z_i) \right\rangle$ generally have several linearly independent solutions. For example, in the Potts model, there are four independent correlation functions of the type $\left\langle \prod_{i=1}^{4} V_{P_{(0,\frac{1}{2})}} \right\rangle$, which can be interpreted as 4-point cluster connectivities [28]. More generally, the global symmetries of the Potts and $O(n)$ models lead us to expect finite-dimensional spaces of 4-point functions. Recently, it was conjectured that the space of solutions of crossing symmetry equations for the 4-point function $\left\langle \prod_{i=1}^{4} V_{(r_i,s_i)} \right\rangle$ has a dimension $\sum_{i=1}^{4} r_i^2 + d_0$, with $d_0 \in \{\frac{1}{2}, 1, 2\}$ a known $r_i$-dependent number [29]. The case of a diagonal 4-point function corresponds to $r_i = 0$ and $d_0 = 1$. Cluster connectivities do not obey this formula because they can have several diagonal fields propagating in the same channel, see Section 3.2 for more details.

Given a 4-point function $\left\langle \prod_{i=1}^{4} V_i(z_i) \right\rangle$, not all solutions of crossing symmetry can have polynomial structure constants $d_{(r,s)}^{(x)}$, as this property is not preserved by taking linear combinations with non-polynomial coefficients. Fortunately, there is a natural basis of solutions, parametrized by combinatorial maps [29]: we conjecture that solutions in this basis have polynomial structure constants.

For example, in the case of $\left\langle V_{(\frac{3}{2},0)} V_{(\frac{1}{2},0)} V_{(1,0)} V_{(1,0)} \right\rangle$, the space of solutions of crossing symmetry equations has dimension 5. Let us call $(Z_1, \ldots, Z_5)$ the basis of solutions that correspond to the following combinatorial maps:



$$Z_1\,, \qquad Z_2\,, \qquad Z_3\,, \qquad Z_4\,, \qquad Z_5\,. \tag{1.15}$$

In a combinatorial map, the primary field $V_{(r,s)}$ corresponds to a vertex of valency $2r$. The maps do not depend on the second Kac index $s$.

Each one of the solutions $Z_1, \ldots, Z_5$ can in principle be singled out by imposing enough constraints on the structure constants $d_{(r,s)}^{(x)}$, in addition to the crossing symmetry equations. We consider constraints that amount to setting finitely many structure constants to zero, depending on their first Kac index:

$$\forall x \in \{s, t, u\}, \quad r < \sigma^{(x)} \implies d_{(r,s)}^{(x)} = 0, \tag{1.16}$$

where the triple $\sigma = \left(\sigma^{(s)}, \sigma^{(t)}, \sigma^{(u)}\right)$ is called a signature. There is a combinatorial definition of the signature of a combinatorial map: $2\sigma^{(x)}$ is the minimum number of lines that are crossed by an $x$-channel loop. For example:

$$\sigma^{(s)} = 2\,, \qquad \sigma^{(t)} = \tfrac{1}{2}\,, \qquad \sigma^{(u)} = \tfrac{3}{2}\,. \tag{1.17}$$

Then the solution that corresponds to a map obeys the constraints that correspond to that map's signature. However, these constraints do not always uniquely characterize that solution. In our example, the signatures are:

| Map | $Z_1$ | $Z_2$ | $Z_3$ | $Z_4$ | $Z_5$ |
|---|---|---|---|---|---|
| Signature | $(1, \frac{3}{2}, \frac{3}{2})$ | $(1, \frac{3}{2}, \frac{3}{2})$ | $(2, \frac{1}{2}, \frac{3}{2})$ | $(2, \frac{3}{2}, \frac{1}{2})$ | $(0, \frac{5}{2}, \frac{5}{2})$ |

(1.18)

Since $Z_1$ and $Z_2$ have the same signature, the corresponding constraints only characterize a two-dimensional space of solutions, and not $Z_1$ and $Z_2$ individually. On the other hand, the solutions that correspond to $Z_3$, $Z_4$ and $Z_5$ are each singled out by the respective constraints.

If a solution has a diagonal field that propagates in the $s$-channel, then by definition $\sigma^{(s)} = 0$. Moreover, we then have $\sigma^{(t)} = \sigma^{(u)} = \max(r_1, r_2) + \max(r_3, r_4)$. Based on numerical results, we then conjecture the degrees of the normalized 4-point structure constants in the weight $w_s$ of the diagonal field:

$$r > 0 \implies \boxed{\deg_{w_s} d^{(s)}_{(r,s)} \le \left\lfloor \frac{r}{2} - \sigma^{(t)} \right\rfloor}, \quad \boxed{\deg_{w_s} d^{(t)}_{(r,s)} = \deg_{w_s} d^{(u)}_{(r,s)} = r - \sigma^{(t)}.} \quad (1.19)$$

By convention, a polynomial of negative degree is zero. The structure constant for the diagonal $s$-channel field is not subject to the bound on $\deg_{w_s} d^{(s)}_{(r,s)}$, and is set to one as a way to fix the solution's overall normalization.

**Evidence from numerical bootstrap results**

We can solve crossing symmetry numerically in order to determine 4-point structure constants to arbitrary precision [30]. Instead of counting solutions as in [29], we are now focussing on specific solutions, in order to find analytic expressions for their structure constants. If a structure constant $d(w)$ is polynomial of degree $k$, it is enough to compute it for $k+2$ values of the variable $w$, so that we can determine its $k+1$ coefficients and test the resulting polynomial. Depending on the solution, our normalized structure constants are polynomials in 1 to 8 loop weights: the weight $n$ in all cases, and some or all of the weights $w_1, w_2, w_3, w_4, w_s, w_t, w_u$. The more variables, the more coefficients to be determined! The situation is not as bad as it looks, for a number of reasons:

- The only correlation function that involves more than one of the three channel weights $w_s, w_t, w_u$ is the 4-point function of diagonal fields. In this 4-point function, these three weights appear in different terms [25], so that we are effectively dealing with three polynomials of one variable, which is better than one polynomial of three variables.

- Furthermore, in the crossing symmetry equations, the channel weight $w_s$ only affects one conformal block. As a result, determining the dependence on $w_s$ is effectively free computationally: determining structure constants for several values of $w_s$ is not sensibly longer than for one value. In contrast, the variables $n, w_1, w_2, w_3, w_4$ affect all conformal blocks, so all calculations must be done from scratch for each one of their values.

- Showing that a function is a polynomial can be done variable by variable. Having multiple variables is inconvenient only when it comes to determining the coefficients.

Eventually, writing a normalized structure constant as a polynomial means reducing it to a list of coefficients. We find that these coefficients are not all rational numbers, as they can involve irrational expressions of the type $\cos(\pi s)$, where $s \in \mathbb{Q}$ is a combination of the second Kac indices of the relevant fields. In the following table, we list the 4-point functions that we

have investigated, and indicate the number of polynomials that we have determined (modulo discrete symmetries). The number $r_{\max}$ is the largest number such that we know *all* structure constants $D_{(r,s)}^{(x)}$ with $r \leq r_{\max}$, and also the largest number such that we know *some* structure constants of the type $D_{(r_{\max},s)}^{(x)}$: when these two numbers differ, both are given.

| Four-point function | $\sigma$ | Map | Weights | $r_{\max}$ | #poly | Eq. |
|---|---|---|---|---|---|---|
| $\left\langle V_{P_1} V_{P_2} V_{P_3} V_{P_4} \right\rangle$ | $(0,0,0)$ | | $n, w_{s,t,u}$ $w_{1,2,3,4}$ | 3 | 6 | (3.7) |
| $\left\langle V_{P_{(0,\frac{1}{2})}}^4 \right\rangle$ | $(0,0,0)$ | | $n^2$ | 6\|10 | 24 | (3.13) |
| $\left\langle V_{(1,0)} V_{P_1} V_{P_2} V_{P_3} \right\rangle$ | $(0,1,1)$ | | $n, w_s$ $w_{1,2,3}$ | 2\|3 | 12 | (3.22) |
| $\left\langle V_{(\frac{1}{2},0)}^2 V_{P_1} V_{P_2} \right\rangle$ | $(0,\frac{1}{2},\frac{1}{2})$ | | $n, w_s$ $w_{1,2}$ | 3 | 11 | (3.25) |
| $\left\langle V_{(\frac{3}{2},0)} V_{(\frac{1}{2},0)} V_{P_1} V_{P_2} \right\rangle$ | $(0,\frac{3}{2},\frac{3}{2})$ | | $n, w_s$ $w_{1,2}$ | 3 | 8 | (3.28) |
| $\left\langle V_{(1,1)} V_{(\frac{1}{2},0)}^2 V_P \right\rangle$ | $(\frac{1}{2},1,\frac{1}{2})$ | | $n$ $w$ | 3 | 8 | (3.30) |
| $\left\langle V_{(\frac{1}{2},0)}^4 \right\rangle$ | $(0,1,1)$ | | $n, w_s$ | 3\|4 | 17 | (3.32) |
| $\left\langle V_{(\frac{1}{2},0)}^2 V_{(1,0/1)}^2 \right\rangle$ | $(0,\frac{3}{2},\frac{3}{2})$ | | $n, w_s$ | 3 | 8/8 | (3.34) |
| $\left\langle V_{(\frac{1}{2},0)}^2 V_{(1,0)}^2 \right\rangle$ | $(1,\frac{3}{2},\frac{1}{2})$ | | $n$ | 3\|4 | 20 | (3.36) |
| $\left\langle V_{(\frac{1}{2},0)}^2 V_{(1,1)}^2 \right\rangle$ | $(1,\frac{3}{2},\frac{1}{2})$ | | $n$ | 3 | 18 | (3.37) |
| $\left\langle V_{(1,0)}^4 \right\rangle, \left\langle V_{(1,1)}^4 \right\rangle$ | $(0,2,2)$ | | $n, w_s$ | 3\|4 | 11 | (3.39) |
| $\left\langle V_{(1,0)}^4 \right\rangle, \left\langle V_{(1,1)}^4 \right\rangle$ | $(2,1,1)$ | | $n$ | 3\|4 | 16 | (3.40) |
| $\left\langle V_{(\frac{3}{2},0)} V_{(1,1)} V_{(1,0)} V_{(\frac{1}{2},0)} \right\rangle$ | $(\frac{3}{2},2,\frac{1}{2})$ | | $n$ | 3 | 13 | (3.42) |
| $\left\langle V_{(\frac{3}{2},\frac{2}{3})} V_{(1,1)} V_{(1,0)} V_{(\frac{1}{2},0)} \right\rangle$ | $(\frac{3}{2},2,\frac{1}{2})$ | | $n$ | 3 | 23 | (3.43) |

| Four-point function | $\sigma$ | Map | Weights | $r_{\max}$ | #poly | Eq. |
|---|---|---|---|---|---|---|
| $\left\langle V_{(\frac{3}{2},0)} V^3_{(\frac{1}{2},0)} \right\rangle$ | $(1,1,1)$ | even | $n$ | 3 | 5 | (3.46) |
| | | odd | $n$ | 3 | 3 | (3.48) |
| $\left\langle V_{(\frac{3}{2},\frac{2}{3})} V^3_{(\frac{1}{2},0)} \right\rangle$ | $(1,1,1)$ | even | $n$ | 3 | 12 | (3.50) |
| | | odd | $n$ | 3 | 12 | (3.52) |

In the last 4 lines, even and odd denote even-spin and odd-spin solutions, see Section 3.5. The total number of polynomials that we have determined in all these examples is 235, as announced in the abstract. In a few examples, let us illustrate the weights on which our 4-point functions depend: weights $w_i$ (in red) associated to vertices, and channel weights $w_x$ (in blue). Anticipating on the lattice approach, we associate each weight to a certain type of loops:

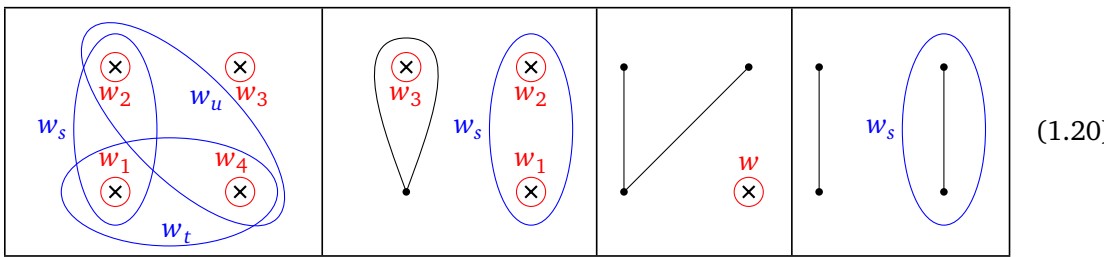

$$(1.20)$$

## 1.4 Lattice approach

**Nienhuis loop model**

We have used the name *critical loop models* for the CFTs that we have been studying, and we will now justify this terminology by comparing our 4-point functions with critical limits of observables in a loop model on a lattice. To be specific, we consider a gas of non-intersecting loops on a honeycomb lattice [10], although different lattice models are expected to share the same critical limit. The partition function is defined as the sum of the weights of all possible loop configurations, where each loop contributes a factor $n$ to the weight, and each vertex a factor 1 or $K$:

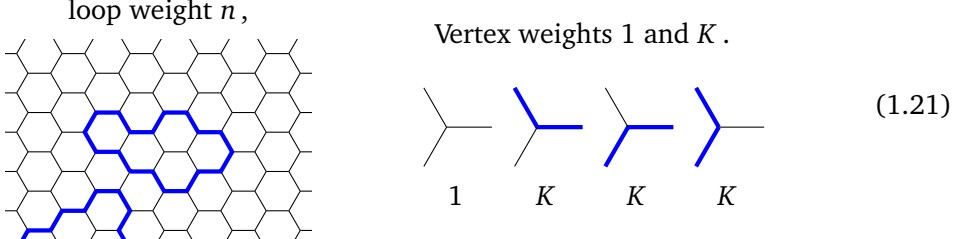

$$(1.21)$$

The critical limit is obtained by sending the lattice size to infinity, and the coupling to the critical value

$$K_c = \frac{1}{\sqrt{2 + \sqrt{2-n}}}, \qquad (1.22)$$

where $n$ is now related to the central charge of the resulting CFT by Eq. (1.12). On this lattice, we also insert the analogs of primary fields, diagonal or non-diagonal. A diagonal primary field $V_P$ simply changes the weight of any loop that goes around it from $n$ to $w(P)$ (1.14). A non-diagonal primary field $V_{(r,s)}$ is the end point of $2r$ open loops, and contributes a weight factor that depends on $s$ and on the relative angles of these loops:

$$
w(P), \qquad \exp \tfrac{i}{2}s \sum_{k=1}^{2r} \theta_k \,. \tag{1.23}
$$

The reader who wonders how we can measure angles, and how $2r$ segments can end at the same point on a trivalent lattice, is invited to consult [29]. While it was long known that the second Kac index $s$ corresponds to the angular momentum, its unambiguous interpretation requires combinatorial maps.

**Results for amplitude ratios**

The loop model allows us to build lattice 4-point functions $C^{\text{loop}}(L, \ell | K, n, w_i, w_x)$, where $L$ is the circumference of our cylindrical lattice (measured in lattice units), and $\ell$ the distance between the two Euclidean time slices where we respectively position the vertices $z_1, z_2$ and $z_3, z_4$. In the critical limit, the geometrical parameters $L, \ell$ go to infinity. The numbers $w_i, w_x$ with $i \in \{1, 2, 3, 4\}$ and $x \in \{s, t, u\}$ are loop weights: external weights $w_i$ for loops around the four fields, and internal weights $w_x$ for loops around two of the fields, if such loops exist, see Figure (1.20). Loops are non-intersecting, and also cannot intersect the segments that end at vertices. In particular, in a given configuration, there cannot be two different loop types among $\{s, t, u\}$. In the transfer matrix formalism, we can compute $C^{\text{loop}}(L, \ell | K, n, w_i, w_x)$ numerically. In practice, we are limited to low values of the lattice width $L \leq 5$.

We decompose the lattice 4-point functions into $s$-channel amplitudes $A_\omega$:

$$
C^{\text{loop}}(L, \ell | K, n, w_i, w_x) = \sum_{\omega \in S(L)} A_\omega(L | K, n, w_i, w_x) \left( \frac{\Lambda_\omega(L | K, n, w_s)}{\Lambda_{\max}(L | K, n, w_s)} \right)^\ell , \tag{1.24}
$$

where $(\Lambda_\omega)_{\omega \in S(L)}$ is the spectrum of eigenvalues of the transfer matrix, with $\Lambda_{\max}$ the largest eigenvalue. This decomposition is reminiscent of the decomposition of a CFT 4-point function into $s$-channel conformal blocks, and the transfer matrix depends on the weights $n, w_s$ of contractible loops and $s$-loops, but not on the other loop weights. Our main lattice result is that a ratio of amplitudes with different internal weights $w_x, w'_x$ agrees with the corresponding ratio of $s$-channel 4-point structure constants,

$$
\boxed{A_{(r,s),\rho}\left(L | K, n, w_i, w_x : w'_x\right) = D^{(s)}_{(r,s)}\left(n, w_i, w_x : w'_x\right),} \tag{1.25}
$$

where we use the notation

$$
f(x : x') = \frac{f(x)}{f(x')} \,. \tag{1.26}
$$

In this equation, the eigenvalue parameter is rewritten as $\omega = (r,s), \rho$ where $(r,s)$ are Kac table indices, and $\rho$ distinguishes states that share the same indices. Our result implies that

the amplitude ratio depends neither on $\rho$, nor on the lattice size $L$, nor on the lattice coupling $K$. Furthermore, since reference structure constants do not depend on $w_x$, the ratio of 4-point structure constants reduces to a ratio of normalized structure constants, which are rational functions of loop weights.

This result is supported by strong numerical evidence in the cases of 4-point functions of the types $\left\langle \prod_{i=1}^{4} V_{P_i} \right\rangle$ and $\left\langle V_{(\frac{1}{2},0)}^4 \right\rangle$, for the first few values of $(r,s)$ (but hundreds of values of $\omega$). We conjecture that Eq. (1.25) holds for arbitrary 4-point functions and arbitrary eigenvalues $\omega$. Having exact rational formulas for amplitude ratios suggests that such ratios have an algebraic origin, and might be computable from the representation theory of appropriate diagram algebras. This might be a hint that lattice loop models are exactly solvable.

## 2 Analytic constraints

Let us work out the constraints on the spectrum and structure constants that can be derived analytically. In Sections 2.1 and 2.2, we will deal with the constraints that follow from the existence of degenerate fields. Mostly, we will be reviewing and simplifying the treatment of [19]. In Section 2.3, we will assume that 4-point functions are holomorphic in the conformal dimensions of the $s$-channel's diagonal field when there is one. The resulting constraints are new, as the possibility that such a field can have an arbitrary conformal dimension was only recently raised in [25]. In Section 2.4, we will derive simple relations following from field permutations: this is in principle straightforward, but signs of conformal blocks have to be treated carefully.

### 2.1 How degenerate fields constrain the spectrum

**Degenerate fusion rules and integer spin condition**

A degenerate representation of the Virasoro algebra may be characterized by its fusion products with other representations. In particular, for $s \in \mathbb{N}^*$, let us call $\mathcal{V}_{\langle 1,s \rangle}^d$ the degenerate representation with momentum $P_{(1,s)}$, and $\mathcal{V}_P$ the Verma module with momentum $P$. We have the fusion products

$$\mathcal{V}_{\langle 1,1 \rangle}^d \times \mathcal{V}_P = \mathcal{V}_P \,, \tag{2.1a}$$

$$\mathcal{V}_{\langle 1,2 \rangle}^d \times \mathcal{V}_P = \mathcal{V}_{P-\frac{1}{2}\beta^{-1}} + \mathcal{V}_{P+\frac{1}{2}\beta^{-1}} \,, \tag{2.1b}$$

$$\mathcal{V}_{\langle 1,3 \rangle}^d \times \mathcal{V}_P = \mathcal{V}_{P-\beta^{-1}} + \mathcal{V}_P + \mathcal{V}_{P+\beta^{-1}} \,, \tag{2.1c}$$

$$\mathcal{V}_{\langle 1,2 \rangle}^d \times \mathcal{V}_{\langle 1,2 \rangle}^d = \mathcal{V}_{\langle 1,1 \rangle}^d + \mathcal{V}_{\langle 1,3 \rangle}^d \,. \tag{2.1d}$$

These fusion rules constrain the operator product expansions of the corresponding fields.

We define the conformal spin of a field with left and right dimensions $\Delta, \bar{\Delta}$ as

$$S = \bar{\Delta} - \Delta \,. \tag{2.2}$$

The single-valuedness of correlation functions requires $S \in \frac{1}{2}\mathbb{Z}$, and that $S$ is conserved modulo $\mathbb{Z}$ in interactions [4]. In order to study bosonic observables in critical loop models, we adopt the slightly stronger assumption $S \in \mathbb{Z}$. (Fields with half-integer spins such as $V_{(\frac{1}{2},1)}$ are fermionic.)

**Degenerate OPEs of diagonal fields**

Let us consider the OPE $V_{(1,2)}^d V_P$, which involves the diagonal primary field $V_P$. According to the fusion rules, we may obtain four primary fields of left and right dimensions

$(P, \bar{P}) \in \left\{ P - \frac{1}{2}\beta^{-1}, P + \frac{1}{2}\beta^{-1} \right\}^2$. However, for generic values of the momentum $P$, the conformal spin cannot be integer if the left and right dimensions differ. Due to the integer spin condition, our OPE can therefore involve only the two primary fields that are diagonal:

$$V^d_{\langle 1,2 \rangle} V_P \sim V_{P - \frac{1}{2}\beta^{-1}} + V_{P + \frac{1}{2}\beta^{-1}} \,. \tag{2.3}$$

By a similar reasoning, we have

$$V^d_{\langle 1,3 \rangle} V_P \sim V_{P - \beta^{-1}} + V_P + V_{P + \beta^{-1}} \,. \tag{2.4}$$

**Degenerate OPEs of non-diagonal fields**

We now consider non-diagonal primary fields $V_{(r,s)}$ as defined in Table 1.3. For the moment, the Kac table indices $r, s$ are arbitrary complex numbers, which provide a convenient parameterization of the left and right conformal dimensions. In particular, the corresponding conformal spin reads

$$S = rs \,. \tag{2.5}$$

According to the fusion rule (2.1b), four primary fields may appear in the degenerate OPE

$$V^d_{\langle 1,2 \rangle} V_{(r,s)} \subset \sum_{\pm} V_{(r,s\pm1)} + \sum_{\pm} V_{(r\pm\beta^{-2},s)} \,. \tag{2.6}$$

We assume that $V_{(r,s)}$ has integer spin $rs \in \mathbb{Z}$. Since the spin is conserved modulo $\mathbb{Z}$, the fields that appear in the OPE must also have integer spins. For $V_{(r,s\pm1)}$, this constraint amounts to $r \in \mathbb{Z}$. For $V_{(r\pm\beta^{-2},s)}$, it amounts to $s\beta^{-2} \in \mathbb{Z}$. Taken together, these two constraints would imply $rs \in \beta^{-2}\mathbb{Z}$, which cannot hold if $\beta$ is generic. We choose to assume $r \in \mathbb{Z}$, so that $s\beta^{-2} \notin \mathbb{Z}$: the choice $s\beta^{-2} \in \mathbb{Z}$ would be equivalent, although it would make our notations remarkably clumsy. Therefore, we find the OPE

$$V^d_{\langle 1,2 \rangle} V_{(r,s)} \sim \sum_{\pm} V_{(r,s\pm1)} \,, \qquad \text{provided} \qquad r \in \mathbb{Z} \,. \tag{2.7}$$

Similarly, had we considered the degenerate field $V^d_{\langle 1,3 \rangle}$ instead of $V^d_{\langle 1,2 \rangle}$, we would have found

$$V^d_{\langle 1,3 \rangle} V_{(r,s)} \sim V_{(r,s-2)} + V_{(r,s)} + V_{(r,s+2)} \,, \qquad \text{provided} \qquad r \in \tfrac{1}{2}\mathbb{Z} \,. \tag{2.8}$$

We therefore recover known features of critical loop model's spectra [16], as reviewed in [23]: in the Potts model, there is a degenerate field $V^d_{\langle 1,2 \rangle}$, and the non-diagonal fields obey $r \in \mathbb{N}^*$. In the $O(n)$ model, there is a degenerate field $V^d_{\langle 1,3 \rangle}$ but no $V^d_{\langle 1,2 \rangle}$, which allows non-diagonal fields with $r \in \frac{1}{2}\mathbb{N}^*$.

In Table (1.3), we have quoted the least stringent constraints for degenerate and non-diagonal fields: we now add the caveat that in a given correlation function, $V^d_{\langle 1,2 \rangle}$ (or more generally $V^d_{\langle 1,s \rangle}$ with $s \in 2\mathbb{N}^*$) cannot coexist with $V_{(r,s)}$ with $r \in \mathbb{N} + \frac{1}{2}$.

## 2.2 How degenerate fields constrain structure constants

**Structure constants**

We schematically define two- and 3-point structure constants $B_i, C_{ijk}$ by

$$\langle V_1 V_2 \rangle = \delta_{12} B_1 \,, \qquad \langle V_1 V_2 V_3 \rangle = C_{123} \,. \tag{2.9}$$

Three-point structure constants can pick non-trivial signs when the three fields are permuted, depending on the fields' conformal spins $S_i$ [4]:

$$C_{\sigma(1)\sigma(2)\sigma(3)} = \text{sign}(\sigma)^{S_1+S_2+S_3} C_{123}\,. \tag{2.10}$$

Our goal is to constrain these structure constants, by exploiting crossing-symmetry and single-valuedness of 4-point functions that involve the degenerate field $V_{\langle 1,2\rangle}^d$. We will therefore also need the OPE coefficients $C_{(r,s)}^{\pm}$ of that degenerate field, which appear in the OPE

$$V_{\langle 1,2\rangle}^d V_{(r,s)} \sim \sum_{\pm} C_{(r,s)}^{\pm} V_{(r,s\pm 1)}\,. \tag{2.11}$$

**Four-point functions with a degenerate field**

In order to constrain three-point structure constants, the basic idea is to exploit crossing symmetry of 4-point functions that involve degenerate fields [3]. Here we will consider a 4-point function of the type

$$Z = \left\langle V_{\langle 1,2\rangle}^d \prod_{i=1}^{3} V_{(r_i,s_i)} \right\rangle\,. \tag{2.12}$$

As we saw in Eq. (2.7), while the indices $r_i$ take half-integer values in critical loop models, only fields with $r_i \in \mathbb{N}$ can coexist with the degenerate field $V_{\langle 1,2\rangle}^d$. If we had fields with $r_i \in \mathbb{N} + \frac{1}{2}$, we could instead consider $\left\langle V_{\langle 1,3\rangle}^d \prod_{i=1}^{3} V_{(r_i,s_i)} \right\rangle$, but would be technically more complicated. And this is in fact unnecessary, as the same constraints can be derived from $Z$ itself, even though it is not single-valued. Here we are following the spirit of [3], which used the field $V_{\langle 1,2\rangle}^d$ although it does not belong to the spectrum of Liouville theory.

Using the OPEs $V_{\langle 1,2\rangle}^d V_{(r_1,s_1)}$ and $V_{\langle 1,2\rangle}^d V_{(r_3,s_3)}$, we can decompose our 4-point function into $s$- and $t$-channel conformal blocks respectively. The equality of these decompositions is the crossing symmetry equation,

$$Z = \sum_{\pm} D_{\pm}^{(s)} \left| \mathcal{F}_{\pm}^{(s)} \right|^2 = \sum_{\pm} D_{\pm}^{(t)} \left| \mathcal{F}_{\pm}^{(t)} \right|^2\,. \tag{2.13}$$

Here, the $x$-channel chiral conformal blocks $\left( \mathcal{F}_{-}^{(x)}, \mathcal{F}_{+}^{(x)} \right)$ are a basis of solutions of the second-order BPZ equation for $Z$. The modulus square notation indicates that we take a product of a left-moving block with a right-moving block. Schematically, the $s$- and $t$-channel blocks may be represented as follows:

$$
\begin{array}{cc}
\begin{array}{c}
V_{(r_1,s_1)} \qquad\qquad V_{(r_2,s_2)} \\[2pt]
V_{(r_1,s_1\pm 1)} \\[2pt]
V_{\langle 1,2\rangle}^d \qquad\qquad V_{(r_3,s_3)}
\end{array}
&
\begin{array}{c}
V_{(r_1,s_1)} \qquad\qquad V_{(r_2,s_2)} \\[2pt]
V_{(r_3,s_3\pm 1)} \\[2pt]
V_{\langle 1,2\rangle}^d \qquad\qquad V_{(r_3,s_3)}
\end{array}
\\[20pt]
s\text{-channel} & t\text{-channel}
\end{array}
\tag{2.14}
$$

The $s$-channel and $t$-channel bases are related by the fusion transformation

$$\mathcal{F}_{\epsilon_1}^{(s)} = \sum_{\epsilon_3=\pm} F_{\epsilon_1,\epsilon_3} \mathcal{F}_{\epsilon_3}^{(t)}\,, \tag{2.15}$$

which involves the fusing matrix elements

$$F_{\epsilon_1,\epsilon_3} = \frac{\Gamma\left(1 + \epsilon_1 2\beta^{-1}P_1\right)\Gamma\left(-\epsilon_3 2\beta^{-1}P_3\right)}{\prod_{\pm}\Gamma\left(\frac{1}{2} + \epsilon_1\beta^{-1}P_1 \pm \beta^{-1}P_2 - \epsilon_3\beta^{-1}P_3\right)}.$$

(2.16)

Here we use the convention that the non-diagonal field $V_{(r,s)}$ has the left and right momenta

$$(P, \bar{P}) = (P_{(r,s)}, P_{(-r,s)}),$$

(2.17)

so that the shift $s \to s + 1$ amounts to $(P, \bar{P}) \to (P + \frac{1}{2}\beta^{-1}, \bar{P} + \frac{1}{2}\beta^{-1})$. Taken together, the crossing symmetry equation and the fusion transformation imply the compatibility condition

$$\frac{F_{++}F_{--}}{F_{+-}F_{-+}} = \frac{\bar{F}_{++}\bar{F}_{--}}{\bar{F}_{+-}\bar{F}_{-+}}.$$

(2.18)

With our fusing matrix elements, this condition boils down to [19]

$$\left(1 - (-)^{2\sum_{i=1}^{3} r_i}\right)\sin\left(2\pi\beta^{-1}P_1\right)\cos\left(2\pi\beta^{-1}P_2\right)\sin\left(2\pi\beta^{-1}P_3\right) = 0.$$

(2.19)

We assume that this holds not only for $Z$ itself, but also for the 4-point functions that are related to $Z$ by shifts of the type $s_i \to s_i + 2$ i.e. $P_i \to P_i + \beta^{-1}$. A trigonometric factor such as $\sin\left(2\pi\beta^{-1}P_1\right)$ may vanish for some special value of $P_1$, but then it cannot vanish for $P_1 + \beta^{-1}$, assuming $\beta$ is generic. Therefore, the compatibility condition is equivalent to the conservation of $r$ modulo $\mathbb{Z}$,

$$r_1 + r_2 + r_3 \in \mathbb{Z}.$$

(2.20)

If compatibility is obeyed, there exists a solution $Z$ of crossing symmetry, which is unique up to a constant factor. This means that we can determine any ratio of the structure constants $D_{\pm}^{(s)}, D_{\pm}^{(t)}$. In particular,

$$\frac{D_{-}^{(s)}}{D_{+}^{(s)}} = -\frac{F_{++}\bar{F}_{+-}}{F_{-+}\bar{F}_{--}},$$

(2.21)

$$\frac{D_{+}^{(t)}}{D_{+}^{(s)}} = \frac{F_{++}}{\bar{F}_{--}}\det\bar{F}.$$

(2.22)

Next, we will translate this into equations for two- and 3-point structure constants.

**Shift equations**

In Eq. (2.21), let us write the coefficients $D_{\pm}^{(s)}$ in terms of structure constants, and the fusing matrix elements as functions of momenta using Eq. (2.16):

$$\frac{C_{(r_1,s_1)}^{-}C_{(r_1,s_1-1)(r_2,s_2)(r_3,s_3)}}{C_{(r_1,s_1)}^{+}C_{(r_1,s_1+1)(r_2,s_2)(r_3,s_3)}} = -\frac{\Gamma(1 + 2\beta^{-1}P_1)}{\Gamma(1 - 2\beta^{-1}P_1)}\frac{\Gamma(1 + 2\beta^{-1}\bar{P}_1)}{\Gamma(1 - 2\beta^{-1}\bar{P}_1)}\frac{(-)^{2r_2}}{\pi^4}$$

$$\times \prod_{\epsilon_2,\epsilon_3=\pm}\cos\left(\pi\beta^{-1}\left(P_1 + \epsilon_2 P_2 + \epsilon_3 P_3\right)\right)$$

$$\times \Gamma\left(\frac{1}{2} - \beta^{-1}(P_1 + \epsilon_2 P_2 + \epsilon_3 P_3)\right)\Gamma\left(\frac{1}{2} - \beta^{-1}(\bar{P}_1 + \epsilon_2\bar{P}_2 + \epsilon_3\bar{P}_3)\right).$$

(2.23)

This expression is invariant under $(P_i) \leftrightarrow (\bar{P}_i)$, thanks to the relation $\beta^{-1}\bar{P}_i = r_i + \beta^{-1}P_i$, together with Eq. (2.20).

Let us also write the same equation in the case of a 4-point function of the type $\left\langle V^d_{(1,2)} V_{(r,s)} V^d_{(1,2)} V_{(r,s)} \right\rangle$, which involves two degenerate fields. This special case amounts to setting $(r_1, s_1) = (r_3, s_3) = (r, s)$, and $(r_2, s_2) = (0, 2\beta P_{(1,2)})$. The compatibility condition (2.20) then boils down to $r \in \frac{1}{2}\mathbb{Z}$. The 3-point structure constants can be rewritten in terms of degenerate OPE coefficients and 2-point structure constants, and we find

$$\frac{(C^-_{(r,s)})^2 B_{(r,s-1)}}{(C^+_{(r,s)})^2 B_{(r,s+1)}} = -\frac{\Gamma\left(1 + 2\beta^{-1}P\right)}{\Gamma\left(1 - 2\beta^{-1}P\right)} \frac{\Gamma\left(1 + 2\beta^{-1}\bar{P}\right)}{\Gamma\left(1 - 2\beta^{-1}\bar{P}\right)} \frac{\Gamma\left(\beta^{-2} - 2\beta^{-1}P\right)\Gamma\left(1 - \beta^{-2} - 2\beta^{-1}P\right)}{\Gamma\left(\beta^{-2} + 2\beta^{-1}\bar{P}\right)\Gamma\left(1 - \beta^{-2} + 2\beta^{-1}\bar{P}\right)}. \quad (2.24)$$

Finally, let us rewrite Eq. (2.22) in the same manner:

$$\frac{C^+_{(r_3,s_3)} C_{(r_1,s_1)(r_2,s_2)(r_3,s_3+1)}}{C^+_{(r_1,s_1)} C_{(r_1,s_1+1)(r_2,s_2)(r_3,s_3)}} = (-)^{r_3} \frac{\Gamma\left(1 + 2\beta^{-1}P_1\right)}{\Gamma\left(-2\beta^{-1}\bar{P}_1\right)} \frac{\Gamma\left(-2\beta^{-1}P_3\right)}{\Gamma\left(1 + 2\beta^{-1}\bar{P}_3\right)} \prod_{\pm} \frac{\Gamma\left(\frac{1}{2} - \beta^{-1}\bar{P}_1 \pm \beta^{-1}\bar{P}_2 + \beta^{-1}\bar{P}_3\right)}{\Gamma\left(\frac{1}{2} + \beta^{-1}P_1 \pm \beta^{-1}P_2 - \beta^{-1}P_3\right)}. \quad (2.25)$$

This equation only makes sense provided $r_i \in \mathbb{N}$. This is because the 3-point structure constants involve fields $V_{(r_1,s_1)}$ and $V_{(r_1,s_1+1)}$, which can both have integer spins only provided $r_1 \in \mathbb{Z}$. For the same reason we have $r_3 \in \mathbb{Z}$, and therefore $r_2 \in \mathbb{Z}$ by Eq. (2.20). The sign factor $(-)^{r_3}$ appears when applying Eq. (2.10) to the permutation that relates the degenerate OPE $V_{(r_3,s_3)} V^d_{(1,2)}$ to $V^d_{(1,2)} V_{(r_3,s_3)}$ (2.11).

In contrast, Eq. (2.23) involved $V_{(r_1,s_1-1)}$ and $V_{(r_1,s_1+1)}$, whose spins differ by $2r_1$, which allows $r_1 \in \frac{1}{2}\mathbb{N}$. Degenerate OPE coefficients involved fields whose spins differ by $r_1$, but these were auxiliary quantities, so we did not impose $r_1 \in \mathbb{N}$ in that case.

**Behaviour of reference structure constants under shifts**

Let us show that reference structure constants provide solutions of shift equations, up to signs. To do this, let us work out how the normalized 3-point structure constant $C^{\mathrm{norm}}$ (1.9) behaves under shifts. We start with the shift equation for the double Gamma function (1.5), and deduce the identity

$$\prod_{\pm} \frac{\Gamma_\beta\left(\frac{\beta + \beta^{-1}}{2} + \frac{\beta}{2}R \pm \frac{\beta^{-1}}{2}(S+1)\right)}{\Gamma_\beta\left(\frac{\beta + \beta^{-1}}{2} + \frac{\beta}{2}R \pm \frac{\beta^{-1}}{2}(S-1)\right)} = \frac{\beta^{-\beta^{-2}S}}{\pi} \cos\frac{\pi}{2}\left(R + \beta^{-2}S\right) \prod_{\pm} \Gamma\left(\frac{1}{2} \pm \frac{1}{2}R - \frac{\beta^{-2}}{2}S\right). \quad (2.26)$$

The values of $R$ that appear in $C^{\mathrm{ref}}$ (1.7) are integers of the type $|r_1 \pm r_2 \pm r_3|$, and we find

$$\frac{C^{\mathrm{ref}}_{(r_1,s_1-1)(r_2,s_2)(r_3,s_3)}}{C^{\mathrm{ref}}_{(r_1,s_1+1)(r_2,s_2)(r_3,s_3)}} = \frac{\beta^{-4\beta^{-2}s_1}}{\pi^4} \prod_{\pm,\pm} \cos\frac{\pi}{2}\left(|r_1 \pm r_2 \pm r_3| + \beta^{-2}(s_1 \pm s_2 \pm s_3)\right)$$
$$\times \Gamma\left(\frac{1}{2} - \beta^{-1}(P_1 \pm P_2 \pm P_3)\right)\Gamma\left(\frac{1}{2} - \beta^{-1}(\bar{P}_1 \pm \bar{P}_2 \pm \bar{P}_3)\right). \quad (2.27)$$

Therefore, $C^{\mathrm{norm}}$ satisfies the shift equation (2.23) provided the degenerate OPE coefficients obey

$$\frac{C^-_{(r,s)}}{C^+_{(r,s)}} = -\beta^{4\beta^{-2}s} \frac{\Gamma(1 + 2\beta^{-1}P)}{\Gamma(1 - 2\beta^{-1}P)} \frac{\Gamma(1 + 2\beta^{-1}\bar{P})}{\Gamma(1 - 2\beta^{-1}\bar{P})}, \quad (2.28)$$

and provided the normalized 3-point structure constant obeys

$$\frac{C^{\mathrm{norm}}_{(r_1,s_1+2)(r_2,s_2)(r_3,s_3)}}{C^{\mathrm{norm}}_{(r_1,s_1)(r_2,s_2)(r_3,s_3)}} = (-)^{2r_3}(-)^{\max(2r_1, 2r_2, 2r_3, r_1 + r_2 + r_3)}, \quad (2.29)$$

where we used the identity

$$\sum_{\pm,\pm} |r_1 \pm r_2 \pm r_3| = 2\max(2r_1, 2r_2, 2r_3, r_1 + r_2 + r_3). \tag{2.30}$$

This result is consistent with permutation symmetry of structure constants (2.10): under the odd permutation $2 \leftrightarrow 3$, the ratio $\frac{C^{\mathrm{norm}}|_{s_1 \to s_1+2}}{C^{\mathrm{norm}}}$ should pick a factor $\frac{(-)^{S_1}|_{s_1 \to s_1+2}}{(-)^{S_1}} = (-)^{2r_1}$. Eq. (2.29) is consistent with this expectation, thanks to the conservation of $r$ modulo integers.

Combining the square of Eq. (2.28) with Eq. (2.24), we obtain the ratio $\frac{B_{(r,s-1)}}{B_{(r,s+1)}}$, which turns out to coincide with the corresponding ratio of reference 2-point functions (1.6),

$$\frac{B^{\mathrm{ref}}_{(r,s-1)}}{B^{\mathrm{ref}}_{(r,s+1)}} = (-)^{2r} \beta^{-8\beta^{-2}s} \frac{\Gamma(1 - 2\beta^{-1}P)}{\Gamma(1 + 2\beta^{-1}P)} \frac{\Gamma(1 - 2\beta^{-1}\bar{P})}{\Gamma(1 + 2\beta^{-1}\bar{P})} \frac{\Gamma(1 - \beta^{-2} - 2\beta^{-1}P)}{\Gamma(1 - \beta^{-2} + 2\beta^{-1}P)} \frac{\Gamma(\beta^{-2} - 2\beta^{-1}\bar{P})}{\Gamma(\beta^{-2} + 2\beta^{-1}\bar{P})}. \tag{2.31}$$

Finally, let us assume $r_i \in \mathbb{N}$, and use Eq. (2.26) for computing

$$\frac{C^{\mathrm{ref}}_{(r_1,s_1)(r_2,s_2)(r_3,s_3+1)}}{C^{\mathrm{ref}}_{(r_1,s_1+1)(r_2,s_2)(r_3,s_3)}} = \frac{\beta^{2\beta^{-2}(s_3-s_1)}}{\pi^2} \prod_{\pm} \cos \frac{\pi}{2} \left( |r_1 \pm r_2 - r_3| + \beta^{-2}(s_1 \pm s_2 - s_3) \right)$$
$$\times \Gamma\left(\tfrac{1}{2} - \beta^{-1}(P_1 \pm P_2 - P_3)\right) \Gamma\left(\tfrac{1}{2} - \beta^{-1}(\bar{P}_1 \pm \bar{P}_2 - \bar{P}_3)\right). \tag{2.32}$$

This agrees with the shift equation (2.25), provided the degenerate OPE coefficient is

$$C^+_{(r,s)} = \beta^{-2\beta^{-2}s} \frac{\Gamma(-2\beta^{-1}P)}{\Gamma(1 + 2\beta^{-1}\bar{P})}, \tag{2.33}$$

and provided the normalized 3-point structure constant obeys

$$\frac{C^{\mathrm{norm}}|_{s_1 \to s_1+1}}{C^{\mathrm{norm}}|_{s_3 \to s_3+1}} = \begin{cases} (-)^{r_1+r_2} & \text{if } r_2 \geq |r_1 - r_3|, \\ (-)^{r_3} & \text{else}. \end{cases} \tag{2.34}$$

This result is consistent with permutation symmetry of structure constants (2.10): under the odd permutation $1 \leftrightarrow 3$, the ratio $\frac{C^{\mathrm{norm}}|_{s_1 \to s_1+1}}{C^{\mathrm{norm}}|_{s_3 \to s_3+1}}$ should pick a factor $\frac{(-)^{S_1+S_3}|_{s_1 \to s_1+1}}{(-)^{S_1+S_3}|_{s_3 \to s_3+1}} = (-)^{r_1+r_3}$, and it does. Moreover, using cyclic permutations, we deduce two more shift equations,

$$\frac{C^{\mathrm{norm}}|_{s_1 \to s_1+1}}{C^{\mathrm{norm}}|_{s_2 \to s_2+1}} = \begin{cases} (-)^{r_2+r_3} & \text{if } r_3 \geq |r_1 - r_2|, \\ (-)^{r_1} & \text{else}, \end{cases} \tag{2.35}$$

$$\frac{C^{\mathrm{norm}}|_{s_2 \to s_2+1}}{C^{\mathrm{norm}}|_{s_3 \to s_3+1}} = \begin{cases} (-)^{r_1+r_3} & \text{if } r_1 \geq |r_2 - r_3|, \\ (-)^{r_2} & \text{else}. \end{cases} \tag{2.36}$$

Our three shift equations are compatible, i.e. the product of the three shifts is one. To see this, we have to consider two cases. The first case is when the three conditions $r_i \geq |r_j - r_k|$ are obeyed, i.e. when there exists a planar triangle with sides of lengths $r_1, r_2, r_3$. Then compatibility boils down to $(-)^{r_1+r_2}(-)^{r_2+r_3}(-)^{r_1+r_3} = 1$. The second case is when say $r_2 > r_1 + r_3$. Then compatibility reduces to $(-)^{r_1+r_2}(-)^{r_1}(-)^{r_2} = 1$.

**Why there is no reference sign factor**

Under shifts, the normalized 3-point structure constant $C^{\mathrm{norm}}$ (1.9) only picks signs. It is natural to wonder whether we could include a reference sign factor in $C^{\mathrm{ref}}$, such that $C^{\mathrm{norm}}$

would be invariant under shifts, and under permutations as well. We will now argue that this is not possible, due to two different obstacles.

The first obstacle is that a reference sign factor would have to be a universal quantity, and depend only on the fields' conformal dimensions. However, the permutation properties of fields do not depend solely on their dimensions. For example, given two fields $V, W$, the permutation relation (2.10), which was based on the assumption that fields have integer spins, implies that $\langle V V W \rangle$ can be nonzero only provided the spin of $W$ is even. On the other hand, given two fields $V^{(1)}, V^{(2)}$ with the same conformal dimensions, nothing prevents $\left\langle V^{(1)} V^{(2)} W \right\rangle$ from being nonzero if the spin of $W$ is odd.

In this case, and in its images under shifts, we could try to overcome the first obstacle by building a reference sign factor $\theta$ that obeys $\theta_{\sigma(1)\sigma(2)\sigma(3)} = -\text{sign}(\sigma)^{S_1+S_2+S_3} \theta_{123}$ instead of Eq. (2.10). After this sign flip, the permutation relation remains compatible with shift equations. After including the resulting reference sign factor in $C^{\text{ref}}$, the normalized structure constant $C^{\text{norm}}$ would be invariant under shifts, but it would have to pick a minus sign under odd permutations.

The second obstacle is that we have diagonal fields, whose momenta $P_1$ may take any complex values. A sign that depends continuously on a complex number must be constant. This would not be a problem if we only worried about the shift $P_1 \to P_1 + \beta^{-1}$ i.e. $s_1 \to s_1 + 2$: in the case $r_1 = 0$ of a diagonal field, the corresponding sign is, according to Eq. (2.29),

$$\frac{C^{\text{norm}}|_{s_1 \to s_1 + 2}}{C^{\text{norm}}} \underset{r_1 = 0}{=} 1. \tag{2.37}$$

However, if $r_2, r_3 \in \mathbb{N}$, we also have shift equations for $s_1 \to s_1 + 1$, and the corresponding sign (2.34) can be $-1$.

Let us rephrase these statements in terms of the loop weight $w_1 = w(P_1)$ (1.14). We assume that the diagonal field $V_{P_1}$ only depends on its conformal dimension, and is therefore invariant under $P_1 \to -P_1$. Together with Eq. (2.37), this implies that $C^{\text{norm}}$ is a function of $w_1$, and shift equations for $s_1 \to s_1 + 1$ imply that it can have a nontrivial behaviour under $w_1 \to -w_1$. Since $w_1 \in \mathbb{C}$, this behaviour cannot be captured by a sign factor.

## 2.3 Analyticity in dimensions of diagonal fields

From the S-matrix bootstrap to Seiberg–Witten theory or integrable models, whenever we have continuous parameters, it is crucial to know the analytic properties of physical observables as functions of these parameters. This is also the case in two-dimensional CFT. For example, in the bootstrap solution of Liouville theory, the analytic properties of correlators as functions of the central charge and conformal dimensions play an essential role [3, 4].

In critical loop models, we have two types of continuous parameters: the central charge, and the momenta of diagonal fields. In the lattice approach, correlation functions depend on these parameters via loop weights, respectively called $n$ (1.12) and $w$ (1.14). For any finite lattice size, unnormalized correlation functions are polynomials in loop weights [29]. In the critical limit, the dependence becomes more complicated, and structure constants are written in terms of Barnes' double Gamma function.

We will now focus on how 4-point functions depend on the momentum $P$ of an $s$-channel diagonal field. Such a field appears in 4-point functions of diagonal fields [25], and in other 4-point functions as well, for example in the solution $Z_5$ (1.15). In the decomposition of such 4-point functions into conformal blocks, some terms have first-order poles in $P$. Assuming that these poles cancel between various terms will lead to non-trivial constraints on structure constants.

**Poles and their cancellation**

Consider a 4-point function $Z(P) = \left\langle \prod_{i=1}^{4} V_{(r_i, s_i)} \right\rangle$, such that the $s$-channel spectrum includes a diagonal field $V_P$. Due to the existence of the degenerate field $V_{\langle 1,3 \rangle}^{d}$, we must in fact have an infinite family of diagonal fields with momenta in $P + \beta^{-1}\mathbb{Z}$, see the OPE (2.4). The $s$-channel decomposition of $Z(P)$ may be written as

$$Z(P) = \sum_{k \in \mathbb{Z}} D_{P + k\beta^{-1}} \mathcal{G}_{P + k\beta^{-1}} + \sum_{r \in \mathbb{N}^*} \sum_{s \in \frac{1}{r}\mathbb{Z}} D_{(r,s)}(P) \mathcal{G}_{(r,s)}. \tag{2.38}$$

Since our diagonal field $V_P$ has $r = 0$, the rule (2.20) forces the first Kac index $r$ to be integer for all fields in the $s$-channel. We use the notations $\mathcal{G}_P$ and $\mathcal{G}_{(r,s)}$ for $s$-channel non-chiral conformal blocks: while $\mathcal{G}_P = |\mathcal{F}_P|^2$ is holomorphically factorized, $\mathcal{G}_{(r,s)}$ is factorized only for $s \notin \mathbb{N}^*$, while for $s \in \mathbb{N}^*$ it is logarithmic, and by convention $\mathcal{G}_{(r,s)} \underset{s \in -\mathbb{N}^*}{=} 0$ [23].

We first assume that the diagonal 4-point structure constant coincides with the reference 4-point structure constant,

$$D_P = D_P^{\text{ref}} = \frac{C_{(r_1,s_1)(r_2,s_2)P}^{\text{ref}} C_{P(r_3,s_3)(r_4,s_4)}^{\text{ref}}}{B_P}, \tag{2.39}$$

with $B_P^{\text{ref}} = B_P$. In the case of 4-point functions of diagonal fields $\left\langle \prod_{i=1}^{4} V_{P_i} \right\rangle$, there are diagonal fields in all channels $s, t, u$. Ratios such as $\frac{D_{P_s}^{(s)}}{D_{P_t}^{(t)}}$ are determined by crossing symmetry, and their numerical agreement with the corresponding reference quantities provides a very non-trivial test of our assumption (2.39) [25]. We now make the same assumption for our more general 4-point function $Z(P)$. If there is a diagonal field in the $s$-channel only, this does not lead to predictions that can be compared with numerical bootstrap results: rather, the assumption amounts to a choice of overall normalization for a solution of crossing symmetry. The suitability of this choice will depend on the simplicity of the resulting expressions for the other structure constants.

Consider the analytic behaviour of the diagonal terms of $Z(P)$ as functions of $P$. Since Barnes' double Gamma function $\Gamma_\beta(x)$ has simple poles for $x \in -\beta\mathbb{N} - \beta^{-1}\mathbb{N}$ and no zeros, $C_{(r_1,s_1)(r_2,s_2)P}^{\text{ref}}$ (1.7) has no poles, and $D_P^{\text{ref}}$ has poles that correspond to the zeros of $B_P^{\text{ref}}$ (1.4). For any $r, s \in \mathbb{N}^*$, there is a double pole at $P = P_{(r,-s)}$, and simple poles at $P = P_{(0,s)}$ and $P = P_{(r,0)}$. Moreover, for $r, s \in \mathbb{N}^*$, the chiral conformal block $\mathcal{F}_P$ has a simple pole, with the residue

$$\operatorname*{Res}_{P = P_{(r,s)}} \mathcal{F}_P = \frac{R_{r,s}}{2P_{(r,s)}} \mathcal{F}_{P_{(r,-s)}}, \tag{2.40}$$

where the quantity $R_{r,s}$ will be given explicitly in Eq. (2.45). Therefore, $\mathcal{G}_P = |\mathcal{F}_P|^2$ has a double pole. According to [23], double poles cancel in the combination $D_P \mathcal{G}_P + D_{P - s\beta^{-1}} \mathcal{G}_{P - s\beta^{-1}}$, and there remains a simple pole, whose residue is proportional to the logarithmic non-chiral block $\mathcal{G}_{(r,s)}$:

$$\operatorname*{Res}_{P = P_{(r,s)}} \left( D_P \mathcal{G}_P + D_{P - s\beta^{-1}} \mathcal{G}_{P - s\beta^{-1}} \right) = \frac{\bar{R}_{r,s}}{2P_{(r,s)}} D_{P_{(r,s)}} \mathcal{G}_{(r,s)}. \tag{2.41}$$

Therefore, the diagonal terms of $Z(P)$ have simple poles, whose residues are proportional to conformal blocks from the non-diagonal sector. We now assume that the non-diagonal

structure constants $D_{(r,s)}(P)$ have simple poles that cancel the poles from the diagonal terms, so that

$$\forall r,s \in \mathbb{Z}, \qquad \operatorname*{Res}_{P=P_{(r,s)}} Z(P) = 0. \tag{2.42}$$

Let us sketch the implications of this assumption. We focus on three cases:

1. Since $D_P \mathcal{G}_P$ is an even function of $P$ and $P_{(0,-s)} = -P_{(0,s)}$, the simple poles at $P = P_{(0,s)}$ cancel between the diagonal terms of $Z(P)$ (2.38). So our assumption is trivially satisfied in this case.

2. The residue $\operatorname{Res}_{P=P_{(r,0)}} D_{(r,0)}(P)\mathcal{G}_{(r,0)}$ must cancel a residue at $P = P_{(r,0)}$ that is itself proportional to $\mathcal{G}_{(r,0)}$. This can only come from the diagonal term $D_P \mathcal{G}_P$, and we must have

$$\operatorname*{Res}_{P=P_{(r,0)}} D_{(r,0)}(P) = -\operatorname*{Res}_{P=P_{(r,0)}} D_P. \tag{2.43}$$

3. For $r,s \in \mathbb{N}^*$, the residue $\operatorname{Res}_{P=P_{(r,s)}} D_{(r,s)}(P)\mathcal{G}_{(r,s)}$ must cancel a residue at $P = P_{(r,s)}$ that is itself proportional to $\mathcal{G}_{(r,s)}$. According to Eq. (2.41), this comes from a combination of two diagonal terms, and we have

$$\operatorname*{Res}_{P=P_{(r,s)}} D_{(r,s)}(P) = -\frac{\bar{R}_{r,s}}{2P_{(r,s)}} D_{P_{(r,s)}}. \tag{2.44}$$

Since the diagonal term of $Z(P)$ is invariant under $P \to P + \beta^{-1}$, the poles of $D_{(r,s)}(P)$ must be invariant too, so that $\operatorname{Res}_{P=P_{(r,s'+2)}} D_{(r,s)}(P) = \operatorname{Res}_{P=P_{(r,s')}} D_{(r,s)}(P)$. This allows us to deduce all the residues of $D_{(r,s)}(P)$ from the three cases that we have just considered.

To summarize, the analyticity assumption (2.42) determines the residues of non-diagonal structure constants $D_{(r,s)}(P)$ in terms of diagonal structure constants, which coincide with reference structure constants according to the assumption (2.39). Poles and residues do not completely determine the structure constant $D_{(r,s)}(P)$, but they provide important hints for guessing the corresponding reference structure constant. We will now compute the residues more explicitly.

**Residues of non-diagonal structure constants**

In order to compute the residue (2.44), let us fist spell out the conformal block residue $R_{r,s}$ (2.40) more explicitly. This is a well-known universal quantity, which plays an important role in Zamolodchikov's recursive representation of Virasoro conformal blocks. Here, we rewrite it in terms of Barnes' double Gamma function, following [31], while using the notation (2.17) for $P, \bar{P}$:

$$\frac{R_{r,s}}{P_{(r,s)}} = \frac{c_{r,s}(P_1, P_2)c_{r,s}(P_4, P_3)}{b_{r,s}}, \qquad \frac{\bar{R}_{r,s}}{P_{(r,s)}} = \frac{c_{r,s}(\bar{P}_1, \bar{P}_2)c_{r,s}(\bar{P}_4, \bar{P}_3)}{b_{r,s}}, \tag{2.45}$$

where we define

$$c_{r,s}(P_1, P_2) = \prod_{\pm,\pm} \frac{\Gamma_\beta\left(\frac{\beta+\beta^{-1}}{2} + P_1 \pm P_2 \pm P\right)}{\Gamma_\beta\left(\frac{\beta+\beta^{-1}}{2} + P_1 \pm P_2 \pm \bar{P}\right)}, \qquad b_{r,s} = \frac{\Gamma_\beta(\beta - 2P)\Gamma_\beta(\beta + 2P)}{\Gamma_\beta(\beta - 2\bar{P})\operatorname{Res}_{\beta+2\bar{P}}\Gamma_\beta}, \tag{2.46}$$

which obey the properties $c_{r,s}(P_2, P_1) = (-)^{rs} c_{r,s}(P_1, P_2)$ and $c_{r,s}(-P_1, P_2) = c_{r,s}(P_1, P_2)$. Let us compute the ratio

$$\rho^{r,s}_{(r_1,s_1)(r_2,s_2)} = c_{r,s}(\bar{P}_1, \bar{P}_2) \frac{C^{\text{ref}}_{P(r_1,s_1)(r_2,s_2)}}{C^{\text{ref}}_{(r,s)(r_1,s_1)(r_2,s_2)}} . \tag{2.47}$$

Assuming for convenience $r_1 \geq r_2$, we find

$$c_{r,s}(\bar{P}_1, \bar{P}_2) C^{\text{ref}}_{P(r_1,s_1)(r_2,s_2)} \underset{r_1 \geq r_2}{=} \prod_{\pm,\pm} \Gamma_\beta^{-1} \left( \tfrac{\beta+\beta^{-1}}{2} - P_1 \pm P_2 \pm P \right) \Gamma_\beta^{-1} \left( \tfrac{\beta+\beta^{-1}}{2} + \bar{P}_1 \pm \bar{P}_2 \pm \bar{P} \right) . \tag{2.48}$$

In the case $r \leq |r_1 - r_2|$ together with $r_1 \geq r_2$, we find that this coincides with $C^{\text{ref}}_{(r,s)(r_1,s_1)(r_2,s_2)}$. We can easily get $r_1 < r_2$ by the permutation $1 \leftrightarrow 2$, therefore

$$\rho^{r,s}_{(r_1,s_1)(r_2,s_2)} \underset{r \leq |r_1-r_2|}{=} (-)^{rs\delta_{r_1<r_2}} . \tag{2.49}$$

In the case $r > |r_1 - r_2|$, we find

$$\rho^{r,s}_{(r_1,s_1)(r_2,s_2)} = \frac{S_\beta \left( \tfrac{\beta+\beta^{-1}}{2} - \bar{P}_1 + \bar{P}_2 + \bar{P} \right)}{S_\beta \left( \tfrac{\beta+\beta^{-1}}{2} - P_1 + P_2 + P \right)} \left[ \frac{S_\beta \left( \tfrac{\beta+\beta^{-1}}{2} - \bar{P}_1 - \bar{P}_2 + \bar{P} \right)}{S_\beta \left( \tfrac{\beta+\beta^{-1}}{2} - P_1 - P_2 + P \right)} \right]^{\delta_{r>r_1+r_2}} , \tag{2.50}$$

where we introduce the double sine function and its shift equation, deduced from Eq. (1.5):

$$S_\beta(x) = \frac{\Gamma_\beta(x)}{\Gamma_\beta(\beta + \beta^{-1} - x)} \implies \frac{S_\beta(x + \beta)}{S_\beta(x)} = 2\sin(\pi\beta x). \tag{2.51}$$

Together with the relation $\bar{P}_i = \beta r_i + P_i$, this shift equation implies that the ratio $\rho^{r,s}_{(r_1,s_1)(r_2,s_2)}$ is trigonometric:

$$\rho^{r,s}_{(r_1,s_1)(r_2,s_2)} = (-)^{s\min(r,|r_1-r_2|)\delta_{r_1<r_2}} \prod_{\pm} \prod_{j=-\frac{r-1-|r_1\pm r_2|}{2}}^{\frac{r-1-|r_1\pm r_2|}{2}} 2\cos\pi\left( j\beta^2 + \tfrac{s-s_1\mp s_2}{2} \right), \tag{2.52}$$

where the product over $j$ runs by increments of 1. This formula is in fact valid irrespective of the signs of $r_1 - r_2$ and $r - |r_1 - r_2|$: the sign prefactor ensures that it picks a factor $(-)^{rs}$ under the permutation $1 \leftrightarrow 2$ as it should, and in the first product, the $+$ factor is one if $r \leq r_1 + r_2$, while the $-$ factor is one if $r \leq |r_1 - r_2|$.

For $r, s \in \mathbb{N}^*$, we compute the reference 2-point structure constant by taking a limit from the generic case $s \notin \mathbb{Z}$ (1.6),

$$B^{\text{ref}}_{(r,s)} = \frac{(-)^{(r+1)(s+1)}}{2\pi\beta \sin(\pi\beta^{-2}s)} \Gamma_\beta^{-1}\left( \beta - 2\bar{P} \right) \left[ \text{Res}_{\beta+2\bar{P}} \Gamma_\beta \right]^{-1} \prod_{\pm} \Gamma_\beta^{-1}\left( \beta \pm 2P \right) . \tag{2.53}$$

Moreover, the diagonal 2-point structure constant (1.4) at any momentum $P$ may be rewritten as

$$B_P = \frac{\sin(2\pi\beta P)}{\sin(2\pi\beta^{-1}P)} \prod_{\pm} \Gamma_\beta^{-2}\left( \beta \pm 2P \right) . \tag{2.54}$$

When combining these 2-point structure constant with the quantity $b_{r,s}$ (2.46), we find

$$b_{r,s} \frac{B_{P_{(r,s)}}}{B^{\text{ref}}_{(r,s)}} = \frac{1}{2}(-)^{(r+1)s} w'\left( P_{(r,s)} \right) , \tag{2.55}$$

where $w'(P)$ is the derivative of the loop weight $w(P)$ (1.14). Now, let us use this result, together with the corresponding combination of 3-point structure constants (2.47), in order to evaluate the residues of 4-point structure constants (2.44):

$$\frac{\underset{w=w(P_{(r,s)})}{\mathrm{Res}} D_{(r,s)}(w)}{D_{(r,s)}^{\mathrm{ref}}} \underset{r\in\mathbb{N}^*, s\in\mathbb{N}}{=} -(-)^{(r+1)s} \rho^{r,s}_{(r_1,s_1)(r_2,s_2)} \rho^{r,s}_{(r_4,s_4)(r_3,s_3)}. \tag{2.56}$$

While we have been assuming $s \in \mathbb{N}^*$, this result is in fact also valid for $s = 0$, as we now quickly sketch. In this case $P = \bar{P}$, so that formally $c_{r,s}(P_1, P_2) = 1$ in Eq. (2.46), and our results for $\rho^{r,s}_{(r_1,s_1)(r_2,s_2)}$ are still valid. The reference 2-point function becomes

$$B^{\mathrm{ref}}_{(r,0)} = \frac{(-)^r}{2\pi^2} \Gamma_\beta^{-2} (\beta - 2P) \left[\mathrm{Res}_{\beta+2P} \Gamma_\beta\right]^{-2},$$

and we have

$$\mathrm{Res}_{P=P_{(r,0)}} B_P^{-1} = \frac{\pi}{2\beta} \frac{(-)^r}{\sin(2\pi\beta P)} \Gamma_\beta^2 (\beta - 2P) \left[\mathrm{Res}_{\beta+2P} \Gamma_\beta\right]^2.$$

Having obtained the residues of $D_{(r,s)}$, let us dispel a cloud of asymmetry that hangs over the whole calculation. The logarithmic conformal block $\mathcal{G}_{(r,s)}$ is normalized with respect to the primary field $V_{(r,s)}$, which is left-degenerate for $r, s \in \mathbb{N}^*$. The same logarithmic module also contains the right-degenerate primary field $V_{(-r,s)}$ [23]. Normalizing with respect to $V_{(-r,s)}$, we would have to do the replacements $\bar{R}_{r,s} \to R_{r,s}$ and $D^{\mathrm{ref}}_{(r,s)} \to D^{\mathrm{ref}}_{(-r,s)}$ in our calculation. We would nevertheless get the same residue (2.56), thanks to the identity $\frac{\bar{R}_{r,s}}{D^{\mathrm{ref}}_{(r,s)}} = \frac{R_{r,s}}{D^{\mathrm{ref}}_{(-r,s)}}$, which follows from $\frac{c_{r,s}(\bar{P}_1, \bar{P}_2)}{C^{\mathrm{ref}}_{(r,s)(r_1,s_1)(r_2,s_2)}} = \frac{c_{r,s}(P_1, P_2)}{C^{\mathrm{ref}}_{(-r,s)(r_1,s_1)(r_2,s_2)}}$, which can itself be deduced from the behaviour of Eq. (2.26) under $R \to -R$.

**Polynomial factors**

Remarkably, the normalized residues (2.56) of $D_{(r,s)}(w)$ are polynomial functions of loop weights. The quantity $\rho^{r,s}_{(r_1,s_1)(r_2,s_2)}$ (2.52) is indeed a polynomial function of the weight $n$ (1.12) of contractible loops. Moreover, if $V_{(r_1,s_1)}$ and/or $V_{(r_2,s_2)}$ are diagonal (i.e. $r_1 = 0$ and/or $r_2 = 0$), then $\rho^{r,s}_{(r_1,s_1)(r_2,s_2)}$ is also polynomial in the corresponding loop weights $w_i = w(P_i)$ (1.14), with $s_i = 2\beta P_i$. The degrees of this polynomial are

$$\deg_n \rho^{r,s}_{(r_1,s_1)(r_2,s_2)} = \left\lfloor \frac{1}{2}\left(r^2 + r_1^2 + r_2^2\right) - r\max(r_1, r_2)\right\rfloor, \tag{2.57}$$

$$\deg_{w_1} \rho^{r,s}_{(0,s_1)(r_2,s_2)} = r - r_2. \tag{2.58}$$

(We assume $r > |r_1 - r_2|$, otherwise our polynomial is constant.) Let us illustrate this in a few examples. We start with the case $\rho^{r,s}_{(0,s_1)(0,s_2)}$ where both fields are diagonal:

$$\rho^{1,0} = w_1 + w_2, \tag{2.59a}$$

$$\rho^{1,1} = -w_1 + w_2, \tag{2.59b}$$

$$\rho^{2,0} = w_1^2 + w_2^2 + n^2 - nw_1w_2 - 4, \tag{2.59c}$$

$$\rho^{2,1} = w_1^2 + w_2^2 + n^2 + nw_1w_2 - 4, \tag{2.59d}$$

$$\rho^{3,0} = (w_1 + w_2)\left(w_1^2 + w_2^2 + (n^2 - 2)^2 + w_1w_2(n^2 - 2) - 4\right), \tag{2.59e}$$

$$\rho^{3,1} = (-w_1 + w_2)\left(w_1^2 + w_2^2 + (n^2 - 2)^2 - w_1w_2(n^2 - 2) - 4\right). \tag{2.59f}$$

Next, we consider an example with two non-diagonal fields $\rho^{r,s}_{(\frac{3}{2},\frac{2}{3})(\frac{1}{2},0)}$:

$$\rho^{1,0} = 1\,, \tag{2.60a}$$

$$\rho^{1,1} = 1\,, \tag{2.60b}$$

$$\rho^{2,0} = 1\,, \tag{2.60c}$$

$$\rho^{2,1} = \sqrt{3}\,, \tag{2.60d}$$

$$\rho^{3,0} = -(n+1)\,, \tag{2.60e}$$

$$\rho^{3,1} = -\sqrt{3}(n-1)\,. \tag{2.60f}$$

Now comes the crux of the argument of this section 2.3, and maybe of the whole article. Since the normalized 4-point structure constant has polynomial residues, we conjecture that its holomorphic term, which is obtained by subtracting the simple pole, is also polynomial. In fact, we further conjecture that an $x$-channel normalized 4-point structure constant is still polynomial when it has no pole, i.e. when $s \notin \mathbb{Z}$, or even when there is no diagonal field in the channel, equivalently when the signature $\sigma^{(x)}$ is nonzero.

Let us write these conjectures more explicitly. We call $d^{(x)}_{(r,s)}$ the normalized 4-point structure constant, after subtracting the pole term if there is one, so that

$$\boxed{\frac{D^{(x)}_{(r,s)}}{D^{(x)\text{ref}}_{(r,s)}} = d^{(x)}_{(r,s)} + \delta_{\sigma^{(x)},0}\delta_{s\in\mathbb{Z}}\frac{f^{(x)}_{(r,s)}}{w_x - w(P_{(r,s)})}\,.} \tag{2.61}$$

Here we have restored the dependence on the channel $x \in \{s, t, u\}$, instead of working in the $s$-channel only. The quantity $w_x$ is the loop weight of the $x$-channel diagonal field, if it exists. The residue $f^{(x)}_{(r,s)}$ is known explicitly; in the $s$-channel it is given by Eq. (2.56). The term $d^{(x)}_{(r,s)}$ will be computed using numerical bootstrap methods; exact expressions can then be inferred from numerical results thanks to the following conjecture:

---

**Conjecture**: The normalized structure constants $d^{(x)}_{(r,s)}$, defined as in Eq. (2.61) to exclude the pole term when there is one, depend polynomially on all relevant loop weights: the weight $n$ of contractible loops, which is related to the central charge via Eq. (1.12), the weights of diagonal fields $w_1, w_2, w_3, w_4$, and the weights of channel diagonal fields $w_s, w_t, w_u$ when applicable.

---

Let us sketch how our conjectured polynomials behave under shifts, according to the results of Section 2.2:

- Whenever we have a diagonal field, normalized structure constants are invariant under the corresponding shift $P \to P + \beta^{-1}$, according to Eq. (2.37). Together with the invariance under $P \to -P$, this implies that $d^{(x)}_{(r,s)}$ is indeed a function of loop weights, and not a more general function of momenta.

- According to Eq. (2.29), we have $d^{(x)}_{(r,s+2)} = \pm d^{(x)}_{(r,s)}$, with a sign that depends on $r, r_i$. This allows us to focus on second indices in an interval of length 2, say $-1 < s \leq 1$. We apply a similar restriction to the indices $s_i$.

- The shift equation (2.34) for normalized 3-point structure constants gives rise to sign flips $w \to -w$ of the corresponding loop weights. This shift equation not only determines how $d^{(x)}_{(r,s)}$ picks sign factors under combinations of shifts of the type $s_i \to s_i + 1$, but it also relates $d^{(x)}_{(r,s)}$ with $d^{(x)}_{(r,s+1)}$.

We will write shift equations for $d_{(r,s)}^{(x)}$ more explicitly in examples in Section 3.

## 2.4 Permutation symmetry of 4-point functions

It is a fundamental axiom of CFT that bosonic fields commute, and a correlation function $\left\langle \prod_{i=1}^{N} V_i(z_i) \right\rangle$ does not depend on the order in which we write the fields. We will now deduce the behaviour of conformal blocks and structure constants under field permutations. We include this elementary technical subject not just to provide comic relief after the previous subsection, but also because this will allow us to write the numerical results of Section 3 more compactly.

**Behaviour of conformal blocks**

We parametrize primary fields by their left and right conformal dimensions $\Delta, \bar{\Delta}$, whose difference is the conformal spin $S = \bar{\Delta} - \Delta$. Let $x = \frac{z_{12}z_{34}}{z_{13}z_{24}}$ be the cross-ratio, then a 4-point function reads

$$\left\langle \prod_{i=1}^{4} V_i(z_i) \right\rangle = \left| z_{13}^{-2\Delta_1} z_{23}^{\Delta_1-\Delta_2-\Delta_3+\Delta_4} z_{34}^{\Delta_1+\Delta_2-\Delta_3-\Delta_4} z_{24}^{-\Delta_1-\Delta_2+\Delta_3-\Delta_4} \right|^2 \mathcal{G}(x) , \qquad (2.62)$$

where we use the notation $\left| z^{\Delta} \right|^2 = z^{\Delta} \bar{z}^{\bar{\Delta}}$, and the function of the cross-ratio may be written as

$$\mathcal{G}(x) = \left\langle V_1(x)V_2(0)V_3(\infty)V_4(1) \right\rangle . \qquad (2.63)$$

Given any permutation of the four fields, we can deduce how $\mathcal{G}(x)$ behaves from the invariance of $\left\langle \prod_{i=1}^{4} V_i(z_i) \right\rangle$. For example, in the case of the transposition $1 \leftrightarrow 2$, which we also call 2134, we have

$$\mathcal{G}(x) = \left| (1-x)^{-\Delta_1-\Delta_2+\Delta_3-\Delta_4} \right|^2 \mathcal{G}\left( 2134 \middle| \tfrac{x}{x-1} \right) . \qquad (2.64)$$

The behaviour of conformal blocks is more complicated, because a permutation can relate different channels, and because blocks can pick signs. Non-chiral conformal blocks are characterized by their asymptotic behaviour:

$$\mathcal{G}_{\Delta,\bar{\Delta}}^{(s)}(x) \underset{x\to 0}{=} \left| x^{\Delta-\Delta_1-\Delta_2} \right|^2 (1 + O(x)) , \qquad (2.65\text{a})$$

$$\mathcal{G}_{\Delta,\bar{\Delta}}^{(t)}(x) \underset{x\to 1}{=} \left| (1-x)^{\Delta-\Delta_1-\Delta_4} \right|^2 (1 + O(1-x)) , \qquad (2.65\text{b})$$

$$\mathcal{G}_{\Delta,\bar{\Delta}}^{(u)}(x) \underset{x\to\infty}{=} \left| \left( \tfrac{1}{x} \right)^{\Delta+\Delta_1-\Delta_3} \right|^2 \left( 1 + O\left( \tfrac{1}{x} \right) \right) . \qquad (2.65\text{c})$$

Tracking the asymptotic behaviour of blocks before and after the permutation, we obtain

$$\mathcal{G}_{\Delta,\bar{\Delta}}^{(s)}(x) = (-)^{S+S_1+S_2} \left| (1-x)^{-\Delta_1-\Delta_2+\Delta_3-\Delta_4} \right|^2 \mathcal{G}_{\Delta,\bar{\Delta}}^{(s)}\left( 2134 \middle| \tfrac{x}{x-1} \right) , \qquad (2.66\text{a})$$

$$\mathcal{G}_{\Delta,\bar{\Delta}}^{(t)}(x) = (-)^{S+S_2+S_3} \left| (1-x)^{-\Delta_1-\Delta_2+\Delta_3-\Delta_4} \right|^2 \mathcal{G}_{\Delta,\bar{\Delta}}^{(u)}\left( 2134 \middle| \tfrac{x}{x-1} \right) , \qquad (2.66\text{b})$$

$$\mathcal{G}_{\Delta,\bar{\Delta}}^{(u)}(x) = (-)^{S+S_1+S_3} \left| (1-x)^{-\Delta_1-\Delta_2+\Delta_3-\Delta_4} \right|^2 \mathcal{G}_{\Delta,\bar{\Delta}}^{(t)}\left( 2134 \middle| \tfrac{x}{x-1} \right) . \qquad (2.66\text{c})$$

Let us write the sign prefactors in a matrix of size 3, whose rows respectively correspond to the blocks $\mathcal{G}_{\Delta,\bar{\Delta}}^{(s)}(x), \mathcal{G}_{\Delta,\bar{\Delta}}^{(t)}(x), \mathcal{G}_{\Delta,\bar{\Delta}}^{(u)}(x)$, and whose colums correspond to the permuted blocks $\mathcal{G}_{\Delta,\bar{\Delta}}^{(s)}\left( 2134 \middle| \tfrac{x}{x-1} \right), \mathcal{G}_{\Delta,\bar{\Delta}}^{(t)}\left( 2134 \middle| \tfrac{x}{x-1} \right), \mathcal{G}_{\Delta,\bar{\Delta}}^{(u)}\left( 2134 \middle| \tfrac{x}{x-1} \right)$. We also write the analogous matrices

for the 5 remaining transpositions, plus 2 other involutive permutations. (We choose these 8 permutations for later convenience.) We use the notations $S_{\text{total}} = S_1 + S_2 + S_3 + S_4$ and $\Delta_{23}^{14} = \Delta_2 + \Delta_3 - \Delta_1 - \Delta_4$ and $\Delta_3^{124} = \Delta_3 - \Delta_1 - \Delta_2 - \Delta_4$. In the left column we reproduce Eq. (2.64), and generalize it to the other permutations:

$$
\begin{array}{|c|c|}
\hline
\mathcal{G}(x) = \left| (1-x)^{\Delta_3^{124}} \right|^2 \mathcal{G}\left(2134 \big| \tfrac{x}{x-1}\right) & \begin{bmatrix} (-)^{S+S_1+S_2} & & \\ & & (-)^{S+S_2+S_3} \\ & (-)^{S+S_1+S_3} & \end{bmatrix} \\
\hline
\mathcal{G}(x) = (-)^{S_{\text{total}}} \left| (1-x)^{-2\Delta_1} \right|^2 \mathcal{G}\left(1243 \big| \tfrac{x}{x-1}\right) & \begin{bmatrix} (-)^{S+S_3+S_4} & & \\ & & (-)^{S+S_2+S_3} \\ & (-)^{S+S_2+S_4} & \end{bmatrix} \\
\hline
\mathcal{G}(x) = (-)^{S_{\text{total}}} \mathcal{G}(1432|1-x) & \begin{bmatrix} & 1 & \\ 1 & & \\ & & (-)^{S+S_1+S_3} \end{bmatrix} \\
\hline
\mathcal{G}(x) = \left| x^{\Delta_{34}^{12}}(1-x)^{\Delta_{23}^{14}} \right|^2 \mathcal{G}(3214|1-x) & \begin{bmatrix} & 1 & \\ 1 & & \\ & & (-)^{S+S_2+S_4} \end{bmatrix} \\
\hline
\mathcal{G}(x) = (-)^{S_{\text{total}}} \left| x^{-2\Delta_1} \right|^2 \mathcal{G}\left(1324 \big| \tfrac{1}{x}\right) & \begin{bmatrix} & & 1 \\ & (-)^{S+S_1+S_4} & \\ 1 & & \end{bmatrix} \\
\hline
\mathcal{G}(x) = \left| x^{\Delta_3^{124}} \right|^2 \mathcal{G}\left(4231 \big| \tfrac{1}{x}\right) & \begin{bmatrix} & & 1 \\ & (-)^{S+S_1+S_4} & \\ 1 & & \end{bmatrix} \\
\hline
\mathcal{G}(x) = (-)^{S_{\text{total}}} \left| x^{\Delta_{34}^{12}}(1-x)^{\Delta_{23}^{14}} \right|^2 \mathcal{G}(3412|x) & \begin{bmatrix} 1 & & \\ & 1 & \\ & & (-)^{S_{\text{total}}} \end{bmatrix} \\
\hline
\mathcal{G}(x) = (-)^{S_{\text{total}}} \left| (1-x)^{\Delta_{23}^{14}} \right|^2 \mathcal{G}(2143|x) & \begin{bmatrix} 1 & & \\ & 1 & \\ & & (-)^{S_{\text{total}}} \end{bmatrix} \\
\hline
\end{array}
\tag{2.67}
$$

For example, $\mathcal{G}_{\Delta,\bar{\Delta}}^{(s)}(x) = (-)^{S+S_3+S_4} \left| (1-x)^{-2\Delta_1} \right|^2 \mathcal{G}_{\Delta,\bar{\Delta}}^{(s)}\left(1243 \big| \tfrac{x}{x-1}\right)$. Beware that the spin $S$ may refer to the $s$-channel, $t$-channel or $u$-channel spin, depending on the context. In our matrices, $S$ should therefore be viewed as an operator whose eigenvalues are channel spins.

**Behaviour of solutions of crossing symmetry**

The crossing symmetry equations for a 4-point function are schematically

$$
\forall x \in \mathbb{C}, \qquad \sum D^{(s)} \mathcal{G}^{(s)}(x) = \sum D^{(t)} \mathcal{G}^{(t)}(x) = \sum D^{(u)} \mathcal{G}^{(u)}(x), \tag{2.68}
$$

where the unknowns are the 4-point structure constants $\left( D^{(s)}, D^{(t)}, D^{(u)} \right)$. (See [29] for more details.) Given a solution for the 4-point function $\langle V_1 V_2 V_3 V_4 \rangle$, Eqs. (2.66) implies that $\left( (-)^{S+S_1+S_2} D^{(s)}, (-)^{S+S_1+S_3} D^{(u)}, (-)^{S+S_2+S_3} D^{(t)} \right)$ is a solution for $\langle V_2 V_1 V_3 V_4 \rangle$. If now $(\Delta_1, \bar{\Delta}_1) = (\Delta_2, \bar{\Delta}_2)$, then the equations for $\langle V_1 V_2 V_3 V_4 \rangle$ and $\langle V_2 V_1 V_3 V_4 \rangle$ coincide, and any solution can be uniquely decomposed as a sum of an even-spin and an odd-spin solution, which we define as solutions that obey the following linear constraints:

$$
\text{Even-spin solutions: } D_{(r,s)}^{(s)} \underset{S \in 2\mathbb{Z}+1}{=} 0, \quad D_{(r,s)}^{(t)} = (-)^{S+S_1+S_3} D_{(r,s)}^{(u)}, \tag{2.69}
$$

$$
\text{Odd-spin solutions: } D_{(r,s)}^{(s)} \underset{S \in 2\mathbb{Z}}{=} 0, \quad D_{(r,s)}^{(t)} = -(-)^{S+S_1+S_3} D_{(r,s)}^{(u)}, \tag{2.70}
$$

where $S = rs$. Let us generalize this to all possible coincidences of conformal dimensions in 4-point functions. If the dimensions of 2 fields coincide, then there is a channel where the conformal spin can be assumed to be either odd or even (the $s$-channel in our example), and

the 4-point structure constants of the other two channels are related. If the dimensions of 3 or 4 fields coincide, we may assume the spin to have definite parity in any channel, but there is no guarantee that an $s$-channel even-spin solution is also even-spin in the other channels. Examples:

- In the case of $\left\langle V_{(\frac{3}{2},\frac{2}{3})} V^3_{(\frac{1}{2},0)} \right\rangle$, the space of solutions of crossing symmetry is two-dimensional [29]. There are three different bases of solutions, parametrized by $x \in \{s, t, u\}$. Each basis is made of an $x$-channel even-spin solution, and an $x$-channel odd-spin solution.

- In the case of $\left\langle V_{(\frac{3}{2},0)} V^3_{(\frac{1}{2},0)} \right\rangle$, the space of solutions is still two-dimensional. There is a basis made of a solution that is even-spin in all channels, and a solution that is odd-spin in all channels.

Let us summarize the relations between 4-point structure constants for even-spin and odd-spin solutions, depending on which fields have the same dimensions. If 3 or 4 fields have the same dimensions, we consider solutions that are even-spin or odd-spin in all channels, if they exist. For solutions that are even-spin or odd-spin in only one channel, the relations are the same as when only 2 fields have the same dimensions:

| Equal dimensions | Even spin solutions | Odd spin solutions |
|---|---|---|
| 1, 2 or 3, 4 | $D^{(t)} = (-)^{S+S_2+S_3} D^{(u)}$ | $D^{(t)} = -(-)^{S+S_2+S_3} D^{(u)}$ |
| 2, 4 or 1, 3 | $D^{(s)} = (-)^{S_{\text{total}}} D^{(t)}$ | $D^{(s)} = -(-)^{S_{\text{total}}} D^{(t)}$ |
| 2, 3 | $D^{(s)} = (-)^{S_{\text{total}}} D^{(u)}$ | $D^{(s)} = -(-)^{S_{\text{total}}} D^{(u)}$ |
| 1, 4 | $D^{(s)} = D^{(u)}$ | $D^{(s)} = -D^{(u)}$ |
| 1, 2, 3 or 2, 3, 4 | $D^{(t)} = D^{(u)} = (-)^{S_{\text{total}}} D^{(s)}$ | $D^{(t)} = D^{(u)} = -(-)^{S_{\text{total}}} D^{(s)}$ |
| 1, 3, 4 or 1, 2, 4 | $D^{(s)} = D^{(u)} = (-)^{S_{\text{total}}} D^{(t)}$ | $D^{(s)} = -D^{(u)} = -(-)^{S_{\text{total}}} D^{(t)}$ |
| 1, 2, 3, 4 | $D^{(s)} = D^{(t)} = D^{(u)}$ | $D^{(s)} = -D^{(t)} = -D^{(u)}$ |

(2.71)

**Structure constants of a permuted 4-point function**

Permutation symmetry implies an equivalence between solutions of crossing symmetry for $\left\langle V_1 V_2 V_3 V_4 \right\rangle$ and for any permuted 4-point function, say $\left\langle V_2 V_1 V_3 V_4 \right\rangle$. The precise relation between solutions depend not only on how conformal blocks behave under the permutation, but also on how the solutions are normalized. In particular, notice that our normalization assumption (2.39) induces unnatural permutation behaviour: under permutations, reference structure constants are invariant, whereas structure constants in general pick a sign. In our example, this implies that if there is an $s$-channel diagonal field, then the $s$-channel structure constants for $\left\langle V_1 V_2 V_3 V_4 \right\rangle$ and $\left\langle V_2 V_1 V_3 V_4 \right\rangle$ differ by $(-)^S$ instead of the expected $(-)^{S+S_1+S_2}$.

**What about parity?**

Parity is another discrete symmetry of correlation functions. Conformal blocks are invariant under the parity transformation $V_{\Delta,\bar{\Delta}}(z) \rightarrow V_{\bar{\Delta},\Delta}(\bar{z})$ applied to all fields. Therefore, the exchange $\Delta \leftrightarrow \bar{\Delta}$ maps a solution of crossing symmetry to another solution. For a 4-point function of spinless fields $\left\langle \prod_{i=1}^4 V_{(r_i,0)} \right\rangle$, any solution can be decomposed into a parity-even term

and a parity-odd term. For example, the odd-spin solution for $\left\langle V_{(\frac{3}{2},0)} V^3_{(\frac{1}{2},0)} \right\rangle$ is also parity-odd.

Parity may be combined with permutation symmetry: for example, $\left\langle V_{(r,s)} V_{(r,-s)} V_{(r_3,0)} V_{(r_4,0)} \right\rangle$ is invariant under parity times the permutation $1 \leftrightarrow 2$. Parity may also be combined with the shift equations of Section 2.2: for example, the field $V_{(r,1)}$ is related to its parity image $V_{(r,-1)}$ by a shift $s \to s+2$. We refrain from systematically investigating all these possibilities, but we will indicate the symmetries of the solutions of crossing symmetry that we will find.

## 3 Numerical bootstrap results

### 3.1 Four-point functions of diagonal fields

Given the central charge $c$ and three loop weights $w_s, w_t, w_u$, there is a unique solution of crossing symmetry for the diagonal 4-point function $\left\langle \prod_{i=1}^4 V_{P_i} \right\rangle$, with diagonal fields of weight $w_x$ propagating in the channel $x \in \{s, t, u\}$ [25]. In any channel, the decomposition into conformal blocks is therefore given by Eq. (2.38). The normalized 4-point structure constants for the diagonal channel fields are known to be one: it remains to determine the structure constants $d^{(x)}_{(r,s)}$ for the non-diagonal $x$-channel fields $V_{(r,s)}$. According to the conjecture of Section 2.3, these structure constants are polynomial functions of the 8 loop weights $n, w_s, w_t, w_u, w_1, w_2, w_3, w_4$.

**Symmetries of normalized structure constants**

| Name | Origin | Equations | |
|------|--------|-----------|---|
| Partition | [25] | $\partial_{w_s} \partial_{w_t} d^{(x)}_{(r,s)} = 0$ | |
| Parity | Section 2.4 | $d^{(x)}_{(r,s)} = d^{(x)}_{(r,-s)}$ | |
| Permutation | Table (2.67) | $d^{(x)}_{(r,s)}\Big|_{\substack{w_1 \leftrightarrow w_2 \\ w_3 \leftrightarrow w_4}} = d^{(x)}_{(r,s)}$ $\quad$ $d^{(s)}_{(r,s)}\Big|_{\substack{w_1 \leftrightarrow w_2 \\ w_t \leftrightarrow w_u}} = (-)^{rs} d^{(s)}_{(r,s)}$ $\quad$ $d^{(s)}_{(r,s)}\Big|_{\substack{w_2 \leftrightarrow w_4 \\ w_s \leftrightarrow w_t}} = d^{(t)}_{(r,s)}$ $\quad$ $d^{(s)}_{(r,s)}\Big|_{\substack{w_2 \leftrightarrow w_3 \\ w_s \leftrightarrow w_u}} = d^{(u)}_{(r,s)}$ | (3.1) |
| Shift | Eq. (2.29) | $d^{(x)}_{(r,s)} = d^{(x)}_{(r,s+2)}$ | |
| Reflection | Eq. (2.34) | $d^{(x)}_{(r,s)}\Big|_{w_{1,2,3,4} \to -w_{1,2,3,4}} = d^{(x)}_{(r,s)}$ $\quad$ $d^{(s)}_{(r,s)}\Big|_{w_{1,2,t,u} \to -w_{1,2,t,u}} = (-)^r d^{(s)}_{(r,s)}$ $\quad$ $d^{(s)}_{(r,s)}\Big|_{w_{2,3,s,u} \to -w_{2,3,s,u}} = (-)^r d^{(s)}_{(r,s+1)}$ | |

Thanks to these symmetries, all structure constants $d^{(x)}_{(r,s)}$ can in principle be deduced from the $s$-channel case, with a second Kac index obeying $0 \le s \le \frac{1}{2}$. We will write a bit more than this minimal set, in order to make some symmetries manifest.

Let us sketch how we derive the reflection equations from the behaviour (2.34) of the 3-point structure constant. We first choose which loop weights should change signs, such that at each vertex in each channel, 0 or 2 edges are affected. In the case $w_{2,3,s,u} \to -w_{2,3,s,u}$, let

us draw the affected edges as dashed lines:

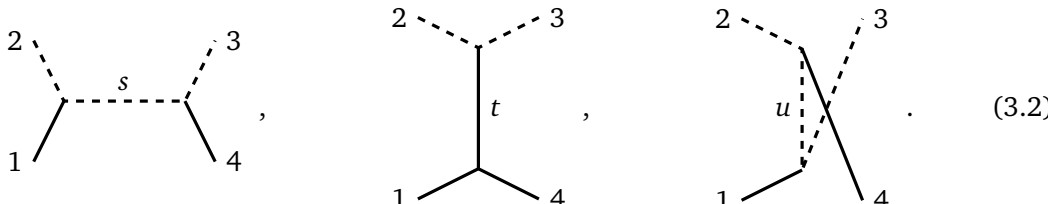

Now, the behaviour of a 4-point structure constant is given by the product of the sign factors coming from its vertices. These sign factors depend on the orientation around each vertex, since Eq. (2.34) is not invariant under orientation-reversing permutations. For example, $d_{(r,s)}^{(s)}$ picks no sign from the vertex $12s$, and a sign $(-)^r$ from the vertex $34s$. And reflection $w_s \to -w_s$ amounts to $(r,s) \to (r,s+1)$ for the non-diagonal fields in that channel.

**Degrees**

In numerical bootstrap results, we observe that normalized structure constants are indeed polynomial in loop weights, with the degrees:

| $(r,s)$ | $\deg_n d_{(r,s)}^{(s)}$ | $\deg_{w_s} d_{(r,s)}^{(s)}$ | $\deg_{w_t,w_u} d_{(r,s)}^{(s)}$ | $\deg_{w_i} d_{(r,s)}^{(s)}$ |
|---|---|---|---|---|
| $(1,0)$ | $0$ | $0$ | $1$ | $0$ |
| $(2,0)$ | $2$ | $1$ | $2$ | $1$ |
| $(2,\frac{1}{2})$ | $2$ | $0$ | $2$ | $1$ |
| $(3,0)(3,\frac{1}{3})$ | $6$ | $1$ | $3$ | $2$ |
| $(4,0)(4,\frac{1}{2})$ | $12$ | $2$ | $4$ | $3$ |
| $(4,\frac{1}{4})$ | $12$ | $1$ | $4$ | $3$ |
| $(5,0)(5,\frac{1}{5})(5,\frac{2}{5})$ | $20$ | $2$ | $5$ | $4$ |
| $(r,s)$ | $r(r-1)$ | $\leq \lfloor \frac{r}{2} \rfloor$ | $r$ | $r-1$ |

(3.3)

In the last row, we have used these results to conjecture the degrees for any $(r,s)$, or an upper bound in the case of $\deg_{w_s} d_{(r,s)}^{(s)}$. In particular, the degrees in $n$ saturate the bound (1.13), and the degrees in $w_s, w_t, w_u$ are consistent with the general formulas (1.19).

**Expression in terms of residues**

While the dependences of $d_{(r,s)}^{(x)}$ on $w_s, w_t, w_u$ do not mix thanks to the partition equations, the dependences on $w_1, w_2, w_3, w_4$ are not so simple. However, we already know a family of polynomials of these four variables: the residues $f_{(r,s)}^{(x)}$ (2.61) of the normalized structure constants with integer second index $s \in \mathbb{Z}$, or rather $s \in \{0,1\}$ since $f_{(r,s+2)}^{(x)} = f_{(r,s)}^{(x)}$. It is tempting to rewrite numerical results in terms of these residues. We have found that this is indeed possible, and that the structure constants in a given channel can actually be written in terms of the residues in that same channel.

For simplicity, we do not quite use the residues (2.56), but we remove their sign prefactors, and include a prefactor $\frac{1}{2}$. We focus on the $s$-channel residues,

$$p_{r,s} = \frac{1}{2} \rho_{(0,s_1)(0,s_2)}^{r,s} \rho_{(0,s_4)(0,s_3)}^{r,s}, \tag{3.4}$$

where $\rho^{r,s}$ is defined in Eq. (2.52), while the first few examples are displayed in Eq. (2.59).

The price to pay is that when rewritten as a polynomial in the residues $p_{r,s}$, structure constants are in general no longer polynomial in $n$. This phenomenon occurs for the first time in the structure constant $d^{(s)}_{(3,\frac{1}{3})}$, due to relations of the type

$$\frac{p_{2,1} - p_{2,0} + np_{1,1}^2 - np_{1,0}^2}{n(n^2 - 4)} = w_1 w_2 + w_3 w_4, \tag{3.5}$$

$$\frac{p_{2,0} + p_{2,1} - 2p_{1,0}^2 - 2p_{1,1}^2}{(n^2 - 4)} = w_1^2 + w_2^2 + w_3^2 + w_4^2 + w_1 w_2 w_3 w_4 + n^2 - 4. \tag{3.6}$$

**Explicit results**

For $r = 0, 1$, we explicitly write the structure constants in all three channels:

$$d^{(s,t,u)}_{\text{diag}} = 1, \tag{3.7a}$$

$$d^{(s)}_{(1,0)} = w_t + w_u, \qquad d^{(t)}_{(1,0)} = w_s + w_u, \qquad d^{(u)}_{(1,0)} = w_s + w_t, \tag{3.7b}$$

$$d^{(s)}_{(1,1)} = w_u - w_t, \qquad d^{(t)}_{(1,1)} = w_u - w_s, \qquad d^{(u)}_{(1,1)} = w_s - w_t. \tag{3.7c}$$

For $r = 2, 3$, we focus on $s$-channel structure constants, and we write the dependence on $w_1, w_2, w_3, w_4$ through the residues $p_{r,s}$ (3.4):

$$2d^{(s)}_{(2,0)} = (n^2 - 4)\left[w_t^2 + w_u^2 + 2w_s - 4\right] - (n-2)(w_t + w_u)p_{1,0} - (n+2)(w_t - w_u)p_{1,1}, \tag{3.7d}$$

$$2d^{(s)}_{(2,1)} = (n^2 - 4)\left[w_t^2 + w_u^2 - 2w_s - 4\right] - (n+2)(w_t + w_u)p_{1,0} - (n-2)(w_t - w_u)p_{1,1}, \tag{3.7e}$$

$$2d^{(s)}_{(2,\frac{1}{2})} = n^2(w_u^2 - w_t^2) + n(w_t - w_u)p_{1,0} + n(w_t + w_u)p_{1,1}, \tag{3.7f}$$

$$3d^{(s)}_{(3,0)} = n^2(n^2 - 4)^2\left[w_t^3 + w_u^3 - w_t - w_u\right] - n(n^2 - 4)^2\left[w_t^2 - w_u^2\right]p_{1,1}$$
$$\quad - n^2(n^2 - 4)\left[nw_t^2 + nw_u^2 - 4w_s - 4n\right]p_{1,0} + 2(n^2 - 4)(w_t - w_u)p_{1,0}p_{1,1}$$
$$\quad - \left[n^2 p_{1,0}^2 + (n^2 - 4)p_{1,1}^2 - n(n-2)p_{2,0} - n(n+2)p_{2,1}\right](w_t + w_u), \tag{3.7g}$$

$$3d^{(s)}_{(3,1)} = n^2(n^2 - 4)^2\left[w_t - w_u - w_t^3 + w_u^3\right] + n(n^2 - 4)^2\left[w_t^2 - w_u^2\right]p_{1,0}$$
$$\quad + n^2(n^2 - 4)\left[nw_t^2 + nw_u^2 + 4w_s - 4n\right]p_{1,1} - 2(n^2 - 4)(w_t + w_u)p_{1,0}p_{1,1}$$
$$\quad + \left[n^2 p_{1,1}^2 + (n^2 - 4)p_{1,0}^2 - n(n-2)p_{2,1} - n(n+2)p_{2,0}\right](w_t - w_u), \tag{3.7h}$$

$$\frac{3}{n^2 - 1}d^{(s)}_{(3,\frac{1}{3})} = (n^2 - 1)(n^2 - 3)\left[4w_t - 4w_u - w_t^3 + w_u^3\right] + n(n^2 - 1)(w_t^2 - w_u^2)p_{1,0}$$
$$\quad + (n^2 - 3)\left[nw_t^2 + nw_u^2 - 2w_s - 4n\right]p_{1,1} - 2(w_t + w_u)p_{1,0}p_{1,1}$$
$$\quad + \frac{w_t - w_u}{n^2 - 4}\left[(n^2 - 3)p_{1,1}^2 + (n^2 - 1)p_{1,0}^2 - \frac{n+1}{n}(n^2 - 3)p_{2,1} - \frac{n-1}{n}(n^2 - 3)p_{2,0}\right], \tag{3.7i}$$

$$\frac{3}{n^2 - 1}d^{(s)}_{(3,\frac{2}{3})} = (n^2 - 1)(n^2 - 3)\left[w_t^3 + w_u^3 - 4w_t - 4w_u\right] - n(n^2 - 1)(w_t^2 - w_u^2)p_{1,1}$$
$$\quad - (n^2 - 3)\left[nw_t^2 + nw_u^2 + 2w_s - 4n\right]p_{1,0} + 2(w_t - w_u)p_{1,0}p_{1,1}$$
$$\quad - \frac{w_t + w_u}{n^2 - 4}\left[(n^2 - 3)p_{1,0}^2 + (n^2 - 1)p_{1,1}^2 - \frac{n+1}{n}(n^2 - 3)p_{2,0} - \frac{n-1}{n}(n^2 - 3)p_{2,1}\right]. \tag{3.7j}$$

For $r \geq 4$, the larger degree of polynomials makes it impractical to determine them with our current code.

## 3.2 Cluster connectivities of the Potts model

Connectivities of Fortuin–Kasteleyn clusters in the $Q$-state Potts model have been receiving much attention, because they are simple geometrical observables of critical loop models. Computing connectivities has allowed various approaches to be tested and compared: Monte-Carlo [17, 32, 33], diagram algebras [26], conformal bootstrap [17, 20, 23]. Here we will write connectivities in terms of our diagonal 4-point functions, and give exact formulas for their structure constants up to $r = 6$, and for some structure constants for $r = 8, 10$.

**Expression in terms of diagonal 4-point functions**

Four-point connectivities are diagonal 4-point functions of the type $\left\langle V_{P_{(0,\frac{1}{2})}}^4 \right\rangle$. There is a unique solution of crossing symmetry $Z(w_s, w_t, w_u)$ for this 4-point function if we specify one diagonal field propagating in each channel. However, connectivities can have multiple propagating diagonal fields, which we now write together with the corresponding loop weights according to Eq. (1.14):

| Diagonal field | $V_{P_{(0,\frac{1}{2})}}$ | $V_{\langle 1,1 \rangle}$ | $V_{\langle 1,2 \rangle}$ |
|---|---|---|---|
| Loop weight | 0 | $n$ | $-n$ |

(3.8)

There exist four independent connectivities. Let us indicate which diagonal fields appear in which channel in each case:

| Connectivity | $s$-channel | $t$-channel | $u$-channel |
|---|---|---|---|
| $P^{aaaa}$ | $V_{P_{(0,\frac{1}{2})}}$ | $V_{P_{(0,\frac{1}{2})}}$ | $V_{P_{(0,\frac{1}{2})}}$ |
| $P^{aabb}$ | $V_{P_{(0,\frac{1}{2})}}, V_{\langle 1,1 \rangle}, V_{\langle 1,2 \rangle}$ | — | — |
| $P^{abba}$ | — | $V_{P_{(0,\frac{1}{2})}}, V_{\langle 1,1 \rangle}, V_{\langle 1,2 \rangle}$ | — |
| $P^{abab}$ | — | — | $V_{P_{(0,\frac{1}{2})}}, V_{\langle 1,1 \rangle}, V_{\langle 1,2 \rangle}$ |

(3.9)

This determines each connectivity as a linear combination of diagonal 4-point functions, up to normalization. To determine the normalizations, we use known relations for $s$-channel structure constants $d_0^{aaaa} = 2$ [22, 24] and $d_0^{aabb} = -d_0^{aaaa}$ [33], where the subscripts refer to the loop weight $w = 0$ of $V_{P_{(0,\frac{1}{2})}}$. This leads to

$$P^{aaaa} = 2Z(0,0,0), \tag{3.10a}$$

$$P^{aabb} = Z(n,0,0) + Z(-n,0,0) - 2Z(0,0,0), \tag{3.10b}$$

$$P^{abba} = Z(0,n,0) + Z(0,-n,0) - 2Z(0,0,0), \tag{3.10c}$$

$$P^{abab} = Z(0,0,n) + Z(0,0,-n) - 2Z(0,0,0). \tag{3.10d}$$

For example, in $P^{aabb}$, the linear combination of three terms leads to the three diagonal fields propagating in the $s$-channel. On the other hand, the contributions of the diagonal field $V_{P_{(0,\frac{1}{2})}}$ cancel between the terms in the $t$- and $u$-channels. This cancellation occurs because of the partition symmetry, which means that $w_s, w_t, w_u$ appear in different terms of $Z(w_s, w_t, w_u)$ [25].

Since we have multiple diagonal fields in the same channel, we must take linear combinations of pole terms. In particular:

$$\frac{D_{(r,0)}^{aabb}}{D_{(r,0)}^{\text{ref}}} = d_{(r,0)}^{aabb} - \frac{4n^2 p_{r,0}}{w(P_{(r,0)})\left(w(P_{(r,0)})^2 - n^2\right)}, \tag{3.11}$$

where $p_{r,0}$ is defined in Eq. (3.4).

**Symmetries of connectivities**

First of all, connectivities belong to the $Q$-state Potts model, so they should be functions of $Q = n^2$. Equivalently, they should be invariant under $n \to -n$. This does not follow from any symmetry that we have considered, although the formulas for $P^{aabb}, P^{abba}, P^{abab}$ (3.10) at least do not break this invariance via their explicit dependence on $n$. We find that numerical bootstrap results are indeed invariant.

We focus on the $s$-channel structure constants of connectivities. For the other channels we can use permutation symmetry, which relates for example the $t$-channel of $P^{abba}$ to the $s$-channel of $P^{aabb}$. Due to the permutation and reflection symmetries (3.1), we have

$$d_{(r,s)}^{aaaa} \neq 0 \implies \begin{cases} r \in 2\mathbb{N}^*, \\ rs \in 2\mathbb{Z}, \end{cases} \qquad d_{(r,s+1)}^{aaaa} = d_{(r,s)}^{aaaa}, \tag{3.12a}$$

$$d_{(r,s)}^{aabb} \neq 0 \implies \begin{cases} r \in 2\mathbb{N}^*, \\ rs \in 2\mathbb{Z}, \end{cases} \qquad d_{(r,s+1)}^{aabb} = d_{(r,s)}^{aaaa}, \tag{3.12b}$$

$$d_{(r,s)}^{abab} \neq 0 \implies r \in 2\mathbb{N}^*, \qquad d_{(r,s+1)}^{abab} = d_{(r,s)}^{abab}, \tag{3.12c}$$

$$d_{(r,s)}^{abba} = (-1)^{rs} d_{(r,s)}^{abab}. \tag{3.12d}$$

In particular, reflection symmetry, which ultimately follows from the existence of the degenerate field $V_{\langle 1,2\rangle}$, explains the vanishing of structure constants with odd first index $r \in 2\mathbb{N}+1$. We recover the known spectrum of connectivities [34], which is a rather sparse subset of the critical loop model spectrum (1.3). Thanks to this sparseness, it is possible to numerically determine structure constants that would be out of reach for a general 4-point function of diagonal fields.

**Polynomial factors of structure constants**

We have determined all structure constants with $r \leq 6$, and we will now display their polynomial factors. We label the structure constants of diagonal fields using their loop weights (3.8).

$$d_0^{aaaa} = 2, \tag{3.13a}$$

$$d_{(2,0)}^{aaaa} = -4(Q-4), \tag{3.13b}$$

$$d_{(4,0)}^{aaaa} = -2(Q-4)^3(Q-1)^2(Q-2), \tag{3.13c}$$

$$d_{(4,\frac{1}{2})}^{aaaa} = 2Q^3(Q-3)^2(Q-2), \tag{3.13d}$$

$$3d_{(6,0)}^{aaaa} = 4(Q-4)^5(Q^2-3Q+1)^2(Q-1)^2\left[Q^2-3(Q-1)^4\right], \tag{3.13e}$$

$$3d_{(6,\frac{1}{3})}^{aaaa} = 4(Q-1)^3(Q-4)(Q-3)Q\left[(Q-3)^2Q-1\right][(Q-4)(Q-3)(Q-2)Q+1], \tag{3.13f}$$

$$d_0^{aabb} = -2,\tag{3.14a}$$

$$d_{\pm n}^{aabb} = 1,\tag{3.14b}$$

$$d_{(2,0)}^{aabb} = 0,\tag{3.14c}$$

$$d_{(4,0)}^{aabb} = (Q-4)^3(Q-1)^2Q^2,\tag{3.14d}$$

$$d_{(4,\frac{1}{2})}^{aabb} = -Q^3(Q-3)^2(Q-2)^2,\tag{3.14e}$$

$$3d_{(6,0)}^{aabb} = 4(Q-4)^5(Q-2)(Q-1)^2Q^2(Q^2-3Q+1)^2,\tag{3.14f}$$

$$3d_{(6,\frac{1}{3})}^{aabb} = -2(Q-4)(Q-3)(Q-2)(Q-1)^3Q\left[(Q-3)^2Q-1\right]$$
$$\times\left[(Q-4)(Q-3)(Q-2)Q+1\right].\tag{3.14g}$$

$$d_{(2,0)}^{abab} = Q(Q-4),\tag{3.15a}$$

$$d_{(2,\frac{1}{2})}^{abab} = Q^2,\tag{3.15b}$$

$$2d_{(4,0)}^{abab} = (Q-4)^3(Q-1)^2Q(Q^2-3Q-2),\tag{3.15c}$$

$$2d_{(4,\frac{1}{2})}^{abab} = (Q-4)(Q-3)^2(Q-2)(Q-1)Q^3,\tag{3.15d}$$

$$2d_{(4,\frac{1}{4})}^{abab} = (Q-4)(Q-3)(Q-2)^2Q^2(Q^2-4Q+1),\tag{3.15e}$$

$$3d_{(6,0)}^{abab} = (Q-4)^5(Q-1)^2Q(Q^2-3Q+1)^2$$
$$\times\left[Q^6-9Q^5+30Q^4-40Q^3+13Q^2+4Q+3\right],\tag{3.15f}$$

$$3d_{(6,\frac{1}{6})}^{abab} = (Q-4)(Q-3)^3Q^2\left[(Q-3)^2Q-3\right]\left[(Q-4)(Q-2)(Q-1)Q+1\right]$$
$$\times\left[Q^5-10Q^4+34Q^3-43Q^2+11Q+5\right],\tag{3.15g}$$

$$3d_{(6,\frac{1}{3})}^{abab} = (Q-4)(Q-3)(Q-1)^3Q\left[(Q-3)^2Q-1\right]\left[(Q-4)(Q-3)(Q-2)Q+1\right]$$
$$\times\left[(Q-4)(Q^2-5Q+7)Q^2+Q+4\right],\tag{3.15h}$$

$$3d_{(6,\frac{1}{2})}^{abab} = (Q-3)^3Q^5(Q^2-5Q+5)^2$$
$$\times\left[Q^6-14Q^5+80Q^4-236Q^3+369Q^2-273Q+64\right].\tag{3.15i}$$

In addition, we have determined a couple of higher structure constants. We find that they are still polynomial, although a bit complicated:

$$d_{(8,0)}^{aaaa} = -(Q-4)^7(Q-1)^2(Q-2)(Q^2-3Q+1)^2\left[(Q-3)(Q-2)Q-1\right]^2$$
$$\times\left[Q^2-2(Q^2-3Q+1)^4\right],\tag{3.16}$$

$$5d_{(10,0)}^{aaaa} = -4(Q-4)^9(Q-1)^6(Q^2-3Q+1)^2\left[(Q-3)^2Q-1\right]^2\left[(Q-3)(Q-2)Q-1\right]^2$$
$$\times\left[5Q^{14}-110Q^{13}+1074Q^{12}-6142Q^{11}+22878Q^{10}-58418Q^9+104850Q^8\right.$$
$$\left.-133450Q^7+119948Q^6-74862Q^5+31496Q^4-8574Q^3+1420Q^2-130Q+5\right].\tag{3.17}$$

These polynomials agree with previous results for the ratios of structure constants [20,23,26]. (To perform the comparison, do not forget the pole terms from Eq. (2.61)!) Our new results are

however more general and more systematic; in particular we determine the structure constants themselves and not just some ratios thereof.

The degrees of our polynomials do not necessarily obey the bound (1.13). For example, $\deg_n d^{abab}_{(2,0)} = 4 > 2$. This is because the weights $w_x$ of channel diagonal fields are themselves functions of $n$. Taking their contributions into account, we obtain the bound $\deg_Q d^y_{(r,s)} \leq \frac{1}{2}r^2$ with $y \in \{aaaa, abab, abba, aabb\}$, which is obeyed in our examples.

We observe that $d^{aaaa}_{(r,0)}$ includes a prefactor $(Q-4)^{\frac{r}{2}-1} \prod_{j=1}^{\frac{r}{2}} \left( T_{2j-1}(\frac{Q-2}{2}) - 1 \right)$, where $T_j$ is a Chebyshev polynomial of the first kind. Using the identity

$$T_{2j+1}(x) - 1 = (x-1)\left(U_j(x) + U_{j-1}(x)\right)^2, \tag{3.18}$$

where $U_j$ is a Chebyshev polynomial of the second kind, this prefactor can be rewritten as $(Q-4)^{r-1} \prod_{j=1}^{\frac{r}{2}} \left( U_{j-1}(\frac{Q-2}{2}) + U_{j-2}(\frac{Q-2}{2}) \right)$. The relevant combinations of Chebyshev polynomials are

$$U_0 = 1, \tag{3.19a}$$
$$U_1 + U_0 = Q - 1, \tag{3.19b}$$
$$U_2 + U_1 = Q^2 - 3Q + 1, \tag{3.19c}$$
$$U_3 + U_2 = (Q-3)(Q-2)Q - 1, \tag{3.19d}$$
$$U_4 + U_3 = (Q-1)\left((Q-3)^2 Q - 1\right). \tag{3.19e}$$

After the prefactor, we have a polynomial whose coefficients have alternating signs, which is a good omen that its expression for all $r$ might be tractable.

## 3.3 Four-point functions of diagonal and non-diagonal fields

When non-diagonal fields are involved, we typically find that some structure constants vanish, for three types of reasons:

- *Vanishing due to symmetry.* This may happen to odd-spin or even-spin structure constants, when our solutions are odd-spin or even-spin. This may also happen to structure constants of the type $d^{(x)}_{(r,0)}$ or $d^{(x)}_{(r,1)}$, due to parity.

- *Vanishing due to signature-dependent bounds.* By construction, we impose the signature-dependent constraints (1.16). Moreover, in the presence of an $s$-channel diagonal field, i.e. if the signature is of the type $(0, \sigma, \sigma)$, we observe the bound (1.19) on the degrees of $s$-channel structure constants, which implies

$$1 \leq r < 2\sigma \implies d^{(s)}_{(r,s)} = 0. \tag{3.20}$$

  For the sake of brevity, we will omit such vanishing structure constants.

- *Vanishing for no known reason.* This happens rarely, and will be pointed out explicitly.

If there is an $s$-channel diagonal field, we fix the overall normalization of the solution by setting the corresponding structure constant $d^{(s)}_{\text{diag}}$ to one. If not, we set the structure constant(s) $d^{(x)}_{(r,s)}$ with the lowest first index $r$ to be $n$-independent numbers. With these normalizations, we observe that the structure constants $d^{(x)}_{(r,s)}$ are polynomial.

**Case of** $\langle V_{(1,0)} V_{P_1} V_{P_2} V_{P_3} \rangle$ **with** $\sigma = (0,1,1)$  

We indicate the symmetries of the normalized 4-point structure constants, in the same way as in Table (3.1):

| Parity | $d^{(x)}_{(r,s)} = d^{(x)}_{(r,-s)}$ | |
|---|---|---|
| Permutation | $d^{(s)}_{(r,s)}\big|_{w_2 \leftrightarrow w_3} = (-)^{rs} d^{(s)}_{(r,s)}$ , $\quad d^{(t)}_{(r,s)}\big|_{w_2 \leftrightarrow w_3} = (-)^{rs} d^{(u)}_{(r,s)}$ | |
| Shift | $d^{(x)}_{(r,s)} = d^{(x)}_{(r,s+2)}$ | |
| Reflection | $d^{(s)}_{(r,s)}\big|_{-w_{2,3}} = (-)^r d^{(s)}_{(r,s)}$ , $\quad d^{(s)}_{(r,s)}\big|_{-w_{1,2,s}} = (-)^{r+\delta_{r \neq 0}} d^{(s)}_{(r,s+1)}$ <br> $d^{(t)}_{(r,s)}\big|_{-w_{2,3}} = (-)^r d^{(t)}_{(r,s+1)}$ , $\quad d^{(t)}_{(r,s)}\big|_{-w_{1,2,s}} = (-)^{r+1} d^{(t)}_{(r,s)}$ | (3.21) |

There is a subtlety with the reflection equations: the structure constant $d^{(s)}_{\text{diag}}$, which we normalize to one, is predicted by Eq. (2.34) to change sign under $w_{1,2,s} \to -w_{1,2,s}$. When writing the behaviour of a structure constant $d^{(x)}_{(r,s)}$, we actually display the behaviour of the ratio $\frac{d^{(x)}_{(r,s)}}{d^{(s)}_{\text{diag}}}$, including a minus sign from the denominator.

Vanishing structure constant $d^{(s)}_{(2,\frac{1}{2})} = 0$. This follows from permutation symmetry, if we assume that $d^{(s)}_{(2,\frac{1}{2})}$ does not depend on $w_i$.

$$d^{(s)}_{\text{diag}} = 1 \,, \tag{3.22a}$$

$$d^{(s)}_{(2,0)} = n^2 - 4 \,, \tag{3.22b}$$

$$d^{(s)}_{(2,1)} = -(n^2 - 4) \,, \tag{3.22c}$$

$$3 d^{(s)}_{(3,0)} = -2n^2 (n^2 - 4)(n - w_1)(w_2 + w_3) \,, \tag{3.22d}$$

$$3 d^{(s)}_{(3,\frac{1}{3})} = -(n^2 - 3)(n^2 - 1)(n + w_1)(w_2 - w_3) \,, \tag{3.22e}$$

$$3 d^{(s)}_{(3,\frac{2}{3})} = (n^2 - 3)(n^2 - 1)(n - w_1)(w_2 + w_3) \,, \tag{3.22f}$$

$$3 d^{(s)}_{(3,1)} = 2n^2 (n^2 - 4)(n + w_1)(w_2 - w_3) \,. \tag{3.22g}$$

$$d^{(t)}_{(1,0)} = 1 \,, \tag{3.22h}$$

$$d^{(t)}_{(1,1)} = -1 \,, \tag{3.22i}$$

$$2 d^{(t)}_{(2,0)} = w_s (n^2 - 4) + (2 + w_3)(2w_1 - n w_2) \,, \tag{3.22j}$$

$$2 d^{(t)}_{(2,\frac{1}{2})} = n(w_2 w_3 - w_s n) \,, \tag{3.22k}$$

$$2 d^{(t)}_{(2,1)} = w_s (n^2 - 4) + (2 - w_3)(2w_1 + n w_2) \,. \tag{3.22l}$$

Let us now display the degrees of polynomials for this 4-point function. We write 0 if the degree vanishes, and "zero" denotes vanishing structure constants. These degrees obey the

conjectures (1.13) and (1.19).

| $(r,s)$ | $\deg_n d^{(s)}_{(r,s)}$ | $\deg_{w_s} d^{(s)}_{(r,s)}$ | $\deg_{w_s} d^{(t,u)}_{(r,s)}$ | $\deg_{w_1} d^{(s)}_{(r,s)}$ | $\deg_{w_1} d^{(t,u)}_{(r,s)}$ |
|---|---|---|---|---|---|
| $(1,0)(1,1)$ | zero | zero | 0 | zero | 0 |
| $(2,0)(2,1)$ | 2 | 0 | 1 | 0 | 1 |
| $(2,\frac{1}{2})$ | zero | zero | 1 | 0 | 0 |
| $(3,0)(3,1)$ | 5 | 0 | 2 | 1 | 2 |
| $(3,\frac{1}{3})(3,\frac{2}{3})$ | 5 | 0 | 2 | 1 | 1 |
| $(4,0)(4,1)$ | 12 | 1 | 3 | 2 | 3 |
| $(4,\frac{1}{2})$ | 12 | 1 | 3 | 2 | 2 |
| $(4,\frac{1}{4})$ | 8 | 1 | 3 | 2 | 2 |
| $(r,s)$ | $\leq r(r-1)$ | $\lfloor \frac{r}{2}-1 \rfloor$ | $r-1$ | $r-2$ | $\leq r-1$ |

(3.23)

**Case of** $\left\langle V_{(\frac{1}{2},0)} V_{(\frac{1}{2},0)} V_{P_1} V_{P_2} \right\rangle$ **with** $\sigma = (0,\frac{1}{2},\frac{1}{2})$ 

| Parity | $d^{(x)}_{(r,s)} = d^{(x)}_{(r,-s)}$ |
|---|---|
| Permutation | $d^{(x)}_{(r,s)}\big|_{w_1 \leftrightarrow w_2} = d^{(x)}_{(r,s)} , \; d^{(s)}_{(r,s)} = (-)^{rs} d^{(s)}_{(r,s)} , \; d^{(t)}_{(r,s)} = (-)^{rs} d^{(u)}_{(r,s)}$ |
| Shift | $d^{(x)}_{(r,s)} = -d^{(x)}_{(r,s+2)}$ |

(3.24)

Vanishing structure constants: $d^{(s)}_{\text{odd-spin}} = 0$ from permutation symmetry, and $d^{(s)}_{(r,1)} = 0$ from the parity and shift equations.

$$d^{(s)}_{\text{diag}} = 1 \,, \tag{3.25a}$$

$$d^{(s)}_{(1,0)} = 2 \,, \tag{3.25b}$$

$$d^{(s)}_{(2,0)} = 2(n-2)\big[n+2-p_{1,0}\big] \,, \tag{3.25c}$$

$$3d^{(s)}_{(3,0)} = 2n^2(n-2)\big[(n^2-4)\big((n+2)(w_s+1)-2p_{1,0}\big)-p_{2,0}-(n+1)p_{1,0}^2\big] \,, \tag{3.25d}$$

$$3d^{(s)}_{(3,\frac{2}{3})} = 2(n^2-3)(n^2-1)(n+1)\Big[-(n-1)(w_s-2)-p_{1,0}+\frac{p_{2,0}}{n^2-4}+\frac{p_{1,0}^2}{n+2}\Big] \,. \tag{3.25e}$$

Here the residues $p_{r,s}$ are given by Eq. (3.4), and their values are

$$p_{1,0} = w_1 + w_2 \,, \tag{3.25f}$$

$$p_{2,0} = 2(1-n)\big(w_1^2+w_2^2+n^2-nw_1w_2-4\big) \,. \tag{3.25g}$$

$$d^{(t)}_{(\frac{1}{2},0)} = 1\,, \tag{3.25h}$$

$$3d^{(t)}_{(\frac{3}{2},0)} = (w_1+2)(w_2+2)-(n+2)(w_s+2)\,, \tag{3.25i}$$

$$3d^{(t)}_{(\frac{3}{2},\frac{2}{3})} = 2\left[(w_1-1)(w_2-1)-(n-1)(w_s-1)\right]\,, \tag{3.25j}$$

$$5d^{(t)}_{(\frac{5}{2},0)} = w_s^2(n+2)^2(n-1)^2 - w_s(n+2)(n-1)\left[n(w_1w_2+2)+w_1(w_2-2)-2w_2\right]$$
$$+ (n-w_1)(n-w_2)\left[3n^2-n(w_1+w_2-2)+w_1w_2-4\right]\,, \tag{3.25k}$$

$$5d^{(t)}_{(\frac{5}{2},\frac{2}{5})} = 2n\left[\varphi(2n^2-n-4)(n-1)-2n^2+n+3\right]$$
$$+ 2\varphi(n-\varphi)\left[2n^2-2n-\varphi-2\right]w_s - 2(n-\varphi)(n+\varphi^{-1})[\varphi(n^2-2)-n-1]w_s^2$$
$$- 2(n-1)(n-\varphi)w_s(w_1+w_2) - 2\left[(n+1)(n-2)^2\varphi-2n^2+2n+3\right](w_1+w_2)$$
$$+ 2\varphi(n-\varphi)(n^2-1)w_sw_1w_2 - 2[\varphi(n^2-2)+n-1]w_1w_2$$
$$+ 2nw_1w_2(w_1+w_2) - 2\left[\varphi(n^2-3)-1\right](w_1^2+w_2^2) - 2\varphi w_1^2 w_2^2\,, \tag{3.25l}$$

$$d^{(t)}_{(\frac{5}{2},\frac{4}{5})} = -d^{(t)}_{(\frac{5}{2},\frac{2}{5})}\Big|_{\varphi\to-\varphi^{-1}}\,, \tag{3.25m}$$

where we use the golden ratio

$$\varphi = \frac{1+\sqrt{5}}{2} = 2\cos\left(\frac{\pi}{5}\right)\,. \tag{3.26}$$

The degrees of these polynomials obey the conjectures (1.13) and (1.19).

**Case of** $\left\langle V_{(\frac{3}{2},0)}V_{(\frac{1}{2},0)}V_{P_1}V_{P_2}\right\rangle$ **with** $\sigma = (0,\frac{3}{2},\frac{3}{2})$ 

| | |
|---|---|
| Parity | $d^{(x)}_{(r,s)} = d^{(x)}_{(r,-s)}$ |
| Permutation | $d^{(s)}_{(r,s)}\Big|_{w_1\leftrightarrow w_2} = (-)^{rs}d^{(s)}_{(r,s)}\quad,\quad d^{(t)}_{(r,s)}\Big|_{w_1\leftrightarrow w_2} = (-)^{rs}d^{(u)}_{(r,s)}$ |
| Shift | $d^{(s)}_{(r,s)} = (-)^{\delta_{r\geq2}}d^{(s)}_{(r,s+2)}\quad,\quad d^{(t)}_{(r,s)} = -d^{(t)}_{(r,s+2)}$ |

(3.27)

Vanishing structure constant: $d^{(s)}_{(3,\frac{1}{3})} = 0$, which follows from permutation symmetry if we assume that this structure constant does not depend on $w_i$. Moreover, the parity and shift equations imply $d^{(s)}_{(r\geq2,1)} = 0$.

$$d^{(s)}_{\text{diag}} = 1\,, \tag{3.28a}$$

$$3d^{(s)}_{(3,0)} = 2n^2\left(n^2-4\right)^2\,, \tag{3.28b}$$

$$3d^{(s)}_{(3,\frac{2}{3})} = -2\left(n^2-3\right)\left(n^2-1\right)^2\,, \tag{3.28c}$$

$$3d^{(t)}_{(\frac{3}{2},0)} = -(n+2), \tag{3.28d}$$

$$3d^{(t)}_{(\frac{3}{2},\frac{2}{3})} = -2(n-1), \tag{3.28e}$$

$$5d^{(t)}_{(\frac{5}{2},0)} = (n-1)(n+2)\Big[(w_2+2)\big(2-(n+1)w_1\big)+(n-1)(n+2)w_s\Big], \tag{3.28f}$$

$$5d^{(t)}_{(\frac{5}{2},\frac{2}{5})} = 2(n-\varphi)\Big[(n-1)(\varphi w_2+1)\big((n+1)w_1+1-\varphi\big)$$
$$-(\varphi n+1)\big(n^2+(1-\varphi)n-\varphi-1\big)w_s\Big], \tag{3.28g}$$

$$5d^{(t)}_{(\frac{5}{2},\frac{4}{5})} = 2\varphi^{-2}(\varphi n+1)\Big[(n-1)(w_2-\varphi)\big((n+1)w_1+\varphi\big)$$
$$-(n-\varphi)\big(n^2+\varphi n+\varphi-2\big)w_s\Big]. \tag{3.28h}$$

The degrees of these polynomials obey the conjectures (1.13) and (1.19).

**Case of** $\left\langle V_{(1,1)}V_{(\frac{1}{2},0)}V_{(\frac{1}{2},0)}V_P\right\rangle$ **with** $\sigma=(\frac{1}{2},1,\frac{1}{2})$ 

| | | |
|---|---|---|
| Parity | $d^{(s)}_{(r,s)} = (-)^{\delta_{r\geq\frac{3}{2}}}d^{(s)}_{(r,-s)},$ | $d^{(t)}_{(r,s)} = -d^{(t)}_{(r,-s)}$ |
| Permutation | $d^{(t)}_{(r,s)} = -(-)^{rs}d^{(t)}_{(r,s)},$ | $d^{(s)}_{(r,s)} = d^{(u)}_{(r,s)}$ |
| Shift | $d^{(s)}_{(r,s)} = (-)^{\delta_{r\geq\frac{3}{2}}}d^{(s)}_{(r,s+2)},$ | $d^{(t)}_{(r,s)} = -d^{(t)}_{(r,s+2)}$ |

$$\tag{3.29}$$

In this case, it would seem that we do not have parity symmetry, because $V_{(1,1)}$ is not invariant under parity. However, its image $V_{(1,-1)}$ is related by a shift. The parity behaviour of structure constants is therefore obtained by using the shift equation for $V_{(1,1)}$. The parity symmetry implies $d^{(x)}_{(r,0)} = 0$ (except $d^{(s)}_{(\frac{1}{2},0)}$), consistently with the fact that we have a $t$-channel odd-spin solution i.e. $d^{(t)}_{\text{even-spin}} = 0$.

$$d^{(s)}_{(\frac{1}{2},0)} = 1, \tag{3.30a}$$

$$3d^{(s)}_{(\frac{3}{2},\frac{2}{3})} = -2\sqrt{3}(w-n), \tag{3.30b}$$

$$5d^{(s)}_{(\frac{5}{2},\frac{2}{5})} = 4\cos\left(\frac{\pi}{10}\right)\varphi^{-2}(\varphi n+1)(n-w)\Big[\varphi w-n^2+(\varphi-2)n+2\varphi+1\Big], \tag{3.30c}$$

$$5d^{(s)}_{(\frac{5}{2},\frac{4}{5})} = 4\cos\left(\frac{\pi}{10}\right)(n-\varphi)(w-n)\Big[\varphi^{-1}w+n^2+(\varphi+1)n+2\varphi-3\Big]. \tag{3.30d}$$

$$d^{(t)}_{(1,1)} = -2, \tag{3.30e}$$

$$d^{(t)}_{(2,\frac{1}{2})} = \sqrt{2}n(w-n), \tag{3.30f}$$

$$3d^{(t)}_{(3,\frac{1}{3})} = 2(n-1)^2(n+1)(w-n)[2w+(n+3)(n-1)], \tag{3.30g}$$

$$3d^{(t)}_{(3,1)} = 2(n-2)(n+2)^2(w-n)(w+2n^2-5n). \tag{3.30h}$$

## 3.4 Four-point functions of non-diagonal fields

**Case of** $\left\langle V_{(\frac{1}{2},0)}V_{(\frac{1}{2},0)}V_{(\frac{1}{2},0)}V_{(\frac{1}{2},0)} \right\rangle$ **with** $\sigma = (0,1,1)$ 

| | |
|---|---|
| Parity | $d^{(x)}_{(r,s)} = d^{(x)}_{(r,-s)}$ |
| Permutation | $d^{(s)}_{(r,s)} = (-1)^{rs}d^{(s)}_{(r,s)}$ , $d^{(t)}_{(r,s)} = (-)^{rs}d^{(u)}_{(r,s)}$ |
| Shift | $d^{(x)}_{(r,s)} = d^{(x)}_{(r,s+2)}$ |

$$(3.31)$$

$$d^{(s)}_{\text{diag}} = 1, \tag{3.32a}$$

$$d^{(s)}_{(2,0)} = n^2 - 4, \tag{3.32b}$$

$$d^{(s)}_{(2,1)} = -(n^2 - 4), \tag{3.32c}$$

$$3d^{(s)}_{(3,0)} = -8n^2(n-2)^2(n+2), \tag{3.32d}$$

$$3d^{(s)}_{(3,\frac{2}{3})} = 4(n^2-1)(n^2-3)(n-2), \tag{3.32e}$$

$$2d^{(s)}_{(4,0)} = n^2(n-2)^3(n+1)^2(n+2)$$
$$\times \left[ w_s(n+2)^2(n-1)^2 + 2n^4 - 6n^2 - 8n + 16 \right], \tag{3.32f}$$

$$2d^{(s)}_{(4,\frac{1}{2})} = -n^3(n^2-2)(n^2-3)$$
$$\times \left[ w_s n(n^2-2)(n^2-3) - 4n^4 + 4n^3 + 8n^2 + 4n - 16 \right], \tag{3.32g}$$

$$2d^{(s)}_{(4,1)} = n^2(n^2-4)^3(n-1)^2 \left[ w_s(n+1)^2 - 2n^2 - 8n - 10 \right]. \tag{3.32h}$$

---

$$d^{(t)}_{(1,0)} = 1, \tag{3.32i}$$

$$d^{(t)}_{(1,1)} = -1, \tag{3.32j}$$

$$2d^{(t)}_{(2,0)} = (n-2)\left[ w_s(n+2) - 8 \right], \tag{3.32k}$$

$$2d^{(t)}_{(2,\frac{1}{2})} = -n(w_s n - 4), \tag{3.32l}$$

$$2d^{(t)}_{(2,1)} = w_s(n^2 - 4), \tag{3.32m}$$

$$3d^{(t)}_{(3,0)} = n^2(n-2)^2 \left[ w_s^2(n+2)^2 - 4w_s(n+2) + n^2 + 8 \right], \tag{3.32n}$$

$$3d^{(t)}_{(3,\frac{1}{3})} = -(n-1)^2(n+1)$$
$$\times \left[ (n^2-3)(n+1)w_s^2 - 2(n+1)(2n-3)w_s - 2(n-2)(n^2+4n+1) \right], \tag{3.32o}$$

$$3d^{(t)}_{(3,\frac{2}{3})} = (n^2-3)(n^2-1)\left[ w_s^2(n^2-1) - 2w_s(2n-1) - 2(n+1)(n-2) \right], \tag{3.32p}$$

$$3d^{(t)}_{(3,1)} = -(n^2-4)^2 \left[ w_s^2 n^2 - 4w_s n + (n+2)^2 \right]. \tag{3.32q}$$

**Case of** $\left\langle V_{(\frac{1}{2},0)}V_{(\frac{1}{2},0)}V_{(1,0)}V_{(1,0)}\right\rangle$ **and** $\left\langle V_{(\frac{1}{2},0)}V_{(\frac{1}{2},0)}V_{(1,1)}V_{(1,1)}\right\rangle$ **with** $\sigma = (0, \frac{3}{2}, \frac{3}{2})$ 

| Parity | $d^{(x)}_{(r,s)} = d^{(x)}_{(r,-s)}$ |
|---|---|
| Permutation | $d^{(s)}_{(r,s)} = (-)^{rs} d^{(s)}_{(r,s)}$ , $d^{(t)}_{(r,s)} = (-)^{rs} d^{(u)}_{(r,s)}$ |
| Shift | $d^{(s)}_{(r,s)} = (-)^{\delta_{r\geq 2}} d^{(s)}_{(r,s+2)}$ , $d^{(t,u)}_{(r,s)} = -d^{(t,u)}_{(r,s+2)}$ |

$$(3.33)$$

Vanishing structure constants due to symmetry: $d^{(s)}_{\text{odd-spin}} = d^{(s)}_{(r,1)} = 0$.

We find that the normalized structure constants $d^{(x)}_{(r,s)}$ for the two 4-point functions that we consider are identical to the leading order in $w_s$. In particular, the structure constants of the types $d^{(s)}_{(r\leq 3,s)}$ and $d^{(t)}_{(r\leq\frac{3}{2},s)}$ coincide, as they do not depend on $w_s$. In the analytic results that we will display, only the structure constants $d^{(t)}_{(\frac{5}{2},0)}, d^{(t)}_{(\frac{5}{2},\frac{2}{5})}$ and $d^{(t)}_{(\frac{5}{2},\frac{4}{5})}$ differ, in which case we indicate the results for $\left\langle V_{(\frac{1}{2},0)}V_{(\frac{1}{2},0)}V_{(1,0)}V_{(1,0)}\right\rangle$ above the results for $\left\langle V_{(\frac{1}{2},0)}V_{(\frac{1}{2},0)}V_{(1,1)}V_{(1,1)}\right\rangle$.

$$d^{(s)}_{\text{diag}} = 1\,, \tag{3.34a}$$

$$3d^{(s)}_{(3,0)} = 2n^2(n^2 - 4)^2\,, \tag{3.34b}$$

$$3d^{(s)}_{(3,\frac{2}{3})} = -2(n^2 - 3)(n^2 - 1)^2\,. \tag{3.34c}$$

$$3d^{(t)}_{(\frac{3}{2},0)} = -(n+2)\,, \tag{3.34d}$$

$$3d^{(t)}_{(\frac{3}{2},\frac{2}{3})} = -2(n-1)\,, \tag{3.34e}$$

$$5d^{(t)}_{(\frac{5}{2},0)} = (n+2)^2(n-1)^2 w_s - \begin{cases} 8n(n+2)(n-1)\,, \\ 0\,, \end{cases} \tag{3.34f}$$

$$5d^{(t)}_{(\frac{5}{2},\frac{2}{5})} = -2(n^2 - n - 1)\left[\varphi n^2 - n - 2\varphi - 1\right] w_s \tag{3.34g}$$

$$+ 2(n-1)\begin{cases} \left[(4\varphi+2)n^2 - (3\varphi+3)n - 4\varphi - 3\right]\,, \\ \left[(4\varphi-2)n^2 + (\varphi-3)n - 4\varphi - 3\right]\,, \end{cases} \tag{3.34h}$$

$$5d^{(t)}_{(\frac{5}{2},\frac{4}{5})} = -2(n^2 - n - 1)\varphi^{-1}\left[n^2 + \varphi n + \varphi - 2\right] w_s \tag{3.34i}$$

$$+ 2(n-1)\begin{cases} \left[(4\varphi-6)n^2 - (3\varphi-6)n - 4\varphi + 7\right]\,, \\ \left[(4\varphi-2)n^2 + (\varphi+2)n - 4\varphi + 7\right]\,. \end{cases} \tag{3.34j}$$

**Case of** $\left\langle V_{(\frac{1}{2},0)}V_{(\frac{1}{2},0)}V_{(1,0)}V_{(1,0)}\right\rangle$ **with** $\sigma = (1, \frac{1}{2}, \frac{3}{2})$

We again consider the 4-point functions $\left\langle V^2_{(\frac{1}{2},0)}V^2_{(\frac{1}{2},0)}\right\rangle$ and $\left\langle V^2_{(\frac{1}{2},0)}V^2_{(1,1)}\right\rangle$, but with a different signature. This signature breaks the permutation symmetry, so that there is no longer a simpler relation between the $t$- and $u$-channels, and the $s$-channel spectrum is no longer even-spin. Moreover, the structure constants for our two 4-point functions are less similar.

| Parity | $d^{(x)}_{(r,s)} = d^{(x)}_{(r,-s)}$ |
|---|---|
| Shift | $d^{(s)}_{(r,s)} = (-)^{\delta_{r\geq 2}} d^{(s)}_{(r,s+2)}$ , $d^{(t,u)}_{(r,s)} = -d^{(t,u)}_{(r,s+2)}$ |

$$(3.35)$$

The shift and parity relations imply $d^{(s)}_{(r\geq 2,1)} = 0$.

$$d^{(s)}_{(1,0)} = 1\,, \tag{3.36a}$$

$$d^{(s)}_{(1,1)} = -1\,, \tag{3.36b}$$

$$d^{(s)}_{(2,0)} = n(n-2)\,, \tag{3.36c}$$

$$2d^{(s)}_{(2,\frac{1}{2})} = -\sqrt{2}n(n-2)\,, \tag{3.36d}$$

$$3d^{(s)}_{(3,0)} = 2n^3(n-4)(n-2)(n+1)\,, \tag{3.36e}$$

$$3d^{(s)}_{(3,\frac{1}{3})} = -\sqrt{3}(n-2)(n-1)^2(n+1)\left(n^2+1\right)\,, \tag{3.36f}$$

$$3d^{(s)}_{(3,\frac{2}{3})} = n(n-1)(n+1)^2(n^2-3)\,, \tag{3.36g}$$

$$2d^{(s)}_{(4,\frac{1}{2})} = -\sqrt{2}n^5\left(n^2-3\right)\left(n^2-2\right)^2\,, \tag{3.36h}$$

$$d^{(s)}_{(4,0)} = n^3(n-2)^3(n+1)^2\left(n^4+2n^3+n^2-n-2\right)\,. \tag{3.36i}$$

---

$$d^{(t)}_{(\frac{1}{2},0)} = 1\,, \tag{3.36j}$$

$$3d^{(t)}_{(\frac{3}{2},0)} = 4\,, \tag{3.36k}$$

$$3d^{(t)}_{(\frac{3}{2},\frac{2}{3})} = 2\,, \tag{3.36l}$$

$$5d^{(t)}_{(\frac{5}{2},0)} = 2\left[n^4+2n^3-n^2-12n+12\right]\,, \tag{3.36m}$$

$$5d^{(t)}_{(\frac{5}{2},\frac{2}{5})} = 2\left[-n^4+2n^2+2n-1\right]+2\varphi\left[n^3-2n^2-2\right]\,, \tag{3.36n}$$

$$5d^{(t)}_{(\frac{5}{2},\frac{4}{5})} = 2\left[n^4-n^3-2n+3\right]+2\varphi\left[n^3-2n^2-2\right]\,. \tag{3.36o}$$

---

$$3d^{(u)}_{(\frac{3}{2},0)} = -2(n+2)\,, \tag{3.36p}$$

$$3d^{(u)}_{(\frac{3}{2},\frac{2}{3})} = -2(n-1)\,, \tag{3.36q}$$

$$5d^{(u)}_{(\frac{5}{2},0)} = 2(n-1)(n+2)\left[n^2-3n+6\right]\,, \tag{3.36r}$$

$$5d^{(u)}_{(\frac{5}{2},\frac{2}{5})} = -2\left[n^4-3n^3+n+1\right]-2\varphi\left[n^4-5n^3+3n^2+4n+2\right]\,, \tag{3.36s}$$

$$5d^{(u)}_{(\frac{5}{2},\frac{4}{5})} = -2\left[2n^4-8n^3+3n^2+5n+3\right]+2\varphi\left[n^4-5n^3+3n^2+4n+2\right]\,. \tag{3.36t}$$

**Case of** $\left\langle V_{(\frac{1}{2},0)}V_{(\frac{1}{2},0)}V_{(1,1)}V_{(1,1)}\right\rangle$ **with** $\sigma = (1,\frac{1}{2},\frac{3}{2})$ 

The symmetries and vanishing structure constants are as in the case $\left\langle V^2_{(\frac{1}{2},0)}V^2_{(1,0)}\right\rangle$, except that we notice the additional vanishing $d^{(t)}_{(\frac{3}{2},0)} = 0$ – a rare example of a vanishing for no known

reason.

$$d^{(s)}_{(1,0)} = 1, \tag{3.37a}$$

$$d^{(s)}_{(1,1)} = -1, \tag{3.37b}$$

$$d^{(s)}_{(2,0)} = -(n-2)(n+4), \tag{3.37c}$$

$$2d^{(s)}_{(2,\frac{1}{2})} = \sqrt{2}n(n+2), \tag{3.37d}$$

$$3d^{(s)}_{(3,0)} = -2n^2(n-2)(n+4)(n^2+n-8), \tag{3.37e}$$

$$3d^{(s)}_{(3,\frac{1}{3})} = \sqrt{3}(n-1)^2(n+1)(n+2)(n^2-7), \tag{3.37f}$$

$$3d^{(s)}_{(3,\frac{2}{3})} = -(n-4)(n-1)(n+1)^2(n^2-3). \tag{3.37g}$$

$$d^{(t)}_{(\frac{1}{2},0)} = 1, \tag{3.37h}$$

$$d^{(t)}_{(\frac{3}{2},0)} = 0, \tag{3.37i}$$

$$d^{(t)}_{(\frac{3}{2},\frac{2}{3})} = 2, \tag{3.37j}$$

$$5d^{(t)}_{(\frac{5}{2},0)} = -2(n+2)^2(n-1)^2, \tag{3.37k}$$

$$5d^{(t)}_{(\frac{5}{2},\frac{2}{5})} = 2[n^4-2n^2-6n+5]-2\varphi[n^3+2n^2-4n+2], \tag{3.37l}$$

$$5d^{(t)}_{(\frac{5}{2},\frac{4}{5})} = 2[-n^4+n^3+4n^2+2n-3]-2\varphi[n^3+2n^2-4n+2]. \tag{3.37m}$$

$$3d^{(u)}_{(\frac{3}{2},0)} = -2(n+2), \tag{3.37n}$$

$$3d^{(u)}_{(\frac{3}{2},\pm\frac{2}{3})} = -2(n-1), \tag{3.37o}$$

$$5d^{(u)}_{(\frac{5}{2},0)} = 2(n+2)^2(n-1)^2, \tag{3.37p}$$

$$5d^{(u)}_{(\frac{5}{2},\frac{2}{5})} = -2[n^4+n^3-12n^2-3n+9]-2\varphi[n^4-n^3-5n^2+4n+14], \tag{3.37q}$$

$$5d^{(u)}_{(\frac{5}{2},\frac{4}{5})} = -2[2n^4-17n^2+n+23]+2\varphi[n^4-n^3-5n^2+4n+14]. \tag{3.37r}$$

**Case of** $\left\langle V_{(1,0)}V_{(1,0)}V_{(1,0)}V_{(1,0)}\right\rangle$ **and** $\left\langle V_{(1,1)}V_{(1,1)}V_{(1,1)}V_{(1,1)}\right\rangle$ **with** $\sigma = (0,2,2)$

By reflection symmetry, these two 4-point functions have the same normalized structure constants.

| Parity | $d^{(x)}_{(r,s)} = d^{(x)}_{(r,-s)}$ | | (3.38) |
|---|---|---|---|
| Permutation | $d^{(s)}_{(r,s)} = (-)^{rs}d^{(s)}_{(r,s)},\ d^{(t)}_{(r,s)} = (-)^{rs}d^{(u)}_{(r,s)}$ | | |
| Shift | $d^{(x)}_{(r,s)} = d^{(x)}_{(r,s+2)}$ | | |

Vanishing structure constants: $d^{(s)}_{\text{odd-spin}} = 0$.

$$d^{(s)}_{\text{diag}} = 1 \,, \tag{3.39a}$$

$$2d^{(s)}_{(4,0)} = n^2 \left(n^2 - 4\right)^3 \left(n^2 - 1\right)^2 \,, \tag{3.39b}$$

$$2d^{(s)}_{(4,\frac{1}{2})} = -n^4 (n^2 - 3)^2 (n^2 - 2)^2 \,, \tag{3.39c}$$

$$2d^{(s)}_{(4,1)} = n^2 \left(n^2 - 4\right)^3 \left(n^2 - 1\right)^2 \,. \tag{3.39d}$$

$$2d^{(t)}_{(2,0)} = n^2 - 4 \,, \tag{3.39e}$$

$$2d^{(t)}_{(2,\frac{1}{2})} = -n^2 \,, \tag{3.39f}$$

$$2d^{(t)}_{(2,1)} = n^2 - 4 \,, \tag{3.39g}$$

$$3d^{(t)}_{(3,0)} = n^2 (n^2 - 4)^2 w_s - 8n^3 (n^2 - 4) \,, \tag{3.39h}$$

$$3d^{(t)}_{(3,\frac{1}{3})} = -(n^2 - 1)^2 (n^2 - 3) w_s + 6n(n^2 - 1)^2 \,, \tag{3.39i}$$

$$3d^{(t)}_{(3,\frac{2}{3})} = (n^2 - 1)^2 (n^2 - 3) w_s - 2n(n^2 - 1)(n^2 - 3) \,, \tag{3.39j}$$

$$3d^{(t)}_{(3,1)} = -n^2 (n^2 - 4)^2 w_s \,. \tag{3.39k}$$

**Case of** $\left\langle V_{(1,0)} V_{(1,0)} V_{(1,0)} V_{(1,0)} \right\rangle$ **and** $\left\langle V_{(1,1)} V_{(1,1)} V_{(1,1)} V_{(1,1)} \right\rangle$ **with** $\sigma = (2,1,1)$

The symmetries are the same as with $\sigma = (0,2,2)$, leading again to $d^{(s)}_{\text{odd-spin}} = 0$.

$$d^{(s)}_{(2,0)} = n^2 - 4 \,, \tag{3.40a}$$

$$d^{(s)}_{(2,1)} = -(n^2 - 4) \,, \tag{3.40b}$$

$$3d^{(s)}_{(3,0)} = 32n^2 (n^2 - 4) \,, \tag{3.40c}$$

$$3d^{(s)}_{(3,\frac{2}{3})} = -4(n^2 - 1)(n^2 - 3) \,, \tag{3.40d}$$

$$d^{(s)}_{(4,0)} = (n-2)^3 (n+1)^2 (n+2) \left[ n^6 + 2n^5 - 2n^4 - 8n^3 + 9n^2 - 4n + 4 \right] \,, \tag{3.40e}$$

$$d^{(s)}_{(4,\frac{1}{2})} = 2n^5 (n^2 - 3)(n^2 - 2) \,, \tag{3.40f}$$

$$d^{(s)}_{(4,1)} = -(n-2)(n-1)^2 (n+2)^3 \left[ n^6 - 2n^5 - 2n^4 + 8n^3 + 9n^2 + 4n + 4 \right] \,. \tag{3.40g}$$

$$d^{(t)}_{(1,0)} = 1\,, \tag{3.40h}$$

$$d^{(t)}_{(1,1)} = 1\,, \tag{3.40i}$$

$$d^{(t)}_{(2,0)} = -(n-2)\,, \tag{3.40j}$$

$$d^{(t)}_{(2,\frac{1}{2})} = -n\,, \tag{3.40k}$$

$$d^{(t)}_{(2,1)} = -(n+2)\,, \tag{3.40l}$$

$$3d^{(t)}_{(3,0)} = 2n^2(n^4 - 6n^2 + 32)\,, \tag{3.40m}$$

$$3d^{(t)}_{(3,\frac{1}{3})} = (n^2 - 1)(n^4 - n^2 + 18)\,, \tag{3.40n}$$

$$3d^{(t)}_{(3,\frac{2}{3})} = -(n^2 - 1)(n^2 - 2)(n^2 - 3)\,, \tag{3.40o}$$

$$3d^{(t)}_{(3,1)} = -2n^2(n^2 - 4)^2\,. \tag{3.40p}$$

**Case of** $\left\langle V_{(\frac{3}{2},0)} V_{(1,1)} V_{(1,0)} V_{(\frac{1}{2},0)} \right\rangle$ **with** $\sigma = (\frac{3}{2}, 2, \frac{1}{2})$ 

| Parity | $d^{(s)}_{(r,s)} = (-)^{\delta_{r\neq\frac{3}{2}}} d^{(s)}_{(r,-s)},\ d^{(t)}_{(r,s)} = (-)^{\delta_{r\geq 2}} d^{(t)}_{(r,s)},\ d^{(u)}_{(r,s)} = (-)^{\delta_{r\geq\frac{3}{2}}} d^{(u)}_{(r,-s)}$ |
|---|---|
| Shift | $d^{(s,u)}_{(r,s)} = (-)^{\delta_{r\geq\frac{5}{2}}} d^{(s,u)}_{(r,s+2)},\ d^{(t)}_{(r,s)} = -d^{(t)}_{(r,s+2)}$ |

$$\tag{3.41}$$

The parity equations are obtained with the help of the shift equation for $V_{(1,1)}$. This equation suffers from an overall ambiguity from the relative normalizations of the solutions for the two 4-point functions $\left\langle V_{(\frac{3}{2},0)} V_{(1,\pm1)} V_{(1,0)} V_{(\frac{1}{2},0)} \right\rangle$. This ambiguity is lifted by the observation that $d^{(u)}_{(\frac{1}{2},0)} \neq 0$ in our numerical solution. Then parity implies the vanishings $d^{(s)}_{(\frac{1}{2},0)} = d^{(u)}_{(\frac{3}{2},0)} = d^{(s,u)}_{(r\geq\frac{5}{2},0)} = d^{(t)}_{(r\geq 2,0)} = 0$.

$$3d^{(s)}_{(\frac{3}{2},0)} = n+2\,, \tag{3.42a}$$

$$3d^{(s)}_{(\frac{3}{2},\frac{2}{3})} = -2(n-1)\,, \tag{3.42b}$$

$$5d^{(s)}_{(\frac{5}{2},\frac{2}{5})} = 4\cos\left(\tfrac{\pi}{10}\right)\left[-n^4 + 4n^2 - 2 + \varphi(n+1)(n^2-3)\right]\,, \tag{3.42c}$$

$$5d^{(s)}_{(\frac{5}{2},\frac{4}{5})} = 4\cos\left(\tfrac{3\pi}{10}\right)\left[n^4 - 4n^2 + 2 + \varphi^{-1}(n+1)(n^2-3)\right]\,. \tag{3.42d}$$

$$2d^{(t)}_{(2,\frac{1}{2})} = \sqrt{2}n^2\,, \tag{3.42e}$$

$$d^{(t)}_{(2,1)} = n^2 - 4\,, \tag{3.42f}$$

$$3d^{(t)}_{(3,\frac{1}{3})} = (n-4)(n-1)(n+1)^2(n^2-3)\,, \tag{3.42g}$$

$$3d^{(t)}_{(3,\frac{2}{3})} = \sqrt{3}(n^2-1)^2(n+2)(n-3)\,, \tag{3.42h}$$

$$3d^{(t)}_{(3,1)} = 2n^2(n^2-4)^2\,. \tag{3.42i}$$

$$d^{(u)}_{(\frac{1}{2},0)} = 1 \,, \tag{3.42j}$$

$$3d^{(u)}_{(\frac{3}{2},\frac{2}{3})} = -2\sqrt{3} \,, \tag{3.42k}$$

$$5d^{(u)}_{(\frac{5}{2},\frac{2}{5})} = 4\cos\left(\tfrac{\pi}{10}\right)\left[n^2 - 2 + \varphi\right] \,, \tag{3.42l}$$

$$5d^{(s)}_{(\frac{5}{2},\frac{4}{5})} = 4\cos\left(\tfrac{3\pi}{10}\right)\left[-n^2 + 2 + \varphi^{-1}\right] \,. \tag{3.42m}$$

**Case of** $\left\langle V_{(\frac{3}{2},\frac{2}{3})}V_{(1,1)}V_{(1,0)}V_{(\frac{1}{2},0)}\right\rangle$ **with** $\sigma = (\frac{3}{2},2,\frac{1}{2})$ 

In this case we have the same shift equations as in Table (3.41), but no parity symmetry due to the presence of the field $V_{(\frac{3}{2},\frac{2}{3})}$. We observe the vanishings $d^{(u)}_{(\frac{3}{2},0)} = d^{(u)}_{(\frac{5}{2},0)} = 0$. This suggests that the structure constants of the type $d^{(u)}_{(r\geq\frac{3}{2},0)}$ vanish, even though there is no symmetry that requires it.

$$3d^{(s)}_{(\frac{3}{2},0)} = (n+1) \,, \tag{3.43a}$$

$$3d^{(s)}_{(\frac{3}{2},\frac{2}{3})} = -2(n-1) \,, \tag{3.43b}$$

$$3d^{(s)}_{(\frac{3}{2},-\frac{2}{3})} = -2(n-1) \,, \tag{3.43c}$$

$$5d^{(s)}_{(\frac{5}{2},0)} = \sqrt{3}(n-1)(n+2)(n^2+3n-2) \,. \tag{3.43d}$$

$$d^{(t)}_{(2,0)} = \cos\left(\tfrac{\pi}{6}\right)(n^2-4) \,, \tag{3.43e}$$

$$d^{(t)}_{(2,\frac{1}{2})} = \cos\left(\tfrac{5\pi}{12}\right)n^2 \,, \tag{3.43f}$$

$$d^{(t)}_{(2,-\frac{1}{2})} = \cos\left(\tfrac{\pi}{12}\right)n^2 \,, \tag{3.43g}$$

$$2d^{(t)}_{(2,1)} = -(n^2-4) \,, \tag{3.43h}$$

$$3d^{(t)}_{(3,0)} = \sqrt{3}n^2(n^2-4) \,, \tag{3.43i}$$

$$3d^{(t)}_{(3,\frac{1}{3})} = (n^2-3)(n+4)(n+1)(n-1) \,, \tag{3.43j}$$

$$3d^{(t)}_{(3,-\frac{1}{3})} = 2(n^2-3)(n^2-1)^2 \,, \tag{3.43k}$$

$$3d^{(t)}_{(3,\frac{2}{3})} = 2\sqrt{3}n(n^2-1)^2 \,, \tag{3.43l}$$

$$3d^{(t)}_{(3,-\frac{2}{3})} = \sqrt{3}(n^2-1)^2(n+3)(n-2) \,, \tag{3.43m}$$

$$3d^{(t)}_{(3,1)} = -n^2(n^2-16)(n^2-4) \,. \tag{3.43n}$$

$$d^{(u)}_{(\frac{1}{2},0)} = 1 \,, \tag{3.43o}$$

$$d^{(u)}_{(\frac{3}{2},0)} = 0 \,, \tag{3.43p}$$

$$3d^{(u)}_{(\frac{3}{2},\frac{2}{3})} = -4\sqrt{3} \,, \tag{3.43q}$$

$$3d^{(u)}_{(\frac{3}{2},-\frac{2}{3})} = -2\sqrt{3} \,, \tag{3.43r}$$

$$d^{(u)}_{(\frac{5}{2},0)} = 0 \,, \tag{3.43s}$$

$$5d^{(u)}_{(\frac{5}{2},\frac{2}{5})} = 4\cos\left(\tfrac{\pi}{10}\right)\left(n + \varphi^{-1}\right)\left(n - 2\cos\left(\tfrac{4\pi}{15}\right)\right) \,, \tag{3.43t}$$

$$5d^{(u)}_{(\frac{5}{2},-\frac{2}{5})} = 4\cos\left(\tfrac{\pi}{10}\right)\left(n + \varphi^{-1}\right)\left(-n + 2\cos\left(\tfrac{14\pi}{15}\right)\right) \,, \tag{3.43u}$$

$$5d^{(u)}_{(\frac{5}{2},\frac{4}{5})} = 4\varphi^{-1}\cos\left(\tfrac{\pi}{10}\right)(n - \varphi)\left(-n + 2\cos\left(\tfrac{2\pi}{15}\right)\right) \,, \tag{3.43v}$$

$$5d^{(u)}_{(\frac{5}{2},-\frac{4}{5})} = 4\varphi^{-1}\cos\left(\tfrac{\pi}{10}\right)(n - \varphi)\left(n - 2\cos\left(\tfrac{8\pi}{15}\right)\right) \,. \tag{3.43w}$$

## 3.5 Some even-spin and odd-spin solutions

Let us consider the 4-point functions $\left\langle V_{(\frac{3}{2},0)} V^3_{(\frac{1}{2},0)} \right\rangle$ and $\left\langle V_{(\frac{3}{2},\frac{2}{3})} V^3_{(\frac{1}{2},0)} \right\rangle$, where we do not allow any diagonal field to propagate. In both cases, the space of solutions of crossing symmetry is two-dimensional, and the solutions correspond to the combinatorial maps [29]

$$\text{} \,, \quad \text{and} \quad \text{} \,. \tag{3.44}$$

Both maps have the same signature $\sigma = (1, 1, 1)$, and we do not know how to characterize the corresponding solutions by imposing constraints on the structure constants. We will focus on another basis of solutions, defined by their behaviour under permuting the third and fourth fields. In other words, we will consider $s$-channel odd-spin and even-spin solutions, as defined in Section 2.4. It will turn out that the normalized structure constants for these solutions are polynomial.

**Case of $\left\langle V_{(\frac{3}{2},0)} V_{(\frac{1}{2},0)} V_{(\frac{1}{2},0)} V_{(\frac{1}{2},0)} \right\rangle$: even-spin solution**

This solution turns out to be even-spin in all channels. We deduce the permutation symmetries of structure constants from Table (2.71).

| Parity | $d^{(x)}_{(r,s)} = d^{(x)}_{(r,-s)}$ |
|---|---|
| Permutation | $d^{(s)}_{(r,s)} = (-)^{rs} d^{(s)}_{(r,s)} = d^{(t)}_{(r,s)} = d^{(u)}_{(r,s)}$ |
| Shift | $d^{(x)}_{(r,s)} = (-)^{\delta_{r=1}} d^{(x)}_{(r,s+2)}$ |

$$\tag{3.45}$$

$$d^{(s)}_{(1,0)} = 1 \,, \tag{3.46a}$$

$$2d^{(s)}_{(2,0)} = (n-2)^2 \,, \tag{3.46b}$$

$$2d^{(s)}_{(2,1)} = -(n^2-4) \,, \tag{3.46c}$$

$$3d^{(s)}_{(3,0)} = 2n^2(n-2)^2(n^2+2n+3) \,, \tag{3.46d}$$

$$3d^{(s)}_{(3,\frac{2}{3})} = -n(n-2)(n^2-1)(n^2-3) \,. \tag{3.46e}$$

**Case of** $\left\langle V_{(\frac{3}{2},0)} V_{(\frac{1}{2},0)} V_{(\frac{1}{2},0)} V_{(\frac{1}{2},0)} \right\rangle$**: odd-spin solution**

This solution turns out to be odd-spin in all channels. It also turns out to be parity-odd. Combined with the shift equations, this implies the vanishings $d^{(x)}_{(r\geq2,1)} = 0$.

| Parity | $d^{(x)}_{(r,s)} = -d^{(x)}_{(r,-s)}$ |
|---|---|
| Permutation | $d^{(s)}_{(r,s)} = -(-)^{rs} d^{(s)}_{(r,s)} = -d^{(t)}_{(r,s)} = -d^{(u)}_{(r,s)}$ |
| Shift | $d^{(x)}_{(r,s)} = (-)^{\delta_{r=1}} d^{(x)}_{(r,s+2)}$ |

$$\tag{3.47}$$

$$d^{(s)}_{(1,1)} = 1 \,, \tag{3.48a}$$

$$2d^{(s)}_{(2,\frac{1}{2})} = -n(n+2) \,, \tag{3.48b}$$

$$3d^{(s)}_{(3,\frac{1}{3})} = \sqrt{3}(n-1)^2(n+1)(n+2)\left(n^2+n-4\right) \,. \tag{3.48c}$$

**Case of** $\left\langle V_{(\frac{3}{2},\frac{2}{3})} V_{(\frac{1}{2},0)} V_{(\frac{1}{2},0)} V_{(\frac{1}{2},0)} \right\rangle$**: even-spin solution**

This solution is even-spin in the $s$-channel only.

| Permutation | $d^{(s)}_{(r,s)} = (-)^{rs} d^{(s)}_{(r,s)} \,, \ d^{(t)}_{(r,s)} = (-)^{rs} d^{(u)}_{(r,s)}$ |
|---|---|
| Shift | $d^{(x)}_{(r,s)} = (-)^{\delta_{r=1}} d^{(x)}_{(r,s+2)}$ |

$$\tag{3.49}$$

$$d^{(s)}_{(1,0)} = 2 \,, \tag{3.50a}$$

$$2d^{(s)}_{(2,0)} = -2(n-2)(n+4) \,, \tag{3.50b}$$

$$2d^{(s)}_{(2,1)} = 2(n^2-4) \,, \tag{3.50c}$$

$$3d^{(s)}_{(3,0)} = 2n^2(n-2)\left[n^3+3n^2-n-9\right] \,, \tag{3.50d}$$

$$3d^{(s}_{(3,\frac{2}{3})} = 2(n^2-1)(n^2-3)\left[n^2-2n+3\right] \,, \tag{3.50e}$$

$$3d^{(s}_{(3,-\frac{2}{3})} = -2(n^2-3)(n^2-1)(n+1)(2n-3) \,. \tag{3.50f}$$

We observe the relations

$$rs \in 2\mathbb{Z} \implies d^{(s)}_{(r,s)} = 2d^{(t)}_{(r,s)} = 2d^{(u)}_{(r,s)} \,, \tag{3.50g}$$

so we only need to write the odd-spin $t$-channel structure constants,

$$d^{(t)}_{(1,1)} = -\sqrt{3}, \tag{3.50h}$$

$$2d^{(t)}_{(2,\frac{1}{2})} = -\sqrt{3}n(n-1-\sqrt{3}), \tag{3.50i}$$

$$2d^{(t)}_{(2,-\frac{1}{2})} = \sqrt{3}n(n-1+\sqrt{3}), \tag{3.50j}$$

$$d^{(t)}_{(3,\frac{1}{3})} = (n-1)(n^2-1)(n^2-4n-1), \tag{3.50k}$$

$$d^{(t)}_{(3,-\frac{1}{3})} = (n^2-1)^2(n^2-3), \tag{3.50l}$$

$$d^{(t)}_{(3,1)} = -(n+2)(n^2-4)(n^3-n^2-n-1). \tag{3.50m}$$

**Case of $\left\langle V_{(\frac{3}{2},\frac{2}{3})} V_{(\frac{1}{2},0)} V_{(\frac{1}{2},0)} V_{(\frac{1}{2},0)} \right\rangle$: odd-spin solution**

This solution is odd-spin in the $s$-channel only.

| Permutation | $d^{(s)}_{(r,s)} = -(-)^{rs}d^{(s)}_{(r,s)},\ d^{(t)}_{(r,s)} = -(-)^{rs}d^{(u)}_{(r,s)}$ |
|---|---|
| Shift | $d^{(x)}_{(r,s)} = (-)^{\delta_{r=1}}d^{(x)}_{(r,s+2)}$ |

$$\tag{3.51}$$

$$d^{(s)}_{(1,1)} = 2, \tag{3.52a}$$

$$d^{(s)}_{(2,\frac{1}{2})} = n(n-1-\sqrt{3}), \tag{3.52b}$$

$$d^{(s)}_{(2,-\frac{1}{2})} = -n(n-1+\sqrt{3}), \tag{3.52c}$$

$$3d^{(s)}_{(3,\frac{1}{3})} = -2\sqrt{3}(n-1)(n^2-1)(n^2-4n-1), \tag{3.52d}$$

$$3d^{(s)}_{(3,-\frac{1}{3})} = -2\sqrt{3}(n^2-1)^2(n^2-3), \tag{3.52e}$$

$$3d^{(s)}_{(3,1)} = 2\sqrt{3}(n+2)(n^2-4)(n^3-n^2-n-1). \tag{3.52f}$$

We observe the relations

$$rs \in 2\mathbb{Z}+1 \implies d^{(s)}_{(r,s)} = -2d^{(t)}_{(r,s)} = -2d^{(u)}_{(r,s)}, \tag{3.52g}$$

so we only need to write the $t$-channel even-spin structure constants,

$$d^{(t)}_{(1,0)} = -\sqrt{3}, \tag{3.52h}$$

$$2d^{(t)}_{(2,0)} = \sqrt{3}(n-2)(n+4), \tag{3.52i}$$

$$2d^{(t)}_{(2,1)} = -\sqrt{3}(n^2-4), \tag{3.52j}$$

$$3d^{(t)}_{(3,0)} = -\sqrt{3}n^2(n-2)\left[n^3+3n^2-n-9\right], \tag{3.52k}$$

$$3d^{(t)}_{(3,\frac{2}{3})} = -\sqrt{3}(n^2-1)(n^2-3)\left[n^2-2n+3\right], \tag{3.52l}$$

$$3d^{(t)}_{(3,-\frac{2}{3})} = \sqrt{3}(n^2-3)(n^2-1)(n+1)(2n-3). \tag{3.52m}$$

Finally, notice that the odd-spin and even-spin solutions for $\left\langle V_{(\frac{3}{2},\frac{2}{3})} V_{(\frac{1}{2},0)} V_{(\frac{1}{2},0)} V_{(\frac{1}{2},0)} \right\rangle$ have a simple relation to one another:

$$rs \in 2\mathbb{Z} \implies \sqrt{3}d^{(s)}_{(r,s)}\Big|_{\text{even-spin}} = -2d^{(t)}_{(r,s)}\Big|_{\text{odd-spin}} = -2d^{(u)}_{(r,s)}\Big|_{\text{odd-spin}}, \tag{3.53}$$

$$rs \in 2\mathbb{Z}+1 \implies \sqrt{3}d^{(s)}_{(r,s)}\Big|_{\text{odd-spin}} = -2d^{(t)}_{(r,s)}\Big|_{\text{even-spin}} = -2d^{(u)}_{(r,s)}\Big|_{\text{even-spin}}. \tag{3.54}$$

Thanks to this relation, it is easy to write solutions that are even-spin or odd-spin the the $t$-channel or in the $u$-channel, by taking simple linear combinations of our two solutions.

# 4 Lattice models

## 4.1 The two lattice models

We will now introduce two lattice models:

- The Nienhuis loop model, which we already sketched in Section 1.4. The model lives on a honeycomb lattice, and has 4 possible configurations (1.21) at each trivalent vertex. We call its partition function $Z^{\mathrm{loop}}$.

- The Fortuin–Kasteleyn cluster model, which lives on a square lattice, and has 2 possible configurations $\rangle\langle$ $\smile\frown$ at each tetravalent vertex. We call its partition function $Z^{\mathrm{cluster}}$.

Both models come with a parameter $n$, which is the weight of loops. If $n \in \mathbb{N}$, the partition function of the loop model coincides with that of the $O(n)$ model, which has a global symmetry group $O(n)$. In fact it is possible to make sense of the $O(n)$ symmetry even if $n$ is not integer. However, we will consider correlation functions that involve loops with arbitrary weights, and this breaks the $O(n)$ symmetry in general. We therefore avoid referring to the loop model as an $O(n)$ model. Similarly, if $Q = n^2 \in \mathbb{N}$, the partition function of the cluster model coincides with that of the $Q$-state Potts model, with its global symmetry group $S_Q$. Again, introducing loops with arbitrary weights breaks that symmetry.

**Loop model**

The loop model is most conveniently defined on a honeycomb lattice $\mathcal{L}$ that is obtained from the square lattice $\mathcal{S}$ by deforming each of its vertices into a pair of degree-three vertices separated by a new horizontal edge:

$$ \tag{4.1} $$

Each configuration $\mathcal{C}$ is a set of loops, obtained by occupying some of the edges of $\mathcal{L}$, subject to the constraint that each vertex be adjacent to 0 or 2 occupied edges: the curves do not end or bifurcate. The partition function is defined by attributing a local weight $K$ to each occupied edge and a non-local weight $n$ to each loop:

$$ Z^{\mathrm{loop}}(K, n) = \sum_{\mathcal{C}} K^{|\mathcal{C}|} n^{\ell(\mathcal{C})} , \tag{4.2} $$

where $|\mathcal{C}|$ is the number of occupied edges and $\ell(\mathcal{C})$ is the number of loops.

Let us build diagonal correlation functions. We give a lattice implementation of the general idea from [25]. We mark $N$ distinct faces of our lattice $\{\mathbf{x}_1, \ldots, \mathbf{x}_N\}$. Each loop partitions these faces into two (possibly empty) subsets, and we allow the loop's weight to depend on this partition. In the case $N = 4$, there are 8 possible partitions, and we write the corresponding

weights as follows:

$$
\begin{array}{cc|cc}
w_1 & (1)(234) & w_s & (12)(34) \\
w_2 & (2)(134) & w_u & (13)(24) \\
w_3 & (3)(124) & w_t & (14)(23) \\
w_4 & (4)(123) & n & (1234)
\end{array}
\tag{4.3}
$$

The diagonal 4-point function of the lattice loop model then reads

$$
C^{\mathrm{loop}}(\mathbf{x}_i|K,n,w_i,w_x) = \frac{1}{Z^{\mathrm{loop}}(K,n)} \sum_{\mathcal{C}} K^{|\mathcal{C}|} n^{\ell_0(\mathcal{C})} \prod_{i=1,2,3,4} w_i^{\ell_i(\mathcal{C})} \prod_{x=s,t,u} w_x^{\ell_x(\mathcal{C})}.
\tag{4.4}
$$

Here $\ell_0(A)$ is the number of contractible loops, while $\ell_i(A)$ and $\ell_x(A)$ denote the numbers of various types of non-contractible loops, with $i = 1,2,3,4$ and $x = s,t,u$.

Since loops are non-intersecting, at most one of the three integers $\ell_s, \ell_t, \ell_u$ can be non-vanishing, in other words $\ell_s \ell_t + \ell_s \ell_u + \ell_t \ell_u = 0$. This leads to the decomposition

$$
C^{\mathrm{loop}}(w_s, w_t, w_u) = C^{(0)} + C^{(s)}(w_s) + C^{(t)}(w_t) + C^{(u)}(w_u),
\tag{4.5}
$$

where

$$
C^{(0)} = C^{\mathrm{loop}}(0,0,0),
\tag{4.6a}
$$

$$
C^{(s)}(w_s) = C^{\mathrm{loop}}(w_s,0,0) - C^{\mathrm{loop}}(0,0,0),
\tag{4.6b}
$$

$$
C^{(t)}(w_t) = C^{\mathrm{loop}}(0,w_t,0) - C^{\mathrm{loop}}(0,0,0),
\tag{4.6c}
$$

$$
C^{(u)}(w_u) = C^{\mathrm{loop}}(0,0,w_u) - C^{\mathrm{loop}}(0,0,0).
\tag{4.6d}
$$

**Cluster model**

The cluster model is in fact a model of completely packed loops, which we call cluster model to distinguish it from the previously considered loop model. The loops live on a diagonally oriented square lattice $\mathcal{S}$. Here is a sample configuration:

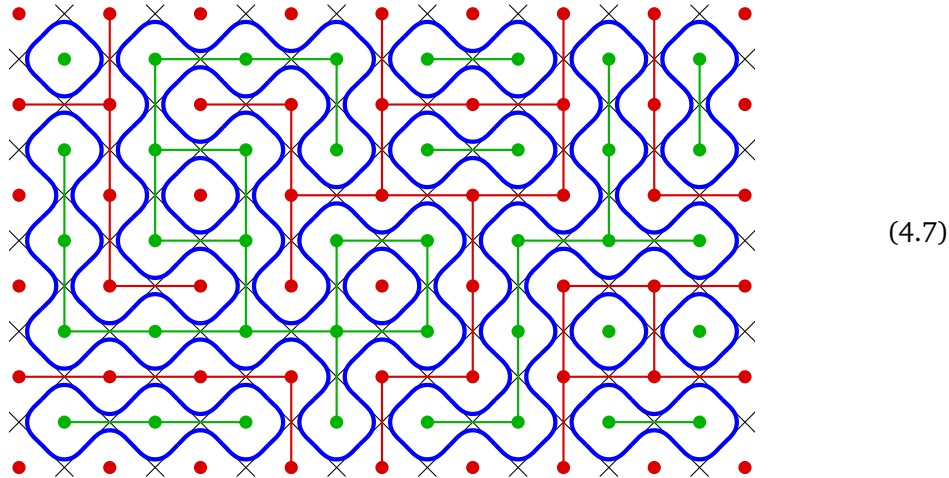

$$
\tag{4.7}
$$

Loops are shown as blue curves that jointly cover all the edges of $\mathcal{S}$ and split in one of two ways at its vertices. Each configuration of loops on $\mathcal{S}$ is bijectively related to a set of clusters (in red) on an axially oriented square lattice $\mathcal{L}$, and to a corresponding set of dual clusters (in green) on a shifted axially oriented square lattice $\mathcal{L}^*$. In other words, $\mathcal{L}$ and $\mathcal{L}^*$ are mutually dual lattices, and $\mathcal{S}$ is their common medial lattice, with a vertex of $\mathcal{S}$ standing on each pair of intersecting edges of $\mathcal{L}$ and $\mathcal{L}^*$. Each loop on $\mathcal{S}$ then forms the boundary between a cluster

on $\mathcal{L}$ and a dual cluster on $\mathcal{L}^*$. The partition function $Z$ is defined by attributing a weight $\frac{v}{n}$ to each occupied (red) edge of $\mathcal{L}$ and a weight $n$ to each (blue) loop:

$$Z^{\text{cluster}}(v, n) = \sum_{A \subseteq E} \left(\frac{v}{n}\right)^{|A|} n^{\ell(A)}, \tag{4.8}$$

where the sum runs over subsets $A$ of the edges $E$ in $\mathcal{L}$, and $\ell(A)$ denotes the number of loops on $\mathcal{S}$ as determined by $A$. The critical value of the lattice weight $v$ is $v_c = n$. This corresponds to the ferromagnetic critical point of the Potts model.

Diagonal correlation functions are defined as in the loop model, with the points $\{\mathbf{x}_1, \ldots, \mathbf{x}_N\}$ now being nodes of the lattice $\mathcal{L}$. In the case of 4-point functions, the expression is

$$C^{\text{cluster}}(\mathbf{x}_i | v, n, w_i, w_x) = \frac{1}{Z^{\text{cluster}}(v, n)} \sum_{A \subseteq E} \left(\frac{v}{n}\right)^{|A|} n^{\ell_0(A)} \prod_{i=1,2,3,4} w_i^{\ell_i(A)} \prod_{x=s,t,u} w_x^{\ell_x(A)}. \tag{4.9}$$

Since the points $\{\mathbf{x}_1, \ldots, \mathbf{x}_4\}$ belong to clusters, and loops separate clusters from dual clusters, any two points are separated by an even number of loops. This leads to the following 6 parity constraints on loop numbers, 3 of which are independent:

$$\ell_{12tu}, \ell_{34tu}, \ell_{13st}, \ell_{24st}, \ell_{14su}, \ell_{23su} \in 2\mathbb{N}, \tag{4.10}$$

where $\ell_{abcd} = \ell_a + \ell_b + \ell_c + \ell_d$. Now, our numerical calculations will turn out to require the stronger constraint that all 7 loop numbers $\ell_i, \ell_x$ have a well-defined parity. In particular, without this constraint, ratios of amplitude would depend on lattice size. To illustrate this constraint, let us sketch two configurations near the points $\{\mathbf{x}_1, \mathbf{x}_2\}$, with loops in blue, clusters in red and dual clusters in white:

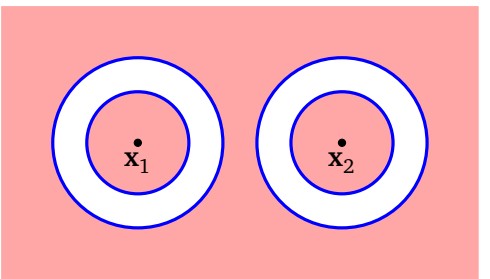 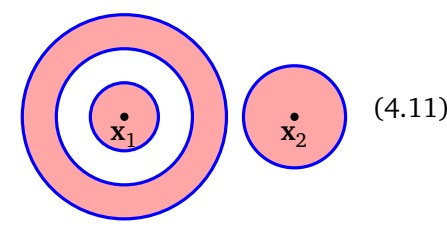 (4.11)

We have $(\ell_1, \ell_2) = (2, 2)$ on the left and $(3, 1)$ on the right. The common parity of $\ell_1$ and $\ell_2$ determines whether the smallest object that contains both points is a cluster (left) or a dual cluster (right). We want our correlation functions to distinguish these two cases.

Equivalently, our constraint means that the 4-point function must be an eigenvector of all 7 reflections $w_i \to -w_i, w_x \to -w_x$. We therefore consider the combinations

$$C^{v_1, v_s, v_t, v_u}(\mathbf{x}_i | v, n, w_i, w_x) = \frac{1}{16} \sum_{\epsilon_a \in \{+, -\}} \left[\prod_{a \in \{1, s, t, u\}} \epsilon_a^{v_a}\right] C^{\text{cluster}}(\mathbf{x}_i | v, n, \epsilon_i w_i, \epsilon_x w_x), \tag{4.12}$$

where $v_a \in \{0, 1\}$, and by convention $\epsilon_2 = \epsilon_3 = \epsilon_4 = 1$. This is the sum over lattice configurations such that $\forall a \in \{1, s, t, u\}, \ell_a \equiv v_a \mod 2$, with the parities of $\ell_2, \ell_3, \ell_4$ then determined by Eq. (4.10). Since loops do not intersect, only the 8 combinations such that $v_s + v_t + v_u \in \{0, 1\}$ are in fact non-vanishing. Then, if we decompose $C^{\text{cluster}}$ as in Eq. (4.5), we have

$$C^{v_1, v_s, v_t, v_u} = \delta_{v_s} \delta_{v_t} \delta_{v_u} C^{(0), v_1} + \delta_{v_t} \delta_{v_u} C^{(s), v_1, v_s} + \delta_{v_s} \delta_{v_u} C^{(t), v_1, v_t} + \delta_{v_t} \delta_{v_u} C^{(u), v_1, v_u}, \tag{4.13}$$

where $\delta_v$ is the Kronecker delta function, and in each term we only sum over the signs of the weights that do appear, for example $C^{(s), v_1, v_s} = \frac{1}{4} \sum_{\epsilon_1, \epsilon_s \in \{+, -\}} \epsilon_1^{v_1} \epsilon_s^{v_s} C^{(s)}(\epsilon_1 w_1, \epsilon_s w_s)$.

## 4.2 Transfer matrix formalism

**Construction of the transfer matrix**

For both lattice models we define a time slice as a horizontal line just below a row of horizontal edges in $\mathcal{L}$. If we take the width of $\mathcal{L}$ to be $L$ lattice spacings, the time slice intersects $2L$ loop strands. The transfer matrix $T$ constructs loop configurations from the bottom up, by adding one row together with its Boltzmann weights. We write the transfer matrix as

$$T = \left( \prod_{i=1}^{L} \check{R}_{2i,2i+1} \right) \times \left( \prod_{i=1}^{L} \check{R}_{2i-1,2i} \right), \tag{4.14}$$

where $\check{R}_{k,k+1}$ is an operator that propagates through a single vertex of $\mathcal{S}$ by acting on a pair of neighbouring loop strands $k$ and $k+1$. We apply periodic boundary conditions horizontally, meaning that the strands labelled $2L+1$ and $1$ are identified.

The explicit expression of the operator $\check{R}_{k,k+1}$ depends on the model. In the loop model, $\check{R}_{k,k+1}$ is obtained by gluing two vertices, which gives rise to 8 possible diagrams:

$$\check{R}_{k,k+1}^{\text{loop}} = \rangle\!-\!\langle + K\left[ \rangle\!\!\rangle\langle + \rangle\langle\!\!\langle \right] + K^2\left[ \underset{\text{cup}}{\cup} + \rangle\!\!\smile + \smile\!\!\langle + \rangle\!\langle + \underset{\text{cap}}{\cap} \right]. \tag{4.15}$$

In particular, we call a cap the concatenation of two loop strands, and a cup the creation of two connected loop strands. In the cluster model, the operator $\check{R}_{k,k+1}^{\text{cluster}}$ takes two forms, depending on the parity of $k$:

$$\check{R}_{2i,2i+1}^{\text{cluster}} = \frac{v}{n}\rangle\langle + \times, \qquad \check{R}_{2i-1,2i}^{\text{cluster}} = \rangle\langle + \frac{v}{n}\times. \tag{4.16}$$

In both forms, the second diagram is a cap with a cup on top.

**Space of states and the partition function**

Let us now define the space of states on which the transfer matrix acts. In the loop model, the space of states has a basis made of dilute defect-free link patterns over the set of sites $\{1, 2, \ldots, 2L\}$. Such a link pattern is a collection of $p$ arcs with $0 \leq p \leq L$, such that each arc connects two distinct sites, each site is connected by 0 or 1 arc, and arcs do not cross. Therefore, $2(L-p)$ sites are left empty. The link patterns can be depicted by arranging sites on a line (the time slice) and drawing a set of $p$ non-intersecting arcs in the half-space below that line. For example, here is a dilute link pattern with $p=4$ arcs in the case $L=6$:

$$\tag{4.17}$$

In order to define the action of the transfer matrix on dilute link patterns, we need only define the action of the operator $\check{R}_{k,k+1}^{\text{loop}}$ (4.15). This operator is written as a linear combination of diagrams, which act by diagram concatenation. Any diagram that does not respect the occupancy of the link pattern for the sites $k$ and $k+1$ acts as zero by definition, so there are always 2 out of 8 diagrams in (4.15) that contribute. In our example (4.17), the sites 2 and 3 are respectively occupied and empty, so that the diagrams 2 and 5 in (4.15) contribute to the action of $\check{R}_{2,3}^{\text{loop}}$. Similarly, the action of $\check{R}_{4,5}^{\text{loop}}$ reduces to the contributions of the diagrams 7 and 8. As a result, the loop model's transfer matrix is an element of the Motzkin algebra [35],

also known [36] as the dilute affine Temperley–Lieb algebra [37, 38]. The space of states is a standard module of that algebra.

In the case of the cluster model, the states are still defect-free link patterns over the set of sites $\{1, 2, \ldots, 2L\}$, but they are no longer dilute: there are always $L$ arcs, which connect all the sites pairwise. Here is an example with $L = 6$:

$$\tag{4.18}$$

As a result, the cluster model's transfer matrix is an element of the affine Temperley–Lieb algebra. The space of states is a standard module of that algebra.

For both models, the least trivial diagram of $\check{R}^{\text{loop}}_{k,k+1}$ or $\check{R}^{\text{cluster}}_{k,k+1}$ is the one that involves a cap, i.e. that contracts the sites $k$ and $k + 1$. If these two sites were already connected by the link pattern, connecting them again produces a closed loop, which is erased from the pattern and replaced with a factor of $n$. If the sites $k$ and $k + 1$ were not connected by the link pattern, we obtain situations where two arcs are concatenated, like in the following example:

$$\tag{4.19}$$

The link patterns that we have described allow us to compute the partition functions $Z^{\text{loop}}(K, n)$ (4.2) and $Z^{\text{cluster}}(v, n)$ (4.8). In particular, the local Boltzmann weights $K$ and $\frac{v}{n}$ come from the prefactors in the operators $\check{R}^{\text{loop}}_{k,k+1}$ (4.15) and $\check{R}^{\text{cluster}}_{k,k+1}$ (4.16), while the non-local loop weight $n$ arises whenever a closed loop is produced. We will now turn to computing correlation functions, which involve more variables and therefore require more complicated constructions.

**Diagonal 4-point functions**

For concreteness, let us focus on the lattice model's diagonal 4-point function (4.4) or (4.9). The weight of each closed loop depends on how it partitions the set $\{\mathbf{x}_1, \mathbf{x}_2, \mathbf{x}_3, \mathbf{x}_4\}$, according to Eq. (4.3). In order to take this information into account in the transfer matrix formalism, we introduce three seam lines $\Sigma_0, \Sigma_1, \Sigma_2$:

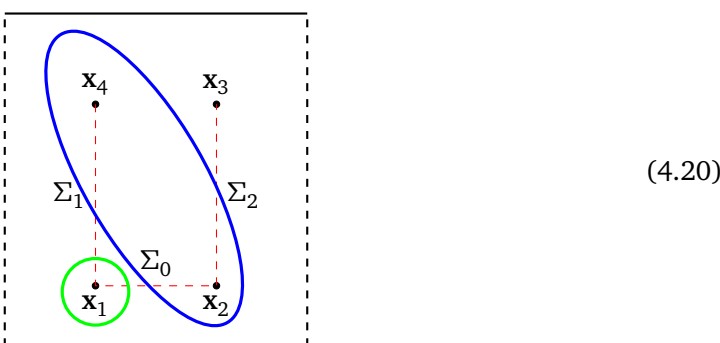

$$\tag{4.20}$$

Our picture is in the cylinder geometry with the left and right boundaries being identified. The seam lines are of combinatorial nature: they can be deformed at will, and can even cross one another, as long as their endpoints are unchanged. These lines are only accounting devices: in spite of this they were called defect lines by Gamsa and Cardy [39].

Given a closed loop, knowing how it partitions the set $\{\mathbf{x}_1, \mathbf{x}_2, \mathbf{x}_3, \mathbf{x}_4\}$ is equivalent to knowing its characteristic $(\sigma_0, \sigma_1, \sigma_2) \in \mathbb{Z}_2^3$, where $\sigma_i \in \mathbb{Z}_2$ is the number of times the loop crosses

the seam line $\Sigma_i$, modulo 2. For instance, in the picture, the blue loop has characteristic $(1,1,1)$ and is hence of type $(13)(24)$ with weight $w_t$, while the green loop has characteristic $(1,1,0)$ and is hence of type $(1)(234)$ with weight $w_1$.

For the transfer matrix to assign the correct loop weights, we associate a characteristic $(\sigma_0, \sigma_1, \sigma_2)$ to each arc. When the transfer matrix acts on the link pattern, we determine the characteristic of the resulting link pattern via local rules:

- When two arcs are concatenated by a cap, their characteristics add.

- When an arc is created by a cup, its initial characteristic is $(0,0,0)$.

- Each time the seam line crosses an arc, that arc's characteristic is modified as $\sigma_i \to \sigma_i + 1$.

- When an arc between $k$ and $k+1$ is turned into a loop by the cap, the corresponding loop weight (4.3) is inferred from the characteristics of the arc.

In particular, the operator $\check{R}^{\text{loop}}_{k,k+1}$ acts on the characteristics, whenever a seam line is present where it is applied. In the presence of the horizontal seam line $\Sigma_0$, the operator $\check{R}^{\text{loop}}_{k,k+1}$ becomes

$$\check{R}^{\text{loop}}_{k,k+1} = \text{⟩⟨} + K\left[\text{⟩⟨} + \text{⟩⟨}\right] + K^2\left[\text{⟩⟨} + \text{⟩⟨} + \text{⟩⟨} + \text{⟩⟨} + \text{⟩⟨}\right]. \qquad (4.21)$$

It might be of interest to formalize these characteristics in the representation theory of the affine (dilute) Temperley–Lieb algebra.

To be complete, we still have to specify the boundary conditions at the top and bottom of the cylinder. In the loop model, we take the top and bottom boundaries to consist of rows of empty sites. To achieve this in the transfer matrix formalism,

- we take the initial state to be the empty link pattern $\bullet\ \bullet\ \bullet\ \bullet\ \bullet\ \bullet\cdots$,

- in the last transfer matrix, $\check{R}^{\text{loop}}_{k,k+1}$ with $k$ even becomes $\check{R}^{\text{loop, last}}_{2i,2i+1} = \text{⟩⟨} + K^2\,\text{⌣}$.

In the cluster model, we take the top and bottom boundaries to reflect the loops, as in Figure (4.7). To achieve this in the transfer matrix formalism,

- we take the inital state to be the all-cups link pattern $\cup\ \cup\ \cup\cdots$,

- we build the last transfer matrix from the cap with a cup operator $\check{R}^{\text{cluster, last}}_{k,k+1} = \text{✕}$,

  i.e. one of the two terms of the operator $\check{R}^{\text{cluster}}_{k,k+1}$ (4.16).

In both models, after repeatedly acting with the transfer matrix on the initial state, we obtain a final state that is proportional to that initial state, and the value of the correlation function is the coefficient of proportionality.

**Non-diagonal 4-point functions**

We believe that it is possible to modify the transfer matrix formalism to compute the correlation function associated with any combinatorial map. We first sketch the general idea before turning to the only case that we will study in this article.

A combinatorial map that is associated to the 4-point function $\left\langle \prod_{i=1}^{4} V_{(r_i,s_i)} \right\rangle$ has four vertices of valencies $2r_i$. This map defines constraints on loop configurations: in particular, there must exist open loops that end at vertices, according to their valencies. A vertex of valency

zero corresponds to a diagonal field, which modifies the weights of surrounding loops. A vertex that corresponds to the non-diagonal field $V_{(r,s)}$ comes with a weight that depends on the angles of the loops that end there, see the diagram (1.23).

In the transfer matrix formalism, we have to add information about the open loops. In a link pattern, some sites may come with labels that indicate to which open loop they belong. When acting with the transfer matrix, this information must be carried to the resulting link pattern. Terms that would be inconsistent with the desired combinatorial map are set to zero using local rules.

Let us focus on the 4-point function $\left\langle V^4_{(\frac{1}{2},0)} \right\rangle$, in the context of the loop model. There are three possible combinatorial maps:

$$
\left. \begin{array}{c} \vert \ \vert \end{array} \ , \ \begin{array}{c} \overline{\phantom{xx}} \\ \underline{\phantom{xx}} \end{array} \ , \ \begin{array}{c} \diagup\!\diagdown \end{array} \right) . \tag{4.22}
$$

In the case of the first map, the open loops on the cylinder are as follows:

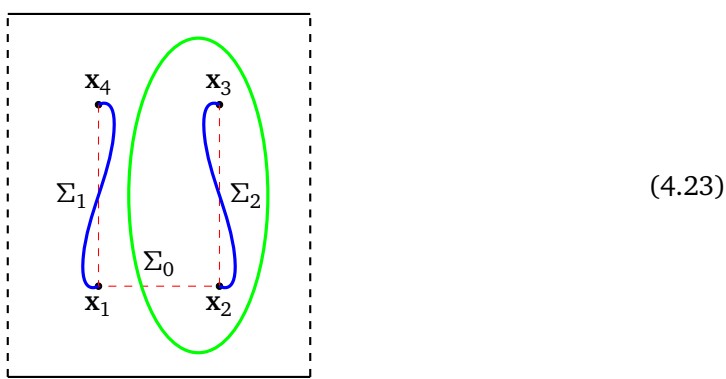 (4.23)

The existence of these two open loops forbids closed loops of type (12)(34) and (13)(24), while allowing closed loops of type (14)(23), such as the green loop in the figure. We still need the seam $\Sigma_0$ to distinguish such loops from contractible loops, but the seams $\Sigma_1$ and $\Sigma_2$ are no longer needed. Any link pattern that occurs after the two bottom vertices but before the two top vertices must include two labelled empty sites, with labels 1 and 2 that indicate which bottom vertex the site is connected to. For example:

$$
\begin{array}{c} \bullet \ \underbrace{\phantom{x} 1 \phantom{x}}_{} \ \bullet \ \underbrace{\phantom{x} 2 \phantom{x}}_{} \ \bullet \ \cup \ \bullet \end{array}
$$

(4.24)

The label $i$ appears when we insert an open loop that ends at $\mathbf{x}_i$. Our convention is to do this insertion using the operator

$$
\check{R}^{\text{loop}}_{k,k+1}(\mathbf{x}_i) = K^{\frac{1}{2}} \left. \begin{array}{c} \diagup\!\diagdown \\ {}_i \end{array} \right\rangle\!\!\langle + K^{\frac{3}{2}} \left. \begin{array}{c} \diagup\!\diagdown \\ {}_i \end{array} \right\rangle\!\!\langle . \tag{4.25}
$$

Then the labels can move when the transfer matrix is applied. For example, if we act with the cap term of $\check{R}_{3,4}$, the label 1 moves 4 sites to the right, and the arc that ends there is erased:

$$
\begin{array}{c} \bullet \ \widehat{\ } \ \bullet \ \underbrace{\phantom{x} 2 \phantom{x}}_{} \ \bullet \ \cup \ \bullet \end{array} \ = \ \begin{array}{c} \bullet \ \bullet \ \bullet \ \bullet \ \underbrace{\phantom{} 2 \ 1 \phantom{}}_{} \ \bullet \ \cup \ \bullet \end{array} . \tag{4.26}
$$

If the two marked sites are neighbours, acting with a cap that would connect them annihilates the state:

$$
\begin{array}{c} \bullet \ \bullet \ \bullet \ \bullet \ \widehat{2 \ 1} \ \bullet \ \cup \ \bullet \end{array} \ = \ 0 . \tag{4.27}
$$

A link pattern that occurs above the two top sites includes four labelled empty sites. It is now allowed to connect site 1 with site 4 and site 2 with site 3, in which case the two corresponding labels disappear. When reaching the top boundary, no labels must remain, so that we obtain the empty link pattern after having generated the desired combinatorial map.

**Tests of the formalism**

We have implemented transfer matrix algorithms for computing 4-point correlation functions, using sparse-matrix factorisation, hashing techniques and arbitrary-precision arithmetics, as described in [34]. We have performed a number of stringent tests to make sure that the implementations are correct:

- Comparison with exact enumeration results for small systems.

- Checks of the asymptotic decay of the correlation functions for large systems.

- Changing the transfer direction from vertically to horizontally, we find that the results do not change, provided free boundary conditions are imposed in both directions.

- Deforming the seams does not lead to different results.

- In the case of diagonal 4-point functions, comparing with results that are already known in special cases [26].

## 4.3 Amplitudes

Four-point functions are complicated quantities whose lattice computation leads to finite-size effects. A direct comparison with conformal field theory could therefore only succeed to good precision if the lattice was large. However, the size of the transfer matrix grows exponentially with the size of the lattice, so we only have access to small lattices.

Fortunately, it is possible to extract lattice observables that are much less sensitive to finite-size effects, called amplitudes. Four-point amplitudes depend on the choice of a channel $x \in \{s, t, u\}$. Roughly speaking, the decomposition of a lattice 4-point function into $x$-channel amplitudes corresponds to the $x$-channel decomposition of a conformal 4-point functions into structure constants and conformal blocks. We will therefore compare amplitudes with 4-point structure constants.

**Definition of 4-point amplitudes**

We consider a 4-point function on a finite cylindrical lattice, represented as a rectangle with the left and right sides identified. The size of the cylinder and the positions of the punctures $\mathbf{x}_1, \ldots, \mathbf{x}_4$ are determined by four lengths $L, \ell, M, d$, where the points form a rectangle of size $d \times \ell$ in a cylinder of size $L \times (2M + \ell)$. In the transfer matrix formalism, with time running upwards, the horizontal size $L$ must be small, since it determines the dimension of the space of states. However, the vertical size $2M + \ell$ can be large. We choose $M$ large enough for the results not to depend on it. Moreover, we choose $d = \lfloor \frac{L}{2} \rfloor$, which gives the richest possible

spectrum. Our results will therefore only depend on the two variables $L, \ell$.

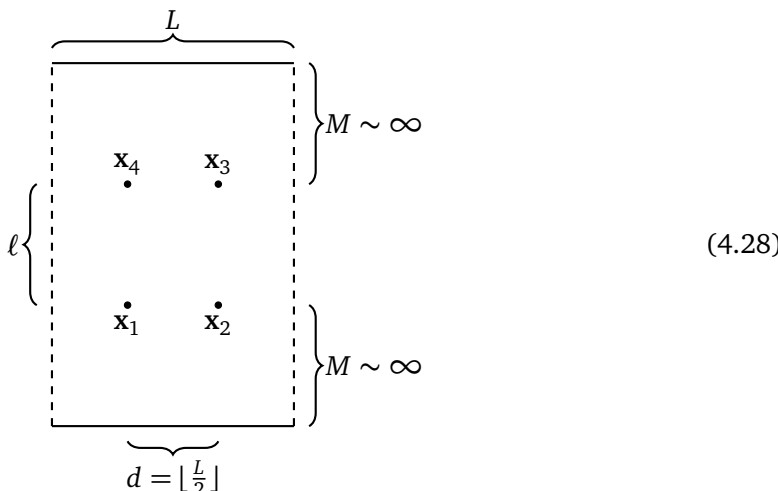

(4.28)

Now, the idea is to expand the 4-point function on the eigenvalues of a simplified transfer matrix $T_0$, which describes the propagation on the cylinder in the absence of the punctures $\mathbf{x}_1, \ldots, \mathbf{x}_4$. The simplified transfer matrix therefore depends on the weights of contractible loops $n$ and of $s$-channel loops $w_s$, but not on the other loop weights $w_1, \ldots, w_4, w_t, w_u$. Contractible loops and $s$-channel loops are the only loops whose numbers are extensive in the cylinder length $\ell$. We therefore write the expansion (1.24), where $\Lambda_\omega$ are the eigenvalues of $T_0$, including the largest eigenvalue $\Lambda_{\max}$, $S(L)$ is the $s$-channel spectrum, and $A_\omega$ are the $s$-channel amplitudes.

**Comparison with the CFT**

The eigenvalues $\Lambda_\omega$ of the simplified transfer matrix $T_0$ are plagued by finite-size effects, and can be compared with CFT quantities only at the critical coupling $K = K_c$, and in the limit $L \to \infty$. In this limit, they behave as [40]

$$-\log \Lambda_{\max}(L, K_c, n, w_s) \underset{L \to \infty}{=} f_\infty(n, w_s) L - \frac{\pi c_{\text{eff}}(n, w_s)}{6L} + o(L^{-1}), \qquad (4.29)$$

$$-\log \frac{\Lambda_\omega(L, K_c, n, w_s)}{\Lambda_{\max}(L, K_c, n, w_s)} \underset{L \to \infty}{=} \frac{2\pi \Delta_\omega(n, w_s)}{L} + o(L^{-1}), \qquad (4.30)$$

where $f_\infty$ is the free energy per unit area, $c_{\text{eff}}$ is the effective central charge, and $\Delta_\omega$ is an effective critical exponent.

On the other hand, it turns out that the spectrum $S(L)$ and amplitudes $A_\omega$ are much less dependent on the coupling $K$ and size $L$, and therefore better suited to a comparison with the CFT. In the case of the spectrum $S(L)$, let us relate the labels $\omega$ of the states to CFT parameters. In the case of the loop model, the simplified transfer matrix $T_0$ belongs to the dilute affine Temperley–Lieb algebra, whose representations are parametrized by the number of through-lines $2r \in \mathbb{N}$, and the pseudomomentum $s \in \frac{1}{r}\mathbb{Z}$ mod 2 [34, 36]. Since we have $2L$ loop strands on each time slice, the value of $r$ must obey

$$r \leq L. \qquad (4.31)$$

This truncation is the only $L$-dependence of the spectrum.

As our notations suggest, the numbers $r, s$ coincide with the Kac table indices of conformal field theory. In addition, the momentum that is associated to rotations around the cylinder (described by the cyclic group $\mathbb{Z}_L$) corresponds to the conformal spin $\Delta - \bar\Delta$ mod $L$. For simplicity, we will omit the conformal spin, and write states as $\omega = (r, s), \rho$, where $\rho$ labels states that share the same quantum numbers $r, s$.

**Computation of 4-point amplitudes**

The amplitude $A_\omega$ is associated to the eigenvalue $\Lambda_\omega$ of the simplified transfer matrix $T_0$, and we could deduce it from the corresponding left and right eigenvectors [34]. However, to do this, we would need to find local operators that create the punctures $\mathbf{x}_i$. This is not easy, and we will use an approach that does not require us to compute eigenvectors, at the price of having to use large values of the geometric parameters $\ell, M$: we compute the 4-point function $C^{\text{loop}}$ for as many values of $\ell$ as the size of the spectrum $|S(L)|$, and solve the resulting linear system for $A_\omega$.

In practice, we compute $C^{\text{loop}}$ to a precision of 2 000 digits for as many as $N_\ell = 500$ different values of $\ell$. This requires taking $M \sim 10^3 L$, a value that is chosen such that $\left(\frac{\Lambda_{\text{sub}}}{\Lambda_{\text{max}}}\right)^M \lessapprox 10^{-2000}$, where $\Lambda_{\text{sub}}$ is the second-largest eigenvalue. This ensures that further increasing $M$ does not change the first 2 000 digits of $C^{\text{loop}}$. The computation of the the eigenvalues $\Lambda_\omega$ is much easier, so we compute them to 4 000 digits, although in principle 2 000 digits could suffice.

We reach $L \leq 4$ in the lattice loop model and $L \leq 5$ in the lattice cluster model. This leads to spectra that are in fact too large, with $|S(L)| \sim 3\,000 > N_\ell$. Moreover $\left(\frac{\Lambda_{\text{min}}}{\Lambda_{\text{max}}}\right)^{N_\ell} \ll 10^{-2000}$, making the smallest contribution $\Lambda_{\text{min}}$ to (1.24) indetectable. In order to make the linear system invertible, we truncate it by keeping only the $N_\ell$ largest eigenvalues. We call an amplitude $A_\omega$ well-determined if its relative variation upon a small change of $\ell_{\text{min}}$ is less than $10^{-20}$. In practice, we find that a fraction of about $\frac{1}{2}$ to $\frac{2}{3}$ of the computed amplitudes are well-determined. This fraction turns out to be optimal for $\ell_{\text{min}} \sim 100$.

In fact, we cannot access amplitudes that saturate the bound (4.31), because for $r = L$ through-lines do not have any wiggle room in our lattice, and this prevents us from distinguishing modules with different values of $s$ [34]. In the loop model, this means that we would need $L \geq 4$ in order to access $r = 3$. But in this case, the $r = 3$ eigenvalues are buried too deeply in the transfer matrix spectrum to be accessible in practice. This is why we will only give results for $r \leq 2$.

With these choices, it nevertheless remains true that the considerable range of magnitudes of the terms in (1.24) adversely affects the numerical stability of the linear system if $N_\ell$ is taken too large. In order for the results to have a good precision, we find it necessary to keep $N_\ell < 390$.

## 4.4 Wonderful simplification of amplitude ratios

To summarize Section 4 so far, we have defined two lattice models, introduced a transfer matrix formalism for computing correlation functions, and advertised amplitudes as the right observables to be extracted from numerical results. We will justify these constructions by displaying our numerical results for amplitudes in Section 4.5. Here, we will qualitatively sketch these results, and compare them to earlier results in the same spirit [26, 34].

**Main results and conjecture**

Our main observation is that ratios of amplitudes at different values of the channel weights $w_x$ depend neither on the lattice size $L$, nor on the coupling $K$, nor on the state $\rho$, and coincide with the corresponding ratios of 4-point structure constants. We have observed this in all the examples that we have investigated, which are not many because lattice computations are numerically heavy. We conjecture that this holds for all amplitudes in all 4-point correlation functions. This conjecture is summarized in Eq. (1.25).

The conjecture first means that our amplitude ratio does not depend on the label $\rho$ of states within a given module of the affine Jones–Temperley–Lieb algebra. To be independent

from $\rho$, which we also call to be *constant over the module*, is necessary for an amplitude ratio to be comparable to conformal field theory. The reason is that while modules correspond to representations of the interchiral algebra (with parameters $(r, s)$), states within modules correspond to CFT states only in the critical limit.

The corresponding CFT quantity is a ratio of $s$-channel 4-point structure constants $D_{(r,s)}$. According to the conjecture of Section 2.3, $D_{(r,s)}$ is the product of a reference structure constant that does not depend on channel momenta, with a rational function of the weights $n, w_i, w_x$. Therefore, in our ratio of 4-point structure constants, the reference structure constants cancel, the result is a function of weights (rather than momenta), and this function is rational.

**Further puzzling results**

It is also tempting to study ratios of amplitudes of the type $A_{(r,s),\rho}\left(L\middle|K, n : n', w_i, w_x\right)$, where we vary the weight of contractible loops. We will now give a lattice argument why this ratio should not be constant over the module. In Eq. (1.24), amplitudes are defined by expanding a 4-point function over eigenvalues of the simplified transfer matrix $T_0$. Changing $n$ changes $T_0$ itself and all its eigenvalues, in a way that also depends on the variables $\rho, L, K$. In particular, our ratio is not expected to be constant over the module, and this is what we find numerically.

One may object that the simplified transfer matrix $T_0$ and its eigenvalues also depend on the $s$-channel weight $w_s$, whereas our conjecture (1.25) predicts the simplification of ratios $w_s : w_s'$. However, the dependence of $T_0$ on $w_s$ is only visible in modules with $r = 0$. Whenever $r > 0$, the presence of $2r$ through-lines forbids $s$-channel loops (of weight $w_s$). Therefore, our conjecture should be understood to apply only if $r \neq 0$ or $x \neq s$, which is a minor caveat.

When it comes to ratios of the type $A_{(r,s),\rho}\left(L\middle|K, n, w_i : w_i', w_x\right)$, our lattice argument does not apply. However, there is a CFT argument why this cannot agree with a ratio of structure constants. In the critical limit, the lattice 4-point function (1.24) should agree with a CFT 4-point function. This 4-point function depends non-trivially on $w_i$ not only through structure constants, but also through conformal blocks. On the lattice side, the only dependence on $w_i$ is through amplitudes, and this dependence should therefore account for the conformal blocks. Since a conformal block is a sum over states in a module, we do not expect $A_{(r,s),\rho}\left(L\middle|K, n, w_i : w_i', w_x\right)$ to be constant over the module.

However, in some (but not all) cases, we find that $A_{(r,s),\rho}\left(L\middle|K, n, w_i : w_i', w_x\right)$ is in fact constant over the module, and does not depend on $L, K$. On top of that, it agrees with the corresponding ratio of 4-point structure constants, stripped of their reference prefactors as in Eq. (2.61). In such cases, the ratio is therefore a rational function of $n, w_i, w_i', w_x$. We have no explanation for these puzzling results, which seem too good to be true.

**Comparison with previous work**

There is a priori little reason to suppose that lattice correlation functions could be more than crude approximations of their CFT counterparts, in particular for lattice sizes as small as $L \leq 5$. Nevertheless, in certain 4-point functions of the lattice cluster model, it was found that the $s$-channel spectrum at finite $L$ coincides with the CFT spectrum up to the truncation (4.31) [34]. In the same work, amplitudes were also introduced, but were found to agree with CFT results only in the critical limit, since ratios of the type $w_x : w_x'$ were not computed. Examples of such ratios were considered in [26]: not by explicitly introducing the variables $w_x$, but by studying various 4-point connectivities in the Potts model. These ratios were found to be constant over the module, $L$-independent, and rational in $Q = n^2$.

We are now generalizing these results by introducing the variables $w_i, w_x$ in diagonal 4-point functions, studying the non-diagonal 4-point function $\left\langle V_{(\frac{1}{2},0)}^4 \right\rangle$, considering non-critical couplings, and comparing lattice results with CFT predictions.

## 4.5 Numerical results

**Loop model diagonal 4-point functions $\left\langle \prod_{i=1}^{4} V_{P_i} \right\rangle$**

The diagonal 4-point function depends on 8 loop weights $n, \{w_i\}, \{w_x\}$. However, we use the decomposition (4.5), which is valid for 4-point functions as well as for the corresponding amplitudes, and reduce the problem to studying the amplitures $A^{(0)}, \{A^{(x)}\}$, which depend on 5 loop weights for $A^{(0)}$ (namely $n, \{w_i\}$) and 6 for $A^{(x)}$ (namely $n, \{w_i\}, w_x$). We choose the following reference values and alternative values for the loop weights:

$$w_1 = \sqrt{0.61}, \; w_2 = \sqrt{0.71}, \; w_3 = \sqrt{0.81}, \; w_4 = \sqrt{0.91}, \; w_x = \sqrt{1.21}, \; n = \sqrt{0.51},$$
$$w_1' = \sqrt{1.61}, \; w_2' = \sqrt{1.71}, \; w_3' = \sqrt{1.81}, \; w_4' = \sqrt{1.91}, \; w_x' = \sqrt{1.41}. \tag{4.32}$$

Let us first display the amplitude ratios that are obtained by varying the channel weight $w_x$. We display the first 15 digits, although we actually computed hundreds of digits:

| $(r,s)$ | $A^{(s)}(w_s' : w_s)$ | $A^{(t)}(w_t' : w_t)$ | $A^{(u)}(w_u' : w_u)$ | |
|---|---|---|---|---|
| diag | not constant | zero | zero | |
| $(1,0)$ | 1.029851066998826 | 1.079485644276174 | 1.079485644276174 | |
| $(1,1)$ | 0.880065202821509 | 1.079485644276174 | 1.079485644276174 | (4.33) |
| $(2,0)$ | 1.110388044669465 | 1.252108197040402 | 1.252927747824670 | |
| $(2,\pm\frac{1}{2})$ | zero | 0.986021203400590 | 0.986021203400590 | |
| $(2,1)$ | $-8.924478873210560$ | 1.121066979295078 | 1.121044569052319 | |

Notations:

- The amplitude for the channel diagonal field i.e. $r = 0$ is labelled 'diag'.

- Amplitudes that are not constant over the module are labelled 'not constant'.

- When an amplitude vanishes, there is no ratio to compute, and we write 'zero'.

These results are consistent with the conjecture (1.25): all amplitude ratios are constant on the module, except $A^{(s)}_{\text{diag}}$, since we are computing $s$-channel amplitudes. Moreover, these ratios agree with the 4-point structure constants

$$\frac{D_{(1,0)}}{D_{(1,0)}^{\text{ref}}} \propto -\frac{(w_1 + w_2)(w_3 + w_4)}{w_s + n} + w_t + w_u, \tag{4.34a}$$

$$\frac{D_{(1,1)}}{D_{(1,1)}^{\text{ref}}} \propto \frac{(w_1 - w_2)(w_4 - w_3)}{w_s - n} + w_t - w_u, \tag{4.34b}$$

$$\frac{D_{(2,0)}}{D_{(2,0)}^{\text{ref}}} \propto -\frac{4(w_1^2 + w_2^2 - nw_1w_2 + n^2 - 4)(w_4^2 + w_3^2 - nw_4w_3 + n^2 - 4)}{w_s - n^2 + 2}$$
$$- (n - 2)(w_t + w_u)(w_1 + w_2)(w_3 + w_4) - (n + 2)(w_t - w_u)(w_1 - w_2)(w_4 - w_3)$$
$$+ 2(n^2 - 4)(w_t^2 + w_u^2 + 2w_s - 4), \tag{4.34c}$$

$$\frac{D_{(2,\frac{1}{2})}}{D^{\text{ref}}_{(2,\frac{1}{2})}} \propto n(w_t^2 - w_u^2) - w_t(w_1 w_4 + w_2 w_3) + w_u(w_1 w_3 + w_2 w_4), \tag{4.34d}$$

$$\frac{D_{(2,1)}}{D^{\text{ref}}_{(2,1)}} \propto \frac{4(w_1^2 + w_2^2 + nw_1 w_2 + n^2 - 4)(w_4^2 + w_3^2 + nw_4 w_3 + n^2 - 4)}{w_s + n^2 - 2}$$

$$- (n+2)(w_t + w_u)(w_1 + w_2)(w_3 + w_4) - (n-2)(w_t - w_u)(w_1 - w_2)(w_4 - w_3)$$

$$+ 2(n^2 - 4)(w_t^2 + w_u^2 - 2w_s - 4). \tag{4.34e}$$

These formulas are special cases of Eq. (2.61), with the polynomial terms given by the bootstrap results (3.7). We have omitted constant or $n$-dependent overall factors, which cancel when taking ratios $w_i : w'_i$ or $w_x : w'_x$. The entries of (4.33) follow from inserting (4.34) into (4.6) before taking the ratios.

In addition, we have computed ratios of amplitudes at different weights $w_i$. In contrast to ratios at different $w_x$, this makes sense for $A^{(0)}$ as well as for $A^{(x)}$. The results are

| $(r,s)$ | $A^{(s)}(w'_i : w_i)$ | $A^{(t)}(w'_i : w_i)$ | $A^{(u)}(w'_i : w_i)$ | $A^{(0)}(w'_i : w_i)$ |
|---|---|---|---|---|
| diag | not constant | zero | zero | zero |
| $(1,0)$ | not constant | not constant | not constant | not constant |
| $(1,1)$ | not constant | not constant | not constant | not constant |
| $(2,0)$ | 1.66936126690 | −0.35481068436 | −0.35852887492 | 2.16917615486 |
| $(2,\pm\frac{1}{2})$ | zero | 3.78379075932 | 3.79931935182 | zero |
| $(2,1)$ | 7.28790351811 | 1.68830922070 | 1.68763016857 | 1.04572417389 |

$$\tag{4.35}$$

The ratios that are constant on the module do agree with ratios of 4-point structure constants (4.34). But we did not expect such ratios to be constant, and these results are puzzling.

Finally, after doing our computations at the critical coupling $K_c(n = \sqrt{0.51}) = 0.564877\cdots$, we repeated them at the coupling $K = 0.25$, which is well into the non-critical low-density phase. This did not change the results for the constant ratios.

## A loop model non-diagonal 4-point function $\left\langle V^4_{(\frac{1}{2},0)} \right\rangle$

There are three distinct 4-point functions of the type $\left\langle V^4_{(\frac{1}{2},0)} \right\rangle$, corresponding to the combinatorial maps (4.22). There are two types of closed loops: contractible loops, with weight $n$, and non-contractible loops, with weight $w_x$. We have computed ratios of amplitudes for $n = \sqrt{0.51}$ and $w_x, w'_x = \sqrt{1.21}, \sqrt{1.41}$. Permutation symmetry leads to the vanishing of the amplitudes with $rs$ odd for $\big|\ \big|$ and $\overset{\cdots}{\smile} + \diagup\!\!\!\diagdown$, and to the vanishing of the amplitudes with $rs$ even for $\overset{\cdots}{\smile} - \diagup\!\!\!\diagdown$. These vanishings make the $s$-channel spectra sparser, which allows us to access amplitudes for all $r \le 3$, whereas we were limited to $r \le 2$ in the case of diagonal

4-point functions. The results are:

| $(r,s)$ | $(w'_x : w_x)$ | or $(w'_x : w_x)$ |
|---|---|---|
| diag | not constant | zero |
| $(1,0)$ | 0.954020159934012 | 1 |
| $(1,1)$ | zero | 1 |
| $(2,0)$ | 0.986202017764047 | 0.952674915680381 |
| $(2,\pm\frac{1}{2})$ | zero | 0.980575011958174 |
| $(2,1)$ | 1 | 1.079485644276174 |
| $(3,0)$ | 0.894756612775786 | 1.095611691091229 |
| $(3,\pm\frac{1}{3})$ | zero | 0.968102239881201 |
| $(3,\pm\frac{2}{3})$ | 1 | 0.939824510321535 |
| $(3,1)$ | zero | 0.969479747742282 |

(4.36)

These amplitude ratios agree with the 4-point structure constants from the conformal bootstrap. In the case of , the relevant results are the $s$-channel structure constants in Eq. (3.32). We omit constant or $n$-dependent prefactors, including the reference structure constants, but restore the pole terms from Eq. (2.61):

$$D^{(s)}_{(1,0)} \propto \frac{1}{n+w_s}, \tag{4.37a}$$

$$D^{(s)}_{(2,0)} \propto n^2 - 4 - \frac{4(n-2)^2}{w_s - n^2 + 2}, \tag{4.37b}$$

$$D^{(s)}_{(2,1)} \propto 1, \tag{4.37c}$$

$$D^{(s)}_{(3,0)} \propto -\frac{8}{3}(n-2)^2(n+2)n^2 - \frac{4(n-2)^2 n^4}{n(n^2-3)+w_s}, \tag{4.37d}$$

$$D^{(s)}_{(3,\frac{2}{3})} \propto 1. \tag{4.37e}$$

In the cases of and , the relevant results are the $t$-channel structure constants in Eq. (3.32). Since there is no diagonal field in this channel, the polynomials do not have to be supplemented with pole terms, and can be directly compared with the lattice results.

**Cluster model diagonal 4-point functions $\left\langle \prod_{i=1}^{4} V_{P_i} \right\rangle$**

In the cluster model, we consider combinations of diagonal 4-point functions (4.12) that are even or odd under the reflections $w_i \to -w_i$. In the conformal field theory, such combinations would not make sense, since reflections do not preserve conformal dimensions. It is only normalized structure constants that behave reasonably under reflections, see Eq. (3.1). Fortunately, it is the normalized structure constants that we want to compare with the amplitudes from the cluster model. We therefore have to consider combinations of the type (4.12) for normalized structure constants. A subtlety is that we have only 2 reflection equations for a given structure constant, since the third reflection equation in Eq. (3.1) relates $d^{(s)}_{(r,s)}$ to $d^{(s)}_{(r,s+1)}$. These 2 reflection equations are one fewer than the cluster model's 3 independent parity constraints

(4.10). In order to obtain a third reflection equation, we consider the $s \to s+1$ eigenvectors

$$\frac{1}{2}\left(d_{(r,s)}^{(s)} + (-)^r d_{(r,s+1)}^{(s)}\right).$$ (4.38)

Equivalently, these eigenvectors can be defined as invariants under the simultaneous reflection of $w_1, w_4, w_s, w_u$. After taking combinations of the type (4.12), these become eigenvectors of all 7 reflections $w_i \to -w_i, w_x \to -w_x$, and can be compared with lattice results. Admittedly, we do not know a priori whether we should consider eigenvectors of $s \to s+1$ for the eigenvalue $+1$ or $-1$: our choice of the eigenvalue $(-)^r$ is what will make the comparison work.

From our results (3.7) for normalized $s$-channel structure constants, let us deduce the relevant combinations. We add the pole terms whenever required by Eq. (2.61) in order to obtain ratios $\frac{D}{D^{\mathrm{ref}}}$. Taking combinations as in Eqs. (4.38) and (4.12), we obtain objects that we call $D^{\nu_1, \nu_s, \nu_t, \nu_u}$. We decompose these objects according to Eq. (4.13), and focus on the term $D^{(s), \nu_1, \nu_s}(w_s)$, while neglecting any factors that do not depend on $w_s$:

$$D_{(1,0)}^{(s), \nu_1, \nu_s}(w_s) \propto \frac{w_s^{2-\nu_s}}{w_s^2 - n^2},$$ (4.39a)

$$D_{(2,0)}^{(s), \nu_1, \nu_s}(w_s) \propto \frac{w_s^{2-\nu_s}}{(n^2-2)^2 - w_s^2},$$ (4.39b)

$$D_{(3,0)}^{(s), \nu_1, 0}(w_s) \propto \frac{w_s^2}{n^2(n^2-3)^2 - w_s^2},$$ (4.39c)

$$D_{(3,\frac{1}{3})}^{(s), \nu_1, \nu_s}(w_s) \propto \delta_{1-\nu_s} w_s.$$ (4.39d)

These results are simple and $w_i$-independent, but this is not true for general $(r,s)$, as we can already see in the cases

$$D_{(3,0)}^{(s),1,1}(w_s) \propto \tfrac{2}{3} n^2(n^2-4) w_s$$
$$+ \frac{\left[(n^2-1)w_1^2 + w_2^2 + n^2(n^2-4)\right]\left[(n^2-1)w_3^2 + w_4^2 + n^2(n^2-4)\right]}{n^2(n^2-3)^2 - w_s^2} w_s,$$ (4.39e)

$$D_{(3,0)}^{(s),0,1}(w_s) \propto \tfrac{2}{3} n^2(n^2-4) w_s$$
$$+ \frac{\left[(n^2-1)w_2^2 + w_1^2 + n^2(n^2-4)\right]\left[(n^2-1)w_4^2 + w_3^2 + n^2(n^2-4)\right]}{n^2(n^2-3)^2 - w_s^2} w_s.$$ (4.39f)

Moreover, for $r = 4$, we could infer the analytic expressions of a few structure constants from numerical lattice results:

$$D_{(4,0)}^{(s),1,0}(w_s) \propto \frac{w_s^2}{(n^4 - 4n^2 + 2)^2 - w_s^2},$$ (4.40a)

$$D_{(4,\frac{1}{2})}^{(s),\nu_1,0}(w_s) \propto w_s^2,$$ (4.40b)

$$D_{(4,\frac{1}{2})}^{(s),0,1}(w_s) \propto w_s.$$ (4.40c)

Let us now display numerical lattice results for the amplitude ratios $A^{(s), \nu_1, \nu_s}$, which correspond to the 4-point functions $C^{(s), \nu_1, \nu_s}$ via Eq. (1.24). (In that expression, the eigenvalues $\Lambda_\omega$ are invariant under $w_s \to -w_s$, so we can combine amplitudes just like we combine 4-point functions in Eq. (4.12).) We choose the following parameters:

$$w_1 = \sqrt{0.81}, \ w_2 = \sqrt{1.11}, \ w_3 = \sqrt{0.71}, \ w_4 = \sqrt{0.91},$$
$$w_s = \sqrt{1.21}, \ w_s' = \sqrt{1.27}, \ n = \sqrt{0.51}.$$ (4.41)

| $(r,s)$ | $A^{(s),0,0}(w'_s : w_s)$ | $A^{(s),1,1}(w'_s : w_s)$ | $A^{(s),0,1}(w'_s : w_s)$ | $A^{(s),1,0}(w'_s : w_s)$ |
|---|---|---|---|---|
| diag | 1 | zero | zero | 1 |
| $(1,0)$ | 0.966724662896 | 0.943612364678 | 0.943612364678 | 0.966724662896 |
| $(2,0)$ | 1.115869490901 | not constant | not constant | 1.115869490901 |
| $(3,0)$ | 1.082870872590 | not constant | not constant | 1.082870872590 |
| $(3,\frac{1}{3})$ | zero | 1.024493424507 | 1.024493424507 | zero |
| $(4,0)$ | 1.002830912628 | 0.988626236815 | 0.989211641764 | 0.998033506172 |
| $(4,\frac{1}{2})$ | 1.049586776859 | 1.024493424507 | 1.024493424507 | zero |

$$(4.42)$$

We find that all amplitude ratios that are constant over the module perfectly agree with the analytic formulas, or with numerical bootstrap formulas when no analytic formulas are available: $A^{(s),v_1,v_s}(w'_s : w_s) = D^{(s),v_1,v_s}(w'_s : w_s)$. It is mysterious to us why the ratios are not constant in a few cases. (This phenomenon persists when we consider other combinations of structure constants, such as $A^{(s),1,1} + A^{(s),0,1}$.) We conjecture that the non-constant ratios are numerical artefacts, and that in fact the lattice and bootstrap results exactly coincide for all $r, s$.

For the other terms $A^{(t),v_1,v_t}, A^{(u),v_1,v_u}$, the lattice results are similar to those for $A^{(s),v_1,v_s}$, but they match the bootstrap results only for certain values of the indices $v_1, v_t, v_u$. We believe that this is not a fundamental problem: rather, we may not be considering the correct linear combinations and/or $s \to s + 1$ eigenvectors.

# 5 Outlook

**Exactly solving critical loop models**

We have provided strong evidence for the exact solvability of critical loop models, by exactly determining a fair number of 4-point structure constants. In order to actually solve the models, it may seem that the next logical step is to determine all 4-point structure constants. Given the complexity of the polynomial factors, this looks difficult. And even if this could be done, we would still know only the 4-point functions.

In order to solve a conformal field theory and compute all correlation functions, we actually need 3-point functions. Therefore, we need to decompose 4-point structure constants into 3-point structure constants. It is not clear how to do this in loop models. Certainly this question should be addressed in terms of combinatorial maps, since these objects describe the solutions of crossing symmetry [29].

Having exact epressions is particularly useful for cases where the central charge is rational i.e. $\beta^2 \in \mathbb{Q}$. These cases include physically interesting systems such as polymers and percolation. However, they are hard to study numerically, because structure constants and conformal blocks can have poles, which cancel when computing 4-point functions. Our results on 4-point structure constants could be useful for understanding rational values of $\beta^2$ as limits from non-rational $\beta^2$.

**Exact solvability and degenerate fields**

We have demonstrated exact solvability in two-dimensional CFTs that include only one degenerate field $V^d_{\langle 1,2 \rangle}$, and not the second independent degenerate field $V^d_{\langle 2,1 \rangle}$. However, it turns out that the resulting structure constants are still built from the double Gamma function, which is defined by two independent shift equations (1.5).

This suggests that the field $V^d_{\langle 2,1 \rangle}$ might actually be present in the theory, while playing a more discrete role than in say Liouville theory. And indeed, this degenerate field is known to live at the endpoints of certain topological defects [41]. This suggests that these defects might be used for constraining structure constants, and explaining the appearance of the double Gamma function.

Much further in that direction, we would hope to find conformal field theories with no degenerate field at all, which would nevertheless still be solvable. Even further, two-dimensional theories with only global conformal symmetry.

### Non-unitarity and logarithms

In the literature, much attention has focussed on CFTs such as critical loop models being non-unitary and/or logarithmic. However, as we have demonstrated, what really matters for solvability is the existence of degenerate fields.

From this point of view, unitarity may be reassuring, as it implies that null vectors vanish. In a unitary CFT, any field that has a null vector is therefore degenerate. However, as the examples of minimal models and critical loop models illustrate, unitarity is not a necessary condition for the existence of degenerate fields.

Logarithmic CFTs are a particular class of non-unitary CFTs. They may be frightening because their spaces of states can have complicated algebraic structures. However, when it comes to structure constants, there is no difference between logarithmic and non-logarithmic representations. The non-diagonal primary field $V_{(r,s)}$ belongs to a logarithmic representation if $(r,s) \in \mathbb{N}^* \times \mathbb{Z}^*$, but this does not affect the reference three-point structure constant (1.7). In fact, if $r \in \mathbb{N}^*$, the OPE $V^d_{\langle 2,1 \rangle} V_{(r,0)} \sim \sum_\pm V_{(r,\pm 1)}$ (2.7) produces fields that belong to a logarithmic representation, from $V_{(r,0)}$ which does not. And the OPE determines the structure of that representation [23,42]. Therefore, logarithms can be both conjured and tamed by degenerate fields. This should still be true if $\beta^2 \in \mathbb{Q}$, although logarithmic representations become more intricate [43].

### Exactly solving lattice loop models?

From lattice results, we have extracted certain amplitude ratios that depend neither on lattice size nor on lattice coupling, and exactly coincide with ratios of normalized structure constants. Is this a hint that the lattice models are exactly solvable off-criticality? Maybe not, in view of our results' limitations:

- We have only determined ratios of amplitudes at different values of a channel weight $w_x$. But in generic solutions of crossing symmetry, there is no channel weight, and our results say nothing on the dependence on the contractible loop weight $n$.

- Our results do not depend on lattice size, whereas solving lattice models must involve determining finite-size effects.

Nevertheless, these limitations could conceivably be overcome:

- The truncation (4.31) is an important finite-size effect that does not contradict the size-independence of amplitudes.

- Our polynomials may well be manifestations of the representation theory of diagram algebra, and/or of the combinatorics of finite ensembles of loops. Then they could be computed even in the absence of a channel weight.

# Acknowledgements

We are grateful to Hubert Saleur for many enlightening discussion, for his constant encouragement, and for collaboration on related problems. Moreover, we benefited from useful discussions with John Cardy and Didina Serban. We wish to thank Hubert Saleur, Paul Roux and Yifei He for helpful comments on the draft text.

**Funding information** This work is partly a result of the project ReNewQuantum, which received funding from the European Research Council. It was also supported by the French Agence Nationale de la Recherche (ANR) under grant ANR-21-CE40-0003 (project CONFICA).

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
