# Peer review of "Critical loop models are exactly solvable"

_SciPost Physics, doi:SciPost Phys. 17, 029 (2024)_

## Round 2 · Referee Report · Anonymous (Referee 1) · 2024-5-19

Strengths

1-Study of a large set of 4-pt functions in critical loop models via the bootstrap approach 2- Numerical results clearly hint to some integrable structure of these models 3-The authors provide an impressive combinatorial labeling of the solutions, supporting again the integrable structure of these family of models. 4-Highly non-trivial lattice numerical checks

Weaknesses

1-No particular weakness

Report

The paper represents another important step in solving critical loop models. This research is currently the center of interest for various communities, including those focused on lattice integrable models, conformal field theory, and mathematical probability theory.
In this paper is the continuation of previous papers by the same authors. Instead of counting the possible bootstrap solution, here the authors focus on specific (but still a very large number )solutions providing analytic expression for the structure constants, verified by bootstrap analysis. In particular, they provide an impressive combinatorial way of labeling the results and a convenient basis of solutions where the structure constants display a clear analytical structure. Finally, they implement a transfer matrix formalism, based on the random cluster lattice model, to study diagonal and non-diagonal 4-point correlation functions. This not only allows them to provide an independent check of their results but also offers a physical interpretation of the bootstrap correlation functions. I strongly recommend this paper.

Recommendation

Publish (surpasses expectations and criteria for this Journal; among top 10%)

---

## Round 2 · Referee Report · Anonymous (Referee 2) · 2024-5-24

Report

The complete analytical solution of two-dimensional conformal field theories (CFTs) other than minimal models is an important and difficult open problem. The present paper addresses this issue for a class of "loop models" which includes the Q-state Potts model and the O(n) model as particular cases.

It was originally observed by Delfino and Viti in Ref. [22] that some structure constants of three-point functions (i.e. the OPE coefficients) in the Q-state Potts model can be obtained as the product of a factor coming from Liouville CFT (in its version with central charge smaller than one) times a factor taking into account that Liouville CFT does not possess the color permutational symmetry of the Potts model. In the present paper the same type of factorization is used as an ansatz for the "structure constants" of four-point functions (i.e. the coefficients entering the expansion over conformal blocks). The non-Liouville factor is conjectured to be a polynomial in variables determined by the central charge and the conformal dimensions of the fields. Using an educated guess for the degree of these polynomials, the authors determine their coefficients through an extensive use of the numerical conformal bootstrap (in its two-dimensional version exploiting degenerate fields). The very high numerical accuracy allows them to obtain analytical expressions for the polynomials associated to 235 structure constants arising in a number of four-point functions. This is important evidence for the conjecture that all four-point functions in this class of models can in principle be determined in the same way.

As observed by the authors, the complete solution of the models would correspond to the determination of all OPE coefficients. Extracting them from the structure constants of the four-point functions appears still difficult for these models, but the results of the present paper clearly are an important step forward. The paper also contains a study of loop models on the lattice through the transfer matrix formalism. Very interestingly, the authors numerically determine amplitude ratios that turn out to not depend on lattice parameters and to coincide with ratios of four-point structure constants determined in the continuum.

I definitely recommend publication in SciPost.

Recommendation

Publish (surpasses expectations and criteria for this Journal; among top 10%)

---

## Editorial Decision

published